

# Timelike and gravitational anomalous entanglement from the inner horizon

**Qiang Wen$^{\star\circ}$, Mingshuai Xu$^{\dagger\S}$ and Haocheng Zhong$^{\ddagger\S}$**

Shing-Tung Yau Center and School of Physics, Southeast University, Nanjing 210096, China

$\star$ wenqiang@seu.edu.cn , $\dagger$ xumingshuai@seu.edu.cn , $\ddagger$ zhonghaocheng@outlook.com

## Abstract

In the context of the AdS$_3$/CFT$_2$, the boundary causal development and the entanglement wedge of any boundary spacelike interval can be mapped to a thermal CFT$_2$ and a Rindler AdS$_3$ respectively via certain boundary and bulk Rindler transformations. Nevertheless, the Rindler mapping is not confined in the entanglement wedges. While the outer horizon of the Rindler AdS$_3$ is mapped to the RT surface, we also identify the pre-image of the inner horizon in the original AdS$_3$, which we call the inner RT surface. In this paper we give some new physical interpretation for the inner RT surface. First, the inner RT surface breaks into two pieces which anchor on the two tips of the causal development. Furthermore, we can take the two tips as the endpoints of a certain timelike interval and the inner RT surface is exactly the spacelike geodesic that represents the real part of the so-called holographic timelike entanglement entropy (HTEE). We also identify a timelike geodesic at boundary of the extended entanglement wedge, which represents the imaginary part of the HTEE. Second, in the duality between the topological massive gravity (TMG) and gravitational anomalous CFT$_2$, the entanglement entropy and the mixed state correlation that is dual to the entanglement wedge cross section (EWCS) receive correction from the Chern-Simons term in the TMG. We find that, the correction to the holographic entanglement entropy can be reproduced by the area of the inner RT surface with a proper regulation, while the mixed state correlation can be represented by the saddle geodesic chord connecting the two pieces of the inner RT surface of the mixed state we consider, which we call the inner EWCS. The equivalence between the twist on the RT surface and the length of inner RT surface is also discussed.

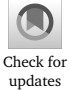

## Contents

---

$^\circ$ Corresponding author.
$^\S$ Co-first authors.

# 1 Introduction

It has long been known that, black holes in classical gravity satisfy thermodynamic laws and the area of the outer horizon $A$ plays the role of the black hole entropy [1–3], which is called the Bekenstein-Hawking entropy,

$$S_{\text{BH}} = \frac{\text{Area}(A)}{4G}. \tag{1}$$

Since then, understanding the microscopic origin of the black hole entropy is believed to be an important window to explore the quantum theory of gravity, and many fruitful steps have been taken towards this task, see [4–6] for an incomplete list.

In the context of the AdS/CFT [7–9], the Bekenstein-Hawking formula is generalized to the Ryu-Takayanagi (RT) formula [10–12] for the entanglement entropy of regions in the dual boundary field theory. More explicitly, consider a static region $\mathcal{A}$ in the boundary CFT, the RT formula gives a holographic formula for the entanglement entropy $S_\mathcal{A}$,

$$S_\mathcal{A} = \frac{\text{Area}(\mathcal{E})}{4G}, \tag{2}$$

where $\mathcal{E}$ is the (minimal) extremal surface in the AdS bulk homologous to the boundary region $\mathcal{A}$. At first, the RT formula was understood for the static spherical regions on the AdS boundary [13], where we can find a Rindler transformation that maps the entanglement wedge $\mathcal{W}_\mathcal{A}$ to a Rindler $\widetilde{\text{AdS}_3}$ black hole, then the entanglement entropy equals to the thermal entropy of the Rindler black hole that is captured by the outer horizon of the Rindler black hole [13].[1] Later, the general case of the RT formula was proved by the Lewkowycz-Maldacena prescription [17], which directly computes the holographic entanglement entropy by applying the replica trick in the gravity side.

Essentially, both the Bekenstein-Hawking and the RT formula are quantum information interpretations for the outer horizon of the black holes, while the analogous aspects associated to the inner horizon are much less understood. So far it is not clear which type of micro-states the inner horizon counts. Nevertheless, there have been observations that strongly indicate that the inner horizon has quantum information interpretation which deserves further investigation:

- It has been observed that the variation of the area of the inner horizon also satisfies a "first law" of thermodynamics for a large class of black holes [18–20].[2]

- It appears that the product of the areas of the inner and outer horizon only depends on the quantized charges and is independent of the mass of the black hole [18, 21–25].

- In the topological massive gravity [26, 27], the entropy of AdS black holes receives a correction from the Chern-Simons term, which is proportional to the area of the inner horizon [28–31].

- More importantly, in the flat limit of the non-extremal BTZ black holes, the outer horizon is pushed to infinity, hence only the inner horizon is left in the bulk, which is called the flat cosmological horizon. In this configuration the asymptotic symmetries are governed by the 3d Bondi-Metzner-Sachs (BMS$_3$) group and the correspondence between the 3d asymptotic flat spacetime and the BMS symmetric field theory (flat$_3$/BMSFT) was proposed [32, 33]. In this context the thermal entropy of the BMS field theory can be calculated by a Cardy-like formula and it matches to the area of the flat cosmological horizon [34, 35]. Also, the analog of the RT formula in this holography has been derived [16] (see also [36]) which introduces two novel null geodesics for the geometric picture of the holographic entanglement entropy.

In this paper we will mainly discuss the inner horizon in the Rindler $\widetilde{\text{AdS}_3}$. The core of the Rindler method is the construction of the Rindler transformations that map the causal development $\mathcal{D}_\mathcal{A}$ of a boundary subregion $\mathcal{A}$ to a boundaryless thermal Rindler space. Since the Rindler transformations are the symmetry transformations of the boundary QFT, the entanglement entropy of $\mathcal{A}$ equals to the thermal entropy of the Rindler space. In holography, these Rindler transformations can be extended into the AdS bulk, which maps the entanglement

---

[1]This is called the Rindler method to compute the holographic entanglement entropy, see also [14–16] for further progress on the Rindler method applied to holography beyond AdS/CFT.

[2]Such a "first law" for the inner horizons has negative energy or temperature.

wedge $\mathcal{W}_{\mathcal{A}}$ of $\mathcal{A}$ to a Rindler $\widetilde{\text{AdS}_3}$. In this case, the Bekenstein-Hawking entropy, given by the area of the outer horizon of the Rindler $\widetilde{\text{AdS}_3}$, gives the entanglement entropy $S_{\mathcal{A}}$. Also we can read the modular flow from the Rindler transformations, which is a geometric flow generated by the modular Hamiltonian and is very useful in our discussions.

As we know that, the inverse Rindler transformations exactly map the outer horizon of the Rindler $\widetilde{\text{AdS}_3}$ to the RT surface $\mathcal{E}_{\mathcal{A}}$ in the original AdS spacetime. Then it is very interesting to ask what is the image of the inner horizon in the original AdS spacetime. For any spacelike interval $\mathcal{A}$, there is a unique timelike interval $\widehat{\mathcal{A}}$ that connects the two tips of $\mathcal{D}_{\mathcal{A}}$. Recently, the holographic entanglement entropy associated with timelike intervals has been studied in [37, 38] (see also [39–48] for relevant discussions). The geometric picture includes two spacelike geodesics anchored on the endpoints of the timelike interval whose length makes the real part of the holographic timelike entanglement entropy, while the imaginary part is given by the length of a timelike geodesic connects the spacelike one at the null infinities. We identify the image of the Rindler inner horizon in the original AdS spacetime, and show that it coincides with the spacelike geodesics in the geometric picture of the holographic timelike entanglement entropy for $\widehat{\mathcal{A}}$. We also find that the timelike geodesic representing the geometric picture of the imaginary part is the boundary of the extended entanglement wedge, which is mapped to the exterior of the inner horizon in the Rindler $\widetilde{\text{AdS}_3}$.

Considering the correction to holographic entanglement entropy due to higher-order curvature terms in the bulk is also interesting. Two typical higher order corrections to the Hilbert-Einstein action are Gauss-Bonnet term [49] and Chern-Simons term [26, 27]. In this paper, we mainly focus on the Chern-Simons term, which is the low energy effective action of string theory. The Chern-Simons term, which explicitly depends on the connection, is not manifestly diffeomorphism invariant (up to a boundary term). This leads to unequal central charges $c_L$ and $c_R$ in the dual conformal field theory. Such a theory is called the conformal field theory with gravitational anomaly. In [50], the authors give a bulk description of the holographic entanglement entropy with gravitational anomaly, which is a spinning particle with a worldline action. However, the worldline action is not a pure geometric quantity, which is the main motivation of this work.

The paper is organized as follows. In Sec.2, we give a brief introduction of the Rindler method and give the explicit Rindler mapping. Then we identify the pre-image of the inner horizon in the original AdS$_3$, which we call the inner RT (IRT) surface. In Sec.3, we briefly introduce the partial entanglement entropy (PEE) and the slicing structure of the entanglement wedge and the extended entanglement wedge. Based on the fine structure we established the partnership between the points in the causal development and the points on the RT and the inner RT surfaces, based on which we further establish the duality between the PEE (or timelike PEE) and the geodesic chords on the RT (or inner RT) surfaces. In Sec.4, in the context of the correspondence between the topological massive gravity (TMG) and the gravitational anomalous CFT$_2$, we give the bulk geometric description of the holographic entanglement entropy, which is the length of a geodesic chord on the IRT surface. This result is consistent with the one obtained by measuring the "twist" of the RT surface introduced in [50]. In Sec.5, we give the geometric picture for the gravitational anomalous correction to the balanced partial entanglement entropy (BPE), which was proposed to be the mixed state correlation dual to the EWCS. The geometric picture is just the saddle geodesic connecting the two pieces of the IRT surface of the mixed state region, and gives the same result as the one obtained by measuring the twist of the EWCS [51]. In Sec.6, we construct the PEE in the twist description based on the fine structure, and the result is consistent with the length of the partner geodesic chord on the IRT surface. We summarize our results and give discussion in the last section. In the appendix, we give details of some calculations and the comparison between our geometric description and the swing surface prescription [16, 52, 53] when taking the flat limit.

Through out this paper, we will put a tilde sign on the top of all the quantities in the Rindler $\widetilde{\text{AdS}_3}$, and put a hat sign on the top of all the quantities associated to the inner RT surface.

# 2 Timelike entanglement entropy from the inner horizon

## 2.1 A brief introduction to the Rindler method in AdS$_3$/CFT$_2$

The key point of the Rindler method is to construct a Rindler transformation, which maps the entanglement entropy to the thermal entropy by a symmetry transformation, such that the calculation of the entanglement entropy is equivalent to calculating the corresponding thermal entropy. Comparing with the entanglement entropy, the calculation of thermal entropy which we already have tools to evaluate is much easier. For example, the Cardy formula [54–58] for the field theory side, and the Bekenstein-Hawking formula for the gravity side. The general strategy to construct a Rindler transformation by the symmetries of the boundary field theory and its bulk extension by holographic dictionary is summarized in Sec.2 of [16]. Here we only list some key points of the Rindler method applied to the AdS$_3$/CFT$_2$ correspondence. First, we consider a spatial interval $\mathcal{A}$ in the vacuum CFT$_2$ on the plane,

$$\mathcal{A} : (-l_U/2, -l_V/2) \to (l_U/2, l_V/2) \qquad (l_U, l_V > 0) , \tag{3}$$

where $U$ and $V$ are the light-cone coordinates,

$$U = \frac{x+t}{2}, \qquad V = \frac{x-t}{2} . \tag{4}$$

In this paper, we set the AdS radius $\ell = 1$. The vacuum CFT$_2$ on the plane is dual to the Poincaré AdS$_3$ with the metric given by[3]

$$ds^2 = 2\rho \, dU dV + \frac{d\rho^2}{4\rho^2} . \tag{7}$$

The causal development $\mathcal{D}_{\mathcal{A}}$ of the interval $\mathcal{A}$ is a diamond shape region,

$$\mathcal{D}_{\mathcal{A}} = \left\{ (U, V) \mid -\frac{l_U}{2} < U < \frac{l_U}{2}, -\frac{l_V}{2} < V < \frac{l_V}{2} \right\} . \tag{8}$$

The bulk Rindler transformation maps the Poincaré AdS$_3$ (7) to a Rindler $\widetilde{\text{AdS}_3}$,

$$ds^2 = T_{\tilde{U}}^2 d\tilde{U}^2 + 2\tilde{\rho} \, d\tilde{U} d\tilde{V} + T_{\tilde{V}}^2 d\tilde{V}^2 + \frac{d\tilde{\rho}^2}{4\left(\tilde{\rho}^2 - T_{\tilde{U}}^2 T_{\tilde{V}}^2\right)} , \tag{9}$$

---

[3]Under the following coordinate transformation,

$$U = \frac{x+t}{2}, \quad V = \frac{x-t}{2}, \quad \rho = \frac{2}{z^2} , \tag{5}$$

the metric (7) can be rewritten in the Poincaré coordinate,

$$ds^2 = \frac{-dt^2 + dx^2 + dz^2}{z^2} . \tag{6}$$

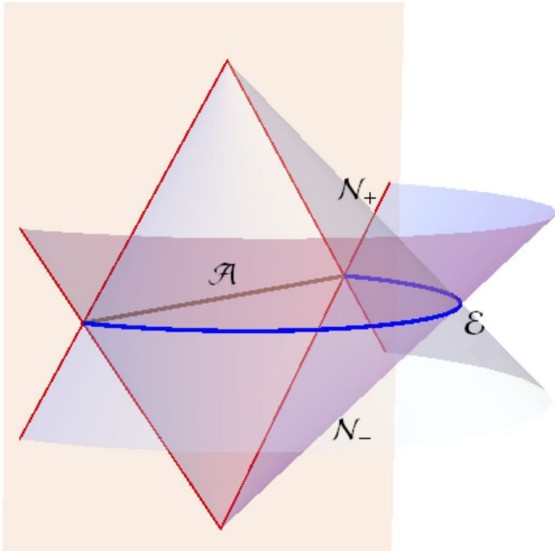

Figure 1: The figure is extracted from [59]. The pink surfaces represent the null hypersurfaces $\mathcal{N}_\pm$, and the blue curve represents the RT surface $\mathcal{E}$.

and the explicit transformation is given by [16, 59],

$$
\begin{aligned}
\tilde{U} &= \frac{1}{4T_{\tilde{U}}} \log\left[\frac{(2-(l_U+2U)(l_V-2V)\rho)(2+(l_U+2U)(l_V+2V)\rho)}{(2-(l_U-2U)(l_V+2V)\rho)(2+(l_U-2U)(l_V-2V)\rho)}\right], \\
\tilde{V} &= \frac{1}{4T_{\tilde{V}}} \log\left[\frac{(2-(l_V+2V)(l_U-2U)\rho)(2+(l_V+2V)(l_U+2U)\rho)}{(2-(l_V-2V)(l_U+2U)\rho)(2+(l_V-2V)(l_U-2U)\rho)}\right], \\
\tilde{\rho} &= \frac{T_{\tilde{U}}T_{\tilde{V}}\left(4+16UV\rho+\left(l_U^2-4U^2\right)\left(l_V^2-4V^2\right)\rho^2\right)}{4\rho l_U l_V},
\end{aligned}
\tag{10}
$$

where $T_{\tilde{U}}$ and $T_{\tilde{U}}$ are the parameters characterizing the size of a thermal circle of the outer horizon,

$$
\text{outer horizon: } (\tilde{U},\tilde{V}) \sim \left(\tilde{U}+i\frac{\pi}{T_{\tilde{U}}},\ \tilde{V}-i\frac{\pi}{T_{\tilde{V}}}\right).
\tag{11}
$$

The bulk Rindler transformation (10) reduces to the boundary Rindler transformation at the asymptotic boundary $\rho \to \infty$,

$$
\tilde{U} = \frac{\beta_{\tilde{U}}}{\pi}\operatorname{arctanh}\left(\frac{2U}{l_U}\right), \quad \tilde{V} = \frac{\beta_{\tilde{V}}}{\pi}\operatorname{arctanh}\left(\frac{2V}{l_V}\right),
\tag{12}
$$

where $\beta_{\tilde{U}} = \pi/T_{\tilde{U}}$ and $\beta_{\tilde{V}} = \pi/T_{\tilde{V}}$. Similarly, the boundary Rindler transformation maps the vacuum $\text{CFT}_2$ on the plane to a thermal $\text{CFT}_2$. Following the logic of the Rindler method, the entanglement entropy of the interval $\mathcal{A}$ (3) is mapped to the thermal entropy of the thermal $\text{CFT}_2$, which can be evaluated holographically. According to the bulk Rindler transformation (10), the outer horizon $\tilde{\rho} = T_{\tilde{U}}T_{\tilde{V}}$ of Rindler $\widetilde{\text{AdS}_3}$ is mapped from two null hypersurfaces $\mathcal{N}_\pm$ in the original $\text{AdS}_3$ space,

$$
\mathcal{N}_+: \rho = \frac{2}{(l_U+2U)(l_V-2V)}, \quad \mathcal{N}_-: \rho = \frac{2}{(l_U-2U)(l_V+2V)},
\tag{13}
$$

whose intersection is precisely the RT surface $\mathcal{E}$,

$$
\mathcal{E}: \rho = \frac{2l_V}{l_U\left(l_V^2-4V^2\right)}, \quad U = \frac{l_U}{l_V}V.
\tag{14}
$$

See Fig.1 for an illustration. The length parameter $\tau$ of the RT surface $\mathcal{E}$ is given by

$$\tau(V) = \text{arctanh}\left(\frac{2V}{l_V}\right), \tag{15}$$

where we set the center point of $\mathcal{E}$ to be the origin for the length parameter. The entanglement wedge $\mathcal{W}_\mathcal{A}$ [60] is the bulk region enclosed by $\mathcal{D}_\mathcal{A}$ and $\mathcal{N}_\pm$, which is mapped to the exterior of the outer horizon of Rindler $\widetilde{\text{AdS}_3}$ (9) under the bulk Rindler transformation.[4] Therefore, the thermal entropy of the boundary thermal $\text{CFT}_2$ equals to the Bekenstein-Hawking entropy of the outer horizon in Rindler $\widetilde{\text{AdS}_3}$ space. Now we denote the image of boundary interval $\mathcal{A}$ (3) under the Rindler transformation (12) as $\tilde{\mathcal{A}}$. Due to the divergence of the entanglement entropy, we need to introduce cutoffs to regulate the interval $\mathcal{A}$,

$$\mathcal{A}^{\text{reg}} : (-l_U/2 + \epsilon_U, -l_V/2 + \epsilon_V) \to (l_U/2 - \epsilon_U, l_V/2 - \epsilon_V). \tag{16}$$

Here, $\epsilon_U$ and $\epsilon_V$ are two infinitesimal constants of the same order.[5] After performing the Rindler transformation (12), the image of this regularized interval $\mathcal{A}^{\text{reg}}$ is

$$\tilde{\mathcal{A}}^{\text{reg}} : \left(-\Delta\tilde{U}/2, -\Delta\tilde{V}/2\right) \to \left(\Delta\tilde{U}/2, \Delta\tilde{V}/2\right),$$
$$\Delta\tilde{U} = \frac{1}{T_{\tilde{U}}}\log\left(\frac{l_U}{\epsilon_U}\right), \quad \Delta\tilde{V} = \frac{1}{T_{\tilde{V}}}\log\left(\frac{l_V}{\epsilon_V}\right). \tag{17}$$

The proper length on the outer horizon in the Rindler $\widetilde{\text{AdS}_3}$ is $ds^2 = \left(T_{\tilde{U}}d\tilde{U} + T_{\tilde{V}}d\tilde{V}\right)^2$. Integration along the proper length gives the length of the outer horizon

$$\ell_{\text{outer horizon}} = T_{\tilde{U}}\Delta\tilde{U} + T_{\tilde{V}}\Delta\tilde{V} = \log\left(\frac{l_U l_V}{\epsilon_U \epsilon_V}\right). \tag{18}$$

Therefore, the holographic entanglement entropy of the interval $\mathcal{A}$ is given by

$$S_\mathcal{A} = \frac{1}{4G}\log\left(\frac{l_U l_V}{\epsilon_U \epsilon_V}\right), \tag{19}$$

which exactly matches the result evaluated by the RT formula [10–12]. On the other hand, a useful byproduct of the Rindler method is the modular flow that manifests the replica symmetry. More explicitly, the Rindler transformation $\tilde{x} = f(x)$ is invariant under imaginary identification of the Rindler coordinates $\tilde{x}^i \sim \tilde{x}^i + i\beta_{\tilde{x}^i}$, which can be referred to as the thermal circle in the Rindler space. With the thermal circle, the modular flow $k_t$ along the thermal circle in the Rindler space can be rewritten as $k_t = \beta_{\tilde{x}^i}\partial_{\tilde{x}^i}$. For example, for the thermal circle of the outer horizon (11), the corresponding bulk modular flow, which is also called the bulk modular Hamiltonian flow, is given by

$$k_t^{\alpha,\text{bulk}} = \beta_{\tilde{U}}\partial_{\tilde{U}} - \beta_{\tilde{V}}\partial_{\tilde{V}} \tag{20}$$
$$= \frac{\pi}{2}\left(l_U - \frac{4U^2}{l_U} - \frac{2}{l_V \rho}\right)\partial_U - \frac{\pi}{2}\left(l_V - \frac{4V^2}{l_V} - \frac{2}{l_U \rho}\right)\partial_V + 4\pi\left(\frac{U}{l_U} - \frac{V}{l_V}\right)\rho\,\partial_\rho,$$

where the negative sign in the first line comes from the negative sign in the thermal circle (11). The superscript $\alpha$ is used to distinguish it from a new bulk modular flow $k_t^{\beta,\text{bulk}}$, which we propose in the next subsection. Interestingly, the null hypersurfaces $\mathcal{N}_\pm$, which are mapped to the outer horizon in Rindler $\widetilde{\text{AdS}_3}$, are just the Killing horizons of the bulk modular Hamiltonian

---

[4]This can be seen clearly in Sec.3.2, where we introduce the concept of modular slice.

[5]The symbols $\epsilon_U, \epsilon_V$ in this paper all represent two infinitesimal constants of the same order.

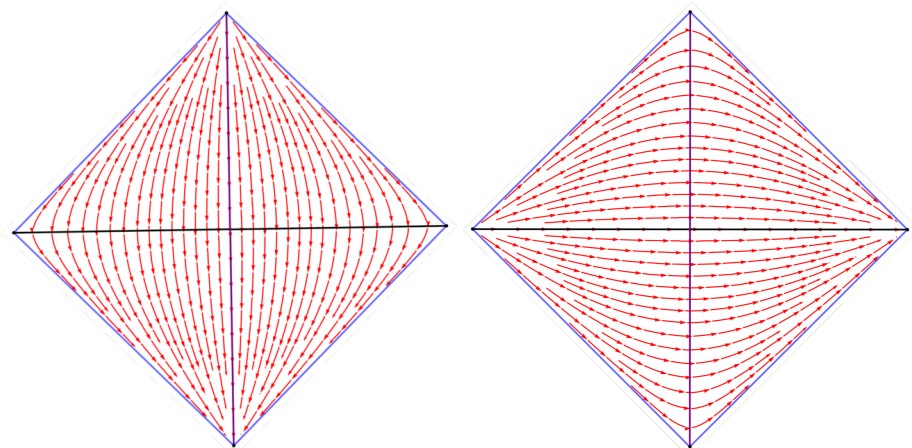

Figure 2: The left and the right figures show the modular flow lines of $k_t^\alpha$ and $k_t^\beta$ in the causal development $\mathcal{D}_\mathcal{A}$, respectively. The black line represent the interval $\mathcal{A}$, and the purple line represent its partner interval $\widehat{\mathcal{A}}$.

flow $k_t^{\alpha,\text{bulk}}$. Furthermore, the RT surface $\mathcal{E}$ is precisely the fixed points of $k_t^{\alpha,\text{bulk}}$ (or the bulk replica symmetry)

$$k_t^{\alpha,\text{bulk}}|_{\mathcal{E}} = 0\,. \tag{21}$$

Asymptotically, we have the boundary modular flow $k_t^\alpha$,

$$k_t^\alpha = \left(\frac{\pi l_U}{2} - \frac{2\pi U^2}{l_U}\right)\partial_U - \left(\frac{\pi l_V}{2} - \frac{2\pi V^2}{l_V}\right)\partial_V\,. \tag{22}$$

See the left figure of Fig.2 for the boundary modular Hamiltonian flow lines generated by $k_t^\alpha$.

In fact, we can also define the temporal and the spatial coordinates in the Rindler $\widetilde{\text{AdS}}_3$

$$\text{Rindler time: } \tilde{t} := \pi\left(\frac{\tilde{U}}{\beta_{\tilde{U}}} - \frac{\tilde{V}}{\beta_{\tilde{V}}}\right) = \frac{1}{4}\log\left(\frac{((l_U+2U)(l_V-2V)\rho-2)^2}{((l_U-2U)(l_V+2V)\rho-2)^2}\right),$$

$$\text{Rindler space: } \tilde{x} := \pi\left(\frac{\tilde{U}}{\beta_{\tilde{U}}} + \frac{\tilde{V}}{\beta_{\tilde{V}}}\right) = \frac{1}{4}\log\left(\frac{((l_U+2U)(l_V+2V)\rho+2)^2}{((l_U-2U)(l_V-2V)\rho+2)^2}\right), \tag{23}$$

where we have used the bulk Rindler transformation (10). In this Rindler coordinate system $(\tilde{t}, \tilde{x})$, the thermal circle of the outer horizon (11) is just $\tilde{t} \sim \tilde{t} + 2\pi i$, and the modular Hamiltonian flow $k_t^{\alpha,\text{bulk}}$ can be rewritten as,

$$k_t^{\alpha,\text{bulk}} = 2\pi\partial_{\tilde{t}}\,. \tag{24}$$

In other words, the modular Hamiltonian flow $k_t^{\alpha,\text{bulk}}$ is precisely the generator of the time translation in the Rindler $\widetilde{\text{AdS}}_3$, which is also why it is called the modular Hamiltonian flow, whose modular flow lines are just the Rindler temporal coordinate lines when mapped to the Rindler $\widetilde{\text{AdS}}_3$.

## 2.2 The inner RT surface and the inner horizon in the Rindler $\widetilde{\text{AdS}}_3$

In the last subsection, we reviewed that the outer horizon $\tilde{\rho} = T_{\tilde{U}}T_{\tilde{V}}$ in the Rindler $\widetilde{\text{AdS}}_3$ is mapped from the null hypersurfaces $\mathcal{N}_\pm$ (13), whose intersection line is the RT surface $\mathcal{E}$ (14). Furthermore, the region outside the outer horizon in the Rindler $\widetilde{\text{AdS}}_3$ is mapped from the

entanglement wedge $\mathcal{W}_\mathcal{A}$ in the original Poincaré AdS$_3$. Since the bulk Rindler transformation is not confined within the entanglement wedge,[6] and there is also a inner horizon in the Rindler $\widetilde{\text{AdS}}_3$, it is also interesting to conduct a similar analysis associated to the inner horizon. For example, what is the pre-image of this inner horizon, as well as the region between the outer and the inner horizons, in the original AdS$_3$. What is the physical interpretation of these pre-images? Is there a modular flow, or a replica story associated to the inner horizon?

First, the thermal circle associated to the inner horizon $\tilde{\rho} = -T_{\tilde{U}} T_{\tilde{V}}$ is given by

$$\text{inner horizon: } (\tilde{U}, \tilde{V}) \sim \left( \tilde{U} + i\frac{\pi}{T_{\tilde{U}}}, \, \tilde{V} + i\frac{\pi}{T_{\tilde{V}}} \right), \tag{25}$$

which corresponds to a geometric flow (or modular momentum flow) $k_t^{\beta,\text{bulk}}$ generated by the so-called modular momentum $P_{\text{mod}}$,[7] whose Killing horizon is just the inner horizon,

$$
\begin{aligned}
k_t^{\beta,\text{bulk}} &= \beta_{\tilde{U}} \partial_{\tilde{U}} + \beta_{\tilde{V}} \partial_{\tilde{V}} \\
&= \frac{\pi}{2}\left( l_U - \frac{4U^2}{l_U} + \frac{2}{l_V \rho} \right) \partial_U + \frac{\pi}{2}\left( l_V - \frac{4V^2}{l_V} + \frac{2}{l_U \rho} \right) \partial_V + 4\pi\left( \frac{U}{l_U} + \frac{V}{l_V} \right) \rho \partial_\rho \,.
\end{aligned}
\tag{26}
$$

Similarly, the positive sign in the first line comes from the positive sign in the thermal circle (25). The difference between $k_t^{\alpha,\text{bulk}}$ and $k_t^{\beta,\text{bulk}}$ is that $l_V$ is replaced with $-l_V$, i.e.

$$V \leftrightarrow -V \Leftrightarrow t \leftrightarrow x, \tag{27}$$

which is understandable since the $t$ direction becomes spacelike while the $x$ direction becomes timelike after we enter the outer horizon. Asymptotically, we have the boundary modular momentum flow $k_t^\beta$,

$$k_t^\beta = \left( \frac{\pi l_U}{2} - \frac{2\pi U^2}{l_U} \right) \partial_U + \left( \frac{\pi l_V}{2} - \frac{2\pi V^2}{l_V} \right) \partial_V \,. \tag{28}$$

See the right figure of Fig.2 for the boundary modular flow lines generated by $k_t^\beta$. The physical significance of $k_t^{\beta,\text{bulk}}$ can be understood more clearly in the Rindler coordinate system $(\tilde{t}, \tilde{x})$, where the modular flow $k_t^{\beta,\text{bulk}}$ can be rewritten as:

$$k_t^{\beta,\text{bulk}} = 2\pi \partial_{\tilde{x}} \,, \tag{29}$$

i.e. the bulk modular flow $k_t^{\beta,\text{bulk}}$ is the generator of the spatial translation in the Rindler $\widetilde{\text{AdS}}_3$, hence we refer to it as the bulk modular momentum flow. In the Rindler coordinate system $(\tilde{t}, \tilde{x})$, the thermal circle of the inner horizon (25) is $\tilde{x} \sim \tilde{x} - 2\pi i$.

Second, solving the equation $\tilde{\rho} = -T_{\tilde{U}} T_{\tilde{V}}$ can also give us two null hypersurfaces $\mathcal{M}_\pm$, which are mapped to the inner horizon under the bulk Rindler transformation (10),

$$\mathcal{M}_+ : \rho = -\frac{2}{(l_U - 2U)(l_V - 2V)}, \quad \mathcal{M}_- : \rho = -\frac{2}{(l_U + 2U)(l_V + 2V)}, \tag{30}$$

and their interaction line is $\widehat{\mathcal{E}}$ which we refer to as the inner Ryu-Takayanagi (IRT) surface in this paper,

$$\widehat{\mathcal{E}} : \rho = -\frac{2l_V}{l_U \left( l_V^2 - 4V^2 \right)}, \quad U = -\frac{l_U}{l_V} V, \tag{31}$$

---

[6]We will return to this point in Sec.3.2.2.

[7]In appendix A, we derive the modular Hamiltonian $H_{\text{mod}}$ and the modular momentum $P_{\text{mod}}$ as the integrals of the stress tensor based on the Rindler method.

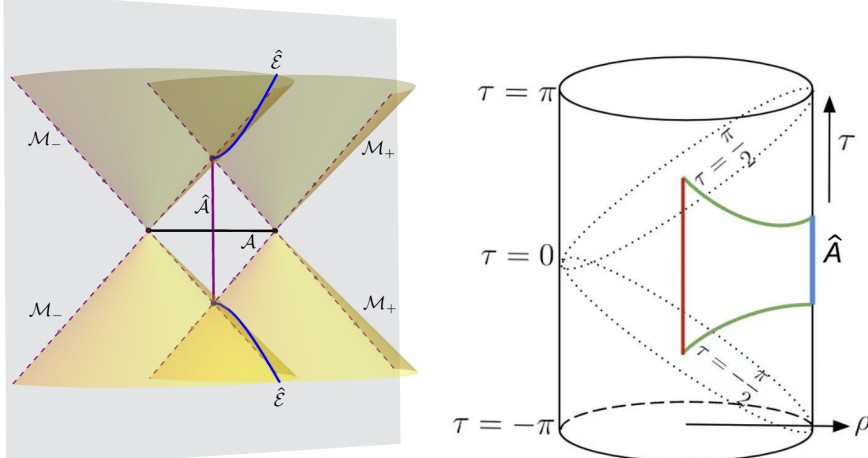

Figure 3: The right figure is extracted from [61]. The left figure: the blue curve is the IRT surface that is the intersection line between the two null hypersurfaces $\mathcal{M}_\pm$. The right figure: the green solid line represents the real part of the timelike entanglement entropy, and the red solid line represents the imaginary part, which correspond to the spacelike geodesic and the timelike geodesic respectively.

see the left figure of Fig.3 for an illustration. The length parameter $\hat{\tau}$ of the IRT surface $\widehat{\mathcal{E}}$ is given by

$$\widehat{\tau} = \frac{1}{2} \log\left(\frac{2V - l_V}{2V + l_V}\right), \tag{32}$$

where we set the null infinity (i.e. $V = \infty$) to be the origin for the length parameter. One can easily verify that $k_t^{\beta,\text{bulk}}$ is vanishing on $\widehat{\mathcal{E}}$. Furthermore, the IRT surface $\widehat{\mathcal{E}}$ intersects with the asymptotic boundary at the endpoints of a boundary timelike interval $\widehat{\mathcal{A}}$, which connects the two tips of the causal development $\mathcal{D}_\mathcal{A}$ and is referred to as the *partner interval* of $\mathcal{A}$ in this paper

$$\widehat{\mathcal{A}} : (-l_U/2, l_V/2) \to (l_U/2, -l_V/2). \tag{33}$$

If we consider a case when $\mathcal{A}$ is a static spacelike interval of length $L$, i.e. $l_U = l_V = L/2$, then the corresponding partner interval $\widehat{\mathcal{A}}$ is a pure timelike interval[8] of the same length (which means the time difference) $L$. Under the $(t, x, z)$ coordinates (5), the corresponding IRT surface $\widehat{\mathcal{E}}$ (31) is given by

$$\widehat{\mathcal{E}} : t^2 - z^2 = \frac{L^2}{4}. \tag{34}$$

Recently, the authors in [37,38] defined the so-called timelike entanglement entropy (TEE) through the analytical continuation of the entropy formula for spacelike intervals. Specifically, consider a pure timelike interval $\widehat{\mathcal{A}}$ of length $L$, the TEE $S_{\widehat{\mathcal{A}}}^{(T)}$ of this pure timelike interval $\widehat{\mathcal{A}}$ can be evaluated via the analytical continuation for the spacelike entanglement entropy, i.e.,

$$S_{\widehat{\mathcal{A}}}^{(T)} = \frac{c}{3} \log\left(\frac{\sqrt{-L^2}}{\epsilon}\right) = \frac{c}{3} \log\left(\frac{L}{\epsilon}\right) + \frac{c\pi i}{6}. \tag{35}$$

Here, $\epsilon$ represents the UV cutoff, $\sqrt{-L^2}$ denotes $\sqrt{ds^2}$ for the pure timelike interval $\widehat{\mathcal{A}}$, and $c$ is the central charge. Accordingly, the geometric picture of the real part of the TEE, was proposed [37, 38] to be represented by the analytical continuation (i.e. Wick rotation) for

---

[8]Here "pure timelike" refers to the fact that $\widehat{\mathcal{A}}$ is located at the $x = 0$ slice.

the RT surface connecting the timelike interval $\hat{\mathcal{A}}$ (see the green lines in the right figure of Fig.3).[9] Later, various principles have been proposed to determine the geometric picture for the holographic TEE.[10]

Interestingly, this geometric picture exactly coincides with the IRT surface $\widehat{\mathcal{E}}$ (34). Note that, the statistical interpretation in terms of density matrix for the timelike entanglement entropy is not clear in the literature (see [37,38,63,64] for relevant discussions). Here we argue that, if we take the IRT surface as the geometric picture for the timelike entanglement entropy, then the timelike entanglement entropy can be associated to a bulk replica algorithm similar to the Lewkowycz-Maldacena (LM) prescription [17], hence it can be taken as a holographic entropic quantity. In Euclidean signature, the LM prescription applies the replica trick on a bulk region in the gravitational theory, by cutting $n$ copies of the bulk open along this region, then cyclically gluing the $n$ copies of the bulk into a $n$-manifold. This bulk picture is just the holographic dual of a boundary replica algorithms that computes the von Neumann entropy of a boundary region $\mathcal{A}$, and the cut open bulk region is just a time slice of the corresponding entanglement wedge $\mathcal{W}_A$ of $\mathcal{A}$. Let us denote the partition function on the $n$-manifold as $Z_n$, and analytically continue $n$ to a positive real number, then the von Neumann entropy $S_{\mathcal{A}}$ is holographically calculated by the replica algorithm,

$$S_{\text{HEE}} = \lim_{n \to 1} \frac{1}{1-n} \left( \log Z_n - n \log Z_1 \right) . \tag{36}$$

When taking the $n \to 1$ limit, (36) can be calculated by studying the action of a bulk geometry with a conical singularity with open angle $2\pi(n-1)$ along a worldline $\mathcal{C}$ extending into the bulk [17],

$$S_{\text{HEE}} = - \left. \partial_n (S_{\text{cone}}) \right|_{n=1} . \tag{37}$$

Remarkably, the authors of [17] found that: 1) due to the consistency of the bulk equations of motion, the worldline $\mathcal{C}$ should be an extremal surface; 2) the entanglement entropy is proportional to the length of $\mathcal{C}$, i.e. $S_{\text{HEE}} = \text{length}(\mathcal{C})/(4G)$. These indeed make a proof for the RT formula.

One can keep track of the whole story by looking at the thermal circle, or the $\tau$ circle as it is usually parameterized by $\tau$, which is a real circle with $\tau \sim \tau + 2\pi$ period in Euclidean signature, and the modular Hamiltonian generates the translation along the $\tau$ circle. After the cyclic gluing of the bulk, the period of thermal circle becomes $\tau \sim \tau + 2n\pi$, hence conical singularity arises at the position where the circle shrinks, which is exactly the extremal surface $\mathcal{C}$. Note that, the IRT surface is also an extremal surface, and furthermore there is also a thermal circle (25) that shrinks at it. Also in Euclidean signature the "timelike" interval $\widehat{\mathcal{A}}$, is also spacelike, so there is no obstacle to apply the LM prescription to the IRT surface and compute (36). In other words, we take the IRT surface as the bulk extremal surface $\mathcal{C}$ in the LM story. The corresponding asymptotic story on the boundary is just cutting $\widehat{\mathcal{A}}$ open and gluing $n$ copies of the boundary cyclically, which is equivalent to inserting twist operators at

---

[9]Furthermore, the difference between the covariant RT surface (14) and the covariant IRT surface (31) is also simply $l_V \leftrightarrow -l_V$ or equivalently $l_t \leftrightarrow l_x$. In other words, the covariant IRT surface (31) can also be obtained by applying a Wick rotation to the covariant RT surface (14), and thus aligns with the result obtained by covariantizing the static case.

[10]For example, in [39], the authors generalize the RT formula by extending the concept of the area of homologous surfaces to include complex values. Specifically, the area of the timelike part is considered imaginary, while that of the spacelike part remains real. When comparing the complex-valued areas of homologous surfaces, the imaginary part is dominant. In [40, 41, 62], the authors claimed that the first order derivative of the timelike part and that of the spacelike part should be equal at the merging point. In this way, when we extremalize the area of the timelike part of the homologous surface, the spacelike part will also be extremalized at the same time due to this requirement. Also in [42], the authors proposed that, the holographic timelike entanglement entropy is captured by the boundary-anchored extremal surfaces in the analytic continuation of the holographic spacetimes with complex coordinates.

the boundary points of $\widehat{\mathcal{A}}$. Then (36) is equivalent to calculating the two-point function of twist operators inserted at the two endpoints of $\widehat{\mathcal{A}}$. Since the quantity that generates the translation along the thermal circle becomes the modular momentum, the physical interpretation for the timelike entanglement should be associated to the modular momentum,[11] rather than the modular Hamiltonian. Also, we expect that the timelike entanglement entropy also satisfies a thermodynamic first law analog to the one associated to the inner horizon.

The geometric picture of the imaginary part proposed in [37, 38], referred to as $\widehat{\mathcal{E}}_{\text{im}}$, is a timelike geodesic that connects the endpoints of the two pieces of the spacelike geodesics $\widehat{\mathcal{E}}$ at the past and future null infinities (see the red line in the right figure of Fig.3), such that $\widehat{\mathcal{E}} \cup \widehat{\mathcal{E}}_{\text{im}}$ remains connected. Interestingly, the length of this timelike geodesic reproduces the imaginary part of the TEE obtained from the analytical continuation [37,38]. However, unlike the real part, the imaginary part is independent of the length $L$ of the timelike interval $\widehat{\mathcal{A}}$, due to the property of timelike geodesics, i.e. the proper lengths between the past and future null infinities are the same for all timelike geodesics.

In this paper, we interpret the spacelike geodesic that represents the real part of the TEE as the IRT surface $\widehat{\mathcal{E}}$. When going back to Lorentzian signature, the IRT surface breaks in the middle and becomes disconnected. Later we will interpret the timelike geodesic that represents the imaginary part of the TEE as the boundary of the extended entanglement wedge, which asymptotically connects the two pieces of the IRT surface, see the discussions around (72). We find that the points on this timelike geodesic are not the fixed points of the modular momentum flow, which makes their role in the analog replica story [17] of the extended entanglement wedge unclear. We hope to investigate this point in the future. Nevertheless, we omit the imaginary part of the TEE temporarily.

# 3 Partial entanglement entropy, modular slice and entanglement wedge

## 3.1 Partial entanglement entropy

In this paper, we mainly focus on the two-dimensional system, and $x$ denotes the spatial coordinate. The *entanglement contour* $s_{\mathcal{A}}(x)$ [65] is a function for the region $\mathcal{A}$, which is conjectured to capture the contribution from each site $x$ inside $\mathcal{A}$ to the entanglement entropy $S_{\mathcal{A}}$. In other words, it is the density function of the entanglement entropy $S_{\mathcal{A}}$ which satisfies

$$S_{\mathcal{A}} = \int_{\mathcal{A}} s_{\mathcal{A}}(x)\,dx\,. \tag{41}$$

---

[11]In fact, we may define the TEE based on the modular momentum, which is similar to the method used to define entanglement entropy based on the modular Hamiltonian. More explicitly, the modular Hamiltonian and the modular momentum operators $H_{\text{mod}}, P_{\text{mod}}$ are derived by substituting the classical generators $L_i, \bar{L}_i$ in $k_t^{\alpha}, k_t^{\beta}$ with their quantum counterparts, which are the generators $\mathcal{L}_i, \bar{\mathcal{L}}_i$ of the Virasoro algebra (E.2)

$$H_{\text{mod}} = \frac{\pi l_U}{2}\mathcal{L}_{-1} - \frac{2\pi}{l_U}\mathcal{L}_1 - \frac{\pi l_V}{2}\bar{\mathcal{L}}_{-1} + \frac{2\pi}{l_V}\bar{\mathcal{L}}_1\,,$$
$$P_{\text{mod}} = \frac{\pi l_U}{2}\mathcal{L}_{-1} - \frac{2\pi}{l_U}\mathcal{L}_1 + \frac{\pi l_V}{2}\bar{\mathcal{L}}_{-1} - \frac{2\pi}{l_V}\bar{\mathcal{L}}_1\,. \tag{38}$$

The entanglement entropy $S_{\mathcal{A}}$ can be expressed as a function of $H_{\text{mod}}$ according to its definition, i.e. $H_{\text{mod}} = -\log\rho_{\mathcal{A}}$

$$S_{\mathcal{A}} = -\text{Tr}\left(\rho_{\mathcal{A}}\log\rho_{\mathcal{A}}\right) = \text{Tr}\left(H_{\text{mod}}e^{-H_{\text{mod}}}\right)\,. \tag{39}$$

Similarly, the TEE may be defined as,

$$S_{\widehat{\mathcal{A}}}^{(T)} \equiv \text{Tr}\left(P_{\text{mod}}e^{-P_{\text{mod}}}\right)\,. \tag{40}$$

The *partial entanglement entropy* (PEE) [59, 66–70] which describes the contribution from a subset $\mathcal{A}_i \subset \mathcal{A}$ to the entanglement entropy $S_{\mathcal{A}}$ can be obtained by the integration of the entanglement contour $s_{\mathcal{A}}(x)$ for the region $\mathcal{A}_i$

$$s_{\mathcal{A}}(\mathcal{A}_i) = \int_{\mathcal{A}_i} s_{\mathcal{A}}(x)\,dx\,. \tag{42}$$

The PEE $s_{\mathcal{A}}(\mathcal{A}_i)$ describes certain type of the correlation between the subset $\mathcal{A}_i$ and the region $\bar{A}$ that purifies $\mathcal{A}$, hence it is useful to rewrite the PEE as,

$$s_{\mathcal{A}}(\mathcal{A}_i) \equiv \mathcal{I}\left(\mathcal{A}_i, \bar{A}\right)\,. \tag{43}$$

The PEE should satisfy a set of requirements [65, 66] including all the requirements satisfied by the mutual information $I(A,B)$[12] and an additional key requirement of additivity according to its physical significance, see [51, 71–82] for recent progresses on PEE. Suppose $A, B, C$ are three non-overlapping regions on a Cauchy surface, we list the physical requirements satisfied by the PEE in the following:

1. Additivity: $\mathcal{I}(A, B \cup C) = \mathcal{I}(A,B) + \mathcal{I}(A,C)$.

2. Permutation symmetry: $\mathcal{I}(A,B) = \mathcal{I}(B,A)$.

3. Normalization:[13] $\mathcal{I}(A,B)|_{B \to \bar{A}} = S_A$.

4. Positivity: $\mathcal{I}(A,B) > 0$.

5. Upper bounded: $\mathcal{I}(A,B) \leq \min\{S_A, S_B\}$.

6. $\mathcal{I}(A,B)$ should be invariant under any local unitary transformations inside $A$ or $B$.

7. Symmetry: For any symmetry transformation $\mathcal{T}$ under which $\mathcal{T}A = A'$ and $\mathcal{T}B = B'$, we have $\mathcal{I}(A,B) = \mathcal{I}(A',B')$.

Up to now, there are many prescriptions to construct the PEE. We mainly focus on two of them in this paper, one is the additive linear combination (ALC) proposal [52, 59, 70] in two-dimensional system,[14] consider an interval $\mathcal{A}$ which is partitioned into three subintervals $\mathcal{A}_1 \cup \mathcal{A}_2 \cup \mathcal{A}_3$ where $\mathcal{A}_2$ is the middle one and $\mathcal{A}_1(\mathcal{A}_3)$ denotes the left (right) subinterval of $\mathcal{A}_2$, the proposal claims that

$$s_{\mathcal{A}}(\mathcal{A}_2) = \frac{1}{2}\left(S_{\mathcal{A}_1 \cup \mathcal{A}_2} + S_{\mathcal{A}_2 \cup \mathcal{A}_3} - S_{\mathcal{A}_1} - S_{\mathcal{A}_3}\right)\,. \tag{44}$$

The other one is the geometric construction applied to the holographic theories with a local modular Hamiltonian [52, 59], which claims there is an one-to-one correspondence between the points in the region $\mathcal{A}$ and the points on the RT surface $\mathcal{E}$, hence gives a natural contour function for $\mathcal{A}$. We will introduce this geometric construction in detail in the next subsection. The PEEs evaluated by these two proposals are highly consistent with each other [52, 59, 83].

---

[12]One should be careful not to confuse the PEE $\mathcal{I}(A,B)$ and the mutual information $I(A,B)$.

[13]Note that the normalization requirement is subtle as we need to match between two infinitely large quantities. In fact this requirement should only be imposed to special regions to make sure the existence of the solution for the whole set of requirements. For example, in the Poincaré AdS$_{d+1}$, the normalization requirement can only be imposed to spherical regions on the boundary [80, 82]

[14]In higher dimension, the ALC proposal only applies to the situation when all degrees of freedom settle in a unique order (for example a line or a circle).

## 3.2 Modular slice, entanglement wedge and holographic PEE

In the Rindler $\widetilde{\mathrm{AdS}}_3$, one can also consider the contour function for the thermal entropy which describes how much each boundary degrees of freedom contribute to the thermal entropy of the BTZ black string. Due to the translation symmetry along the $\tilde{x}$ direction, the contour function should be just a constant. On the other hand, the thermal entropy is calculated by the length of the outer horizon. This leads us to establish an effective partnership between the points on the boundary and the points on the horizon [59,67,68], and the points on the outer horizon represent the contribution from its boundary partner point to the thermal entropy. This partnership is not hard to determine as it should respect the translation symmetry along $\tilde{x}$. The motivation will be clearer if we consider a boundary sub-interval $\tilde{\mathcal{A}}_i$ and its partner points on the horizon, which forms a sub-interval $\tilde{\mathcal{E}}_i$ on the horizon. Hence, the contribution from $\tilde{\mathcal{A}}_i$ to the thermal entropy is just given by the length of $\tilde{\mathcal{E}}_i$.

Now we map the point-to-point partnership back to the original $\mathrm{AdS}_3$ and then get the partnership between the points on a boundary interval $\mathcal{A}$ and the points on the corresponding RT surface $\mathcal{E}$. Since the Rindler transformations are symmetries of the theory, the constant contour function of the thermal entropy is mapped to the entanglement contour of $\mathcal{A}$ in the original $\mathrm{AdS}_3$. Furthermore, for a subinterval $\mathcal{A}_i$ of $\mathcal{A}$, the PEE $s_{\mathcal{A}}(\mathcal{A}_i)$ can be captured by the length of a geodesic chord $\mathcal{E}_i$ on $\mathcal{E}$, which is formed by the partner points of $\mathcal{A}_i$. The PEE calculated in such a way exactly matches to the ALC proposal.

We will first give a brief review on how to explicitly build up the point-to-point partnership for a spacelike interval $\mathcal{A}$ and its RT surface $\mathcal{E}$ based on the slicing of the entanglement wedge using the so-called modular Hamiltonian slices [59,67,68]. Then we generalize this picture to build up a similar partnership between the points on a timelike interval $\widehat{\mathcal{A}}$ and the points on the inner RT surface, using the so-called modular momentum slices.

### 3.2.1 Modular Hamiltonian slice

The concept of modular (Hamiltonian) slice is proposed to provide a natural slicing of the entanglement wedge [59,67,68], which we call the fine structure of the entanglement wedge. This slicing is natural in the Rindler $\widetilde{\mathrm{AdS}}_3$ where there are translational symmetries along the Rindler coordinate $(\tilde{t}, \tilde{x})$, and modular slices are exactly the 2-dimensional spacetime slices with fixed $\tilde{x}$ coordinates, and the slicing respects the spatial translational symmetry. Also the modular slices in the Rindler $\widetilde{\mathrm{AdS}}_3$ can be mapped to certain spacetime slices in the entanglement wedge $\mathcal{W}_{\mathcal{A}}$, hence we obtain the slicing structure in the entanglement wedge.

We can also identify the modular slices in the entanglement wedge without referring to the Rindler $\widetilde{\mathrm{AdS}}_3$ by the following prescription:

- First, given an arbitrary point $P$ in the causal development $\mathcal{D}_{\mathcal{A}}$, we can identify a saddle geodesic $\gamma_P$ that connects $P$ to the RT surface $\mathcal{E}$ (or IRT surface $\widehat{\mathcal{E}}$) with the minimal length. Note that, the saddle geodesic $\gamma_P$ intersects with $\mathcal{E}$ vertically, and we call the intersection point $H$ on the RT surface the partner point of $P$. See the right figure of Fig.4.

- Second, we let the point $P$ move along the boundary modular flow line that passes through $P$, then the saddle geodesic $\gamma_P$ moves accordingly and sweeps out a 2-dimensional hypersurface $\mathcal{P}_H$ in the entanglement wedge, which is exactly the modular slice defined in [59].

The above construction maps the modular slices in the original space exactly to the modular slices with fixed $\tilde{x}$ coordinate in the Rindler $\widetilde{\mathrm{AdS}}_3$. Let us consider the boundary point

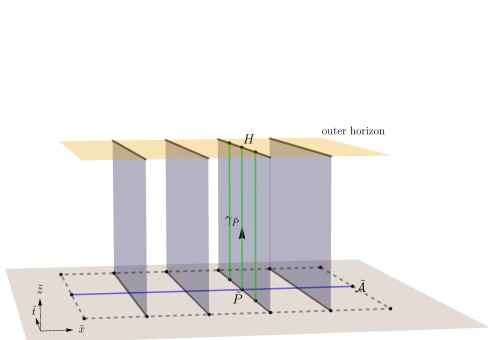
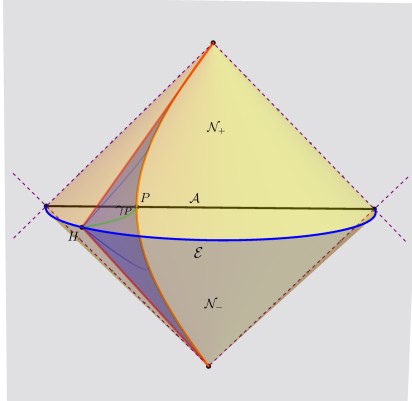

Figure 4: The left figure shows the modular Hamiltonian slices in the Rindler $\widetilde{\text{AdS}_3}$, i.e. the spacetime slices with fixed $\tilde{x}$ coordinates shaded in gray blue. The green lines represent the $\tilde{\rho}$ coordinate lines, with the one indicated by an arrow being the specific line that passes through the boundary point $\tilde{P}$, denoted as $\gamma_{\tilde{P}}$. The right figure shows the modular Hamiltonian slice that passes through $P$ in the original Poincaré $\text{AdS}_3$, and the orange curve represents the modular Hamiltonian flow $\mathcal{L}_H$. The red curves represent the intersection lines between the modular Hamiltonian slice and the null hypersurface $\mathcal{N}_\pm$.

$P = (U_1, V_1)$[15] and its image point $\tilde{P} = (\tilde{U}_1, \tilde{V}_1)$ on the Rindler boundary, one can find that the saddle geodesic $\gamma_P$ maps to the line $\gamma_{\tilde{P}}$ that emanates from $\tilde{P}$, moving along the $\tilde{\rho}$ direction and ending on the outer horizon $\tilde{\rho} = T_{\tilde{U}} T_{\tilde{V}}$ (see the green lines in the left figure of Fig.4), i.e.,

$$\gamma_{\tilde{P}}: \quad \tilde{t} = \tilde{t}_{\tilde{P}}, \qquad \tilde{x} = \tilde{x}_{\tilde{P}}, \qquad \tilde{\rho} \geq T_{\tilde{U}} T_{\tilde{V}}, \tag{45}$$

where

$$\tilde{t}_{\tilde{P}} = \frac{1}{2} \log\left( \frac{(l_U + 2U_1)(l_V - 2V_1)}{(l_U - 2U_1)(l_V + 2V_1)} \right), \qquad \tilde{x}_{\tilde{P}} = \frac{1}{2} \log\left( \frac{(l_U + 2U_1)(l_V + 2V_1)}{(l_U - 2U_1)(l_V - 2V_1)} \right), \tag{46}$$

represent the Rindler temporal and spatial coordinates of $\tilde{P}$ respectively. It is obvious that, $\gamma_{\tilde{P}}$ is the saddle geodesic connecting $\tilde{P}$ and the outer horizon $\tilde{\rho} = T_{\tilde{U}} T_{\tilde{V}}$.

We can obtain the pre-image $\gamma_P$ in the original Poincaré $\text{AdS}_3$ of $\gamma_{\tilde{P}}$ by using the mapping (23) and then solving the following equations:

$$\begin{aligned} \frac{1}{4} \log\left( \frac{((l_U + 2U)(l_V - 2V)\rho - 2)^2}{((l_U - 2U)(l_V + 2V)\rho - 2)^2} \right) &= \tilde{t}_{\tilde{P}}, \\ \frac{1}{4} \log\left( \frac{((l_U + 2U)(l_V + 2V)\rho + 2)^2}{((l_U - 2U)(l_V - 2V)\rho + 2)^2} \right) &= \tilde{x}_{\tilde{P}}. \end{aligned} \tag{47}$$

Note that, there are two solutions for these equations: one is a spacelike geodesic (48) that corresponds to the $\gamma_{\tilde{P}}$ (45) outside the horizon, and the other is a timelike geodesic (62) that corresponds to the extension of $\gamma_{\tilde{P}}$ inside the horizon. Then the saddle geodesic $\gamma_P$ is given by the spacelike solution

$$\gamma_P: \quad \rho(V) = \frac{2\widetilde{l_V}}{\widetilde{l_U}\left(\widetilde{l_V}^2 - 4(V - V_0)^2\right)}, \qquad U(V) = \frac{\widetilde{l_U}}{\widetilde{l_V}}(V - V_0) + U_0, \tag{48}$$

---

[15]From now on, if we omit the $\rho$ coordinate of a point, then this is a boundary point, i.e. $\rho = \infty$.

where the parameters $(\widetilde{l}_U, \widetilde{l}_V, V_0, U_0)$ read,

$$\widetilde{l}_U = \frac{-l_U^2 + 4U_1^2}{4U_1}, \quad \widetilde{l}_V = \frac{-l_V^2 + 4V_1^2}{4V_1}, \quad V_0 = \frac{l_V^2 + 4V_1^2}{8V_1}, \quad U_0 = \frac{l_U^2 + 4U_1^2}{8U_1}, \tag{49}$$

see the green line in the right figure of Fig.4. Compared with (31), it is easy to find that $\gamma_P$ is part of the RT surface for the interval with extension $\widetilde{l}_U$ and $\widetilde{l}_V$ and the center point $(U_0, V_0)$. One can further check that, $\gamma_P$ is normal to the RT surface $\mathcal{E}$, hence is the saddle geodesic. We can also identify the intersection point $H$ between $\gamma_P$ (48) and the RT surface $\mathcal{E}$, whose coordinate is given by

$$U_H = \frac{l_U}{l_V} V_H, \qquad V_H = \frac{l_V (l_V U_1 + l_U V_1)}{l_U l_V + 4U_1 V_1}. \tag{50}$$

Consider an arbitrary point $H = (U_H, V_H)$ on the RT surface $\mathcal{E}$. We can also obtain a boundary curve by fixing $V_H$ in (50), which we denote as $\mathcal{L}_H$.

$$\mathcal{L}_H: \ U(V) = \frac{l_U l_V (V_H - V)}{l_V^2 - 4V V_H}. \tag{51}$$

One can easily verify that all points on this curve have the same partner point $H = (U_H, V_H)$. More importantly, this curve is precisely the boundary modular Hamiltonian flow line of $k_t^\alpha$. The curve $\mathcal{L}_H$ is mapped to a boundary line $\tilde{\mathcal{L}}_H$ along the $\tilde{t}$ direction in the Rindler $\widetilde{\text{AdS}_3}$, see the black lines on the Rindler boundary in the left figure of Fig.4. Consider the boundary point $P = (U_1, V_1)$ that lies within this modular flow line $\mathcal{L}_H$, i.e., it satisfies the following equation,

$$U_1 = \frac{l_U l_V (V_H - V_1)}{l_V^2 - 4V_1 V_H}. \tag{52}$$

On the other hand, each point on $\mathcal{L}_H$ (51) determines a corresponding saddle geodesic, and the set of all these saddle geodesics forms the modular Hamiltonian slice $\mathcal{P}_H$, see the blue surface in the right figure of Fig.4,

$$\mathcal{P}_H: \ \rho = \frac{2\widetilde{l}_V}{\widetilde{l}_U \left(\widetilde{l}_V^2 - 4(V - V_0)^2\right)}, \qquad U = \frac{\widetilde{l}_U}{\widetilde{l}_V}(V - V_0) + U_0 \tag{53}$$

$$(V_H < V < V_1, -l_V/2 < V_1 < l_V/2),$$

where the parameters $(\widetilde{l}_U, \widetilde{l}_V, U_0, V_0)$ in the above expression are given by (49) and (52). Also we can check that, all the points on $\mathcal{P}_H$ maps to points on the modular slice with fixed $\tilde{x}$ coordinate in the Rindler $\widetilde{\text{AdS}_3}$,

$$\tilde{x} = \frac{1}{2} \log\left(\frac{l_V + 2V_H}{l_V - 2V_H}\right). \tag{54}$$

Note that, the saddle geodesics for different points on $\tilde{\mathcal{L}}_H$ intersect with the outer horizon at different points (see the green lines in the left figure of Fig.4), but all these intersection points are mapped to the only point $H$ on the RT surface $\mathcal{E}$ via the inverse Rindler transformations. In summary, we come up with the following conclusions:

- All points on the modular flow line $\mathcal{L}_H$ have the same partner point $H$ on $\mathcal{E}$.

- The points (e.g. $H$) on $\mathcal{E}$, the modular flow lines (e.g. $\mathcal{L}_H$) and the modular slices (e.g. $\mathcal{P}_H$) are in one-to-one correspondence.

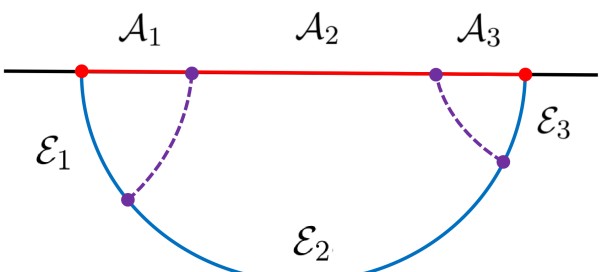

Figure 5: The figure is extracted from [75]. The PEEs $s_{\mathcal{A}}(\mathcal{A}_i)$ are captured by the length of $\mathcal{E}_i$. The purple lines are the saddle geodesics that are normal to the RT surface $\mathcal{E}$. Although the figure looks static, we should consider it to be a covariant configuration.

- The construction of modular slice gives a natural slicing for the entanglement wedge,

$$\mathcal{W}_{\mathcal{A}} = \left\{ \mathcal{P}_H \,\middle|\, -\frac{l_V}{2} < V_H < \frac{l_V}{2} \right\}. \tag{55}$$

One motivation to introduce the above fine structure of the entanglement wedge is to compute the PEE $s_{\mathcal{A}}(\mathcal{A}_i)$ [59, 67], where $\mathcal{A}_i$ is any subinterval of $\mathcal{A}$. Based on the partner relation between the points in $\mathcal{D}_{\mathcal{A}}$ and those on $\mathcal{E}$, we obtain the correspondence between any subinterval $\mathcal{A}_i$ of $\mathcal{A}$ and the geodesic chord $\mathcal{E}_i$ on $\mathcal{E}$. Here the points in $\mathcal{A}_i$ has partner points on $\mathcal{E}$, which are exactly the points that make the geodesic chord $\mathcal{E}_i$. It has been argued in [59, 67] that the PEE $s_{\mathcal{A}}(\mathcal{A}_i)$ is captured by the length of the partner subinterval $\mathcal{E}_i$ on $\mathcal{E}$, i.e.

$$s_{\mathcal{A}}(\mathcal{A}_i) = \frac{\text{Length}(\mathcal{E}_i)}{4G}, \tag{56}$$

see Fig.5 for an illustration. For later convenience, we define the length parameter $\lambda$ along the RT surface $\mathcal{E}$, and establish the relation between any boundary point, for example $P = (U_1, V_1)$, and the length parameter of its partner point $H$ (50) on $\mathcal{E}$,

$$\lambda(P) \equiv \tau_H = \frac{1}{2} \log\left( \frac{(l_U + 2U_1)(l_V + 2V_1)}{(l_U - 2U_1)(l_V - 2V_1)} \right), \tag{57}$$

where we have used (15). Then for any subinterval $\mathcal{A}_i$ with the following two endpoints,

$$\mathcal{A}_i : \ P = (U_1, V_1) \to Q = (U_2, V_2), \tag{58}$$

the PEE $s_{\mathcal{A}}(\mathcal{A}_i)$ is just given by

$$s_{\mathcal{A}}(\mathcal{A}_i) = \frac{\lambda(Q) - \lambda(P)}{4G} = \frac{1}{8G} \log\left( \frac{(l_U - 2U_1)(l_V - 2V_1)(l_U + 2U_2)(l_V + 2V_2)}{(l_U + 2U_1)(l_V + 2V_1)(l_U - 2U_2)(l_V - 2V_2)} \right). \tag{59}$$

This result aligns exactly with the result evaluated by the ALC proposal (44).

We point out that, the PEE $s_{\mathcal{A}}(\mathcal{A}_i)$ is invariant under the modular flow, which means that, any subinterval that intersects with the same class of modular flow lines or modular slices, has the same PEE [67]. More explicitly, if the endpoints of the subintervals $\mathcal{A}_i$ and $\mathcal{A}_i'$ are in the same pair of modular flow lines, then $s_{\mathcal{A}}(\mathcal{A}_i) = s_{\mathcal{A}}(\mathcal{A}_i')$, as their partner geodesic chords coincide. On the other hand, one can compute the PEE using the ALC proposal, and the requirement of modular invariance is sufficient to determine the boundary modular flow lines (see [67] for various examples).



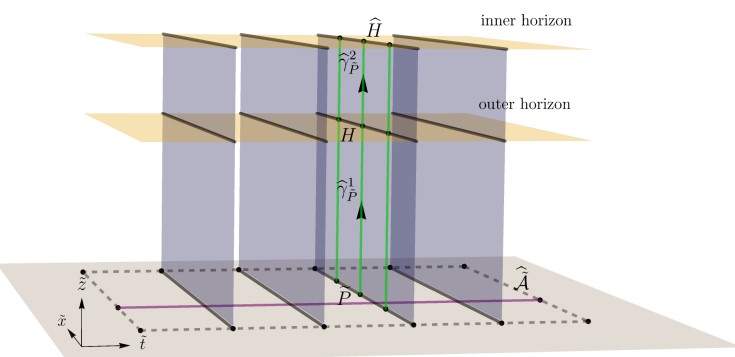

Figure 6: The gray blue surfaces are the spacetime slices with fixed $\tilde{t}$ coordinate, which are also the modular momentum slices. The green lines represent the $\tilde{\rho}$ coordinate lines, with the one indicated by an arrow being the specific line that passes through the boundary point $\tilde{P}$, denoted as $\widehat{\gamma}_{\tilde{P}}$, which is further divided into two parts $\widehat{\gamma}_{\tilde{P}}^1$ and $\widehat{\gamma}_{\tilde{P}}^2$. The purple line $\hat{\tilde{\mathcal{A}}}$ represents the partner interval of $\tilde{\mathcal{A}}$.

### 3.2.2 Modular momentum slice

Now we want to define a similar fine structure which is invariant under the modular momentum flow $k_t^\beta$. In this prescription, the coordinates $\tilde{t}$ and $\tilde{x}$ exchange their role, such that the $\tilde{x}$ direction becomes timelike, hence $\partial_{\tilde{x}}$ plays the role of the Hamiltonian in the region between the outer and the inner horizon. Also, the bulk modular momentum flow $k_t^{\beta,\text{bulk}}$ vanishes on the IRT surface $\widehat{\mathcal{E}}$. In this subsection, we follow a similar prescription to construct the modular momentum slices.

The bulk modular momentum flow represents the bulk replica symmetry for the timelike interval $\widehat{\mathcal{A}}$, whose image interval is a line along the $\tilde{t}$ direction in the Rindler $\widetilde{\text{AdS}}_3$. And in the Rindler $\widetilde{\text{AdS}}_3$, the modular momentum flow lines are the $\tilde{x}$ coordinate lines, hence the modular momentum slices are just the spacetime slices with fixed $\tilde{t}$ coordinate, see Fig.6. Note that, unlike the modular (Hamiltonian) slices, the modular momentum slice here is normal to the $\tilde{t}$ direction, rather than the $\tilde{x}$ direction. Now we start from an arbitrary point $\tilde{P}$, which is the image of a point $P$ in $\mathcal{D}_{\mathcal{A}}$, and search for a saddle curve $\widehat{\gamma}_{\tilde{P}}$ connecting $\tilde{P}$ and the inner horizon. This is again the line along the $\tilde{\rho}$ direction (see the green lines in Fig.6), i.e.,

$$\widehat{\gamma}_{\tilde{P}}: \quad \tilde{t} = \tilde{t}_P, \qquad \tilde{x} = \tilde{x}_P, \qquad \tilde{\rho} \geq -T_{\tilde{U}}T_{\tilde{V}}. \tag{60}$$

Since $\widehat{\gamma}_{\tilde{P}}$ transitions from spacelike to timelike when passing through the outer horizon, we divide it into the following two parts,

$$\begin{aligned} \widehat{\gamma}_{\tilde{P}}^1: \quad & \tilde{t} = \tilde{t}_P, \qquad \tilde{x} = \tilde{x}_P, \qquad \tilde{\rho} \geq T_{\tilde{U}}T_{\tilde{V}}, \\ \widehat{\gamma}_{\tilde{P}}^2: \quad & \tilde{t} = \tilde{t}_P, \qquad \tilde{x} = \tilde{x}_P, \qquad -T_{\tilde{U}}T_{\tilde{V}} \leq \tilde{\rho} \leq T_{\tilde{U}}T_{\tilde{V}}. \end{aligned} \tag{61}$$

When mapping back to the original Poincaré AdS$_3$, $\widehat{\gamma}_P^1$ is the spacelike solution of (47), which is precisely $\gamma_P$ as given by (48). And $\widehat{\gamma}_P^2$ is the timelike solution of (47), which is given by

$$\widehat{\gamma}_P^2: \quad \rho(V) = C_1 C_2 \left( -1 + \cos\left[ 2\text{arccot}\left( 2C_2(C_4 - V) \right) \right] \right), \quad U(V) = C_3 + \frac{C_2(V - C_4)}{C_1}, \tag{62}$$

where the parameters $(C_1, C_2, C_3, C_4)$ read,

$$C_1 = \frac{l_U^2 + 4U_1^2}{l_U(l_U^2 - 4U_1^2)}, \quad C_2 = -\frac{l_V^2 + 4V_1^2}{l_V(l_V^2 - 4V_1^2)}, \quad C_3 = \frac{2l_U^2 U_1}{l_U^2 + 4U_1^2}, \quad C_4 = \frac{2l_V^2 V_1}{l_V^2 + 4V_1^2}. \tag{63}$$

Since $\widehat{\gamma}_P$ intersects with both the outer and the inner horizons, $P$ has partner points on both the RT and the IRT surfaces. Again we denote the partner point on $\mathcal{E}$ as $H$ and the partner point on $\widehat{\mathcal{E}}$ as $\widehat{H}$. Given the coordinate of $P = (U_1, V_1)$, $H$ can again be specified by (50), while the partner $\widehat{H}$ is given by the intersection point between $\widehat{\gamma}_P^2$ and $\widehat{\mathcal{E}}$,[16] i.e.,

$$\widehat{H} = \widehat{\gamma}_P^2 \cap \widehat{\mathcal{E}}, \qquad V_{\widehat{H}} = -\frac{l_V(l_U l_V - 4U_1 V_1)}{4(l_V U_1 - l_U V_1)}. \tag{64}$$

One can see that, when $P$ is on the interval $\mathcal{A}$, its partner point on $\widehat{\mathcal{E}}$ goes to infinity, and when $P$ approaches one of the tips of $\mathcal{D}_{\mathcal{A}}$, its partner point also approaches the same tip. Similarly, for an arbitrary point $\widehat{H} = (U_{\widehat{H}}, V_{\widehat{H}})$ on $\widehat{\mathcal{E}}$, the corresponding modular momentum flow line can be also given by fixing $V_{\widehat{H}}$ in (64), which we denote as $\widehat{\mathcal{L}}_{\widehat{H}}$

$$\widehat{\mathcal{L}}_{\widehat{H}}: \ U(V) = \frac{l_U\left(l_V^2 - 4V V_{\widehat{H}}\right)}{4l_V\left(V - V_{\widehat{H}}\right)}. \tag{65}$$

Also, one can easily verify that all points on $\widehat{\mathcal{L}}_{\widehat{H}}$ have the same partner point $\widehat{H} = (U_{\widehat{H}}, V_{\widehat{H}})$ on $\widehat{\mathcal{E}}$. Note that, the points on $\widehat{\mathcal{L}}_{\widehat{H}}$ have different partner points on the RT surface $\mathcal{E}$. When we move $P = (U_1, V_1)$ along $\widehat{\mathcal{L}}_{\widehat{H}}$, which satisfies

$$U_1 = \frac{l_U(l_V^2 - 4V_1 V_{\widehat{H}})}{4l_V(V_1 - V_{\widehat{H}})}, \tag{66}$$

its saddle geodesics $\widehat{\gamma}_P^1$ and $\widehat{\gamma}_P^2$ sweep out two spacetime slices, $\widehat{\mathcal{P}}_{\widehat{H}}^1$ and $\widehat{\mathcal{P}}_{\widehat{H}}^2$, respectively. See the two blue surfaces in the left and the right figure of Fig.7. Then the modular momentum slice $\widehat{\mathcal{P}}_{\widehat{H}}$ associated with $\widehat{\mathcal{L}}_{\widehat{H}}$ is defined as the union of these two slices

$$\begin{aligned}
\widehat{\mathcal{P}}_{\widehat{H}}^1: \ & \rho = \frac{2\widetilde{l}_V}{\widetilde{l}_U\left(\widetilde{l}_V^2 - 4(V - V_0)^2\right)}, \qquad U = \frac{\widetilde{l}_U}{\widetilde{l}_V}(V - V_0) + U_0 \\
& (V_1 < V < V_H, -l_V/2 < V_1 < l_V/2), \\
\widehat{\mathcal{P}}_{\widehat{H}}^2: \ & \rho = C_1 C_2\left(-1 + \cos\left[2\operatorname{arccot}\left(2C_2\left(C_4 - V\right)\right)\right]\right), \qquad U = C_3 + \frac{C_2\left(V - C_4\right)}{C_1} \\
& \left(V_{\widehat{H}} < V < V_H, -l_V/2 < V_1 < l_V/2\right).
\end{aligned} \tag{67}$$

The parameters $(C_1, C_2, C_3, C_4, \widetilde{l}_U, \widetilde{l}_V, U_0, V_0)$ in the above expression are given by (49), (63) and (66). As expected, $\widehat{\mathcal{P}}_{\widehat{H}}$ maps to a fixed $\widetilde{t}$ slice in the Rindler $\widetilde{\mathrm{AdS}}_3$

$$\widetilde{t} = \frac{1}{2}\log\left(\frac{2V_{\widehat{H}} - l_V}{2V_{\widehat{H}} + l_V}\right). \tag{68}$$

Again the modular momentum flow lines (e.g. $\widehat{\mathcal{L}}_{\widehat{H}}$), the points (e.g. $\widehat{H}$) on the IRT surface $\widehat{\mathcal{E}}$ and the modular momentum slices (e.g. $\widehat{\mathcal{P}}_{\widehat{H}}$) are in one-to-one correspondence.

As we have shown that the entanglement wedge $\mathcal{W}_{\mathcal{A}}$ is the union of all the modular Hamiltonian slices, and covers the region outside the outer horizon in the Rindler $\widetilde{\mathrm{AdS}}_3$. Now we define the extended entanglement wedge $\widehat{\mathcal{W}}_{\mathcal{A}}$, which is the union of all the modular momentum slices and is mapped to the region outside the inner horizon in the Rindler $\widetilde{\mathrm{AdS}}_3$ correspondingly, i.e.,

$$\widehat{\mathcal{W}}_{\mathcal{A}} = \left\{\widehat{\mathcal{P}}_{\widehat{H}}\,\middle|\,|V_{\widehat{H}}| > \frac{l_V}{2}\right\}. \tag{69}$$

---

[16]In appendix D, we also present a trick for locating the partner point $\widehat{H}$ on $\widehat{\mathcal{E}}$ using the null hypersurface.

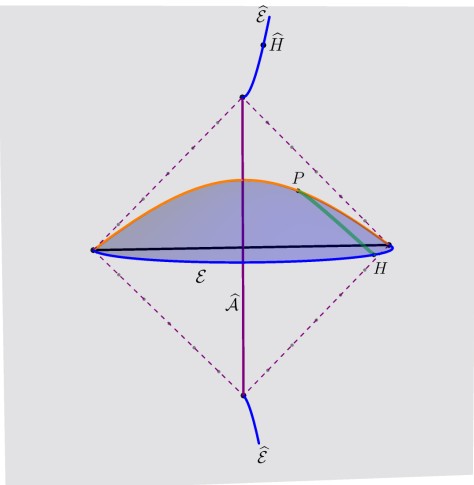 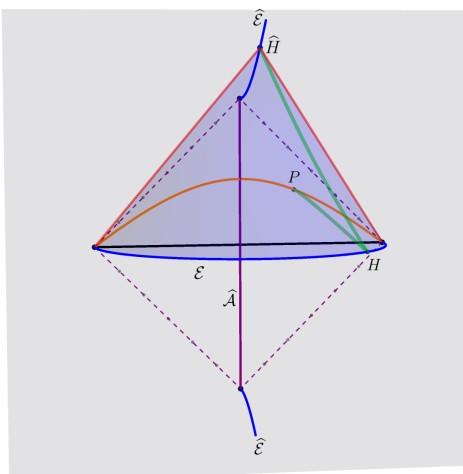

Figure 7: The orange curve is the boundary modular momentum flow line $\widehat{\mathcal{L}}_{\widehat{H}}$. $P$ is a boundary point on $\widehat{\mathcal{L}}_{\widehat{H}}$, with its partner points on $\mathcal{E}$ and $\widehat{\mathcal{E}}$ denoted by $H$ and $\widehat{H}$, respectively. The blue surfaces in the left figure and that in the right figure represent the hypersurfaces $\widehat{\mathcal{P}}^1_{\widehat{H}}$ and $\widehat{\mathcal{P}}^2_{\widehat{H}}$, respectively, and their union forms the modular momentum slice $\widehat{\mathcal{P}}_{\widehat{H}}$. The green curve in the left figure is the spacelike saddle geodesic $\widehat{\gamma}^1_P$. The two green curves in the right figure are the spacelike saddle geodesic $\widehat{\gamma}^1_P$ and the timelike saddle geodesic $\widehat{\gamma}^2_P$ respectively. The red lines are the intersection lines between $\widehat{\mathcal{P}}_{\widehat{H}}$ and $\mathcal{M}_{\pm}$.

The extended entanglement wedge $\widehat{\mathcal{W}}_{\mathcal{A}}$ contains the entanglement wedge $\mathcal{W}_{\mathcal{A}}$ as a subregion. As $|V_{\widehat{H}}|$ increases, the modular momentum slice goes deeper into the bulk. It is also interesting to note that, when $V_{\widehat{H}} \to \pm\infty$, the union of these two corresponding modular momentum slice $\widehat{\mathcal{P}}^2_{\widehat{H}}$ makes a smooth timelike hypersurface, which is given by,

$$
\widehat{\mathcal{P}}^2_{\infty} \equiv \lim_{V_{\widehat{H}} \to \infty} \widehat{\mathcal{P}}^2_{\widehat{H}} \cup \lim_{V_{\widehat{H}} \to -\infty} \widehat{\mathcal{P}}^2_{\widehat{H}}
$$
$$
= \left\{ \rho = \frac{2l_V \left(l_V^2 + 4V_1^2\right)}{l_U \left(l_V^4 + 4l_V^2 \left(V^2 - 4VV_1 + V_1^2\right) + 16V^2V_1^2\right)}, \quad U = l_U \left(\frac{4l_V V_1}{l_V^2 + 4V_1^2} - \frac{V}{l_V}\right) \right\} \tag{70}
$$
$$
(-\infty < V < \infty, -l_V/2 < V_1 < l_V/2) \,.
$$

See the blue surface in the left figure of Fig.8. Therefore, the extended entanglement wedge $\widehat{\mathcal{W}}_{\mathcal{A}}$ is the region bounded by the null hypersurfaces $\mathcal{M}_{\pm}$, the causal development $\mathcal{D}_{\mathcal{A}}$ and the timelike hypersurface $\widehat{\mathcal{P}}^2_{\infty}$. According to (65), in the limit $\widehat{H} \to \infty$, the modular momentum flow line $\widehat{L}_{\widehat{H}}$ becomes $\widehat{L}_{\infty}$, which is precisely the boundary straight line overlapping with the interval $\mathcal{A}$. In other words, the timelike hypersurface $\widehat{\mathcal{P}}^2_{\infty}$ is the modular momentum slice associated with the modular momentum flow line $\widehat{L}_{\infty}$ that overlaps with the interval $\mathcal{A}$. For convenience, we define two hypersurfaces $\Xi_{\mathcal{A}}$ and $\Xi_{\widehat{\mathcal{A}}}$ as the spacetime slices where $\mathcal{A}$ and $\widehat{\mathcal{A}}$ are located, respectively,

$$
\Xi_{\mathcal{A}} : \ Ul_U - Vl_V = 0 \,, \qquad \Xi_{\widehat{\mathcal{A}}} : \ Ul_U + Vl_V = 0 \,. \tag{71}
$$

It is well known that the RT surface $\mathcal{E}$ is a boundary of the intersection surface between the entanglement wedge and the hypersurface $\Xi_{\mathcal{A}}$. Similarly, we can also study the intersection surface between the extended entanglement wedge $\widehat{\mathcal{W}}_{\mathcal{A}}$ and the hypersurface $\Xi_{\widehat{\mathcal{A}}}$. The inter-

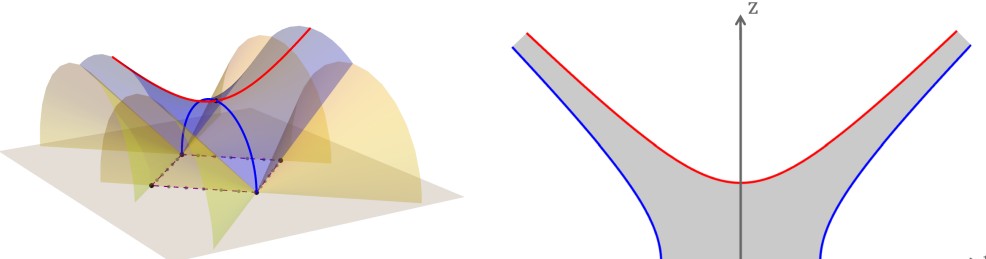

Figure 8: In the left figure, the blue surface is the modular momentum slice $\widehat{\mathcal{P}}_\infty^2$. The extended entanglement wedge $\widehat{\mathcal{W}}_\mathcal{A}$ is the bulk region bounded by $\mathcal{M}_\pm$ (the yellow surfaces), $\mathcal{D}_\mathcal{A}$ and $\widehat{\mathcal{P}}_\infty^2$. The red curves in the left and the right figures represent the intersection line between $\widehat{\mathcal{P}}_\infty^2$ and $\Xi_{\widehat{\mathcal{A}}}$, which we denote as $\widehat{\gamma}_\infty^2$. In the right figure, the blue curve represents the IRT surface $\widehat{\mathcal{E}}$. And the shadow region is the intersection surface between $\widehat{\mathcal{W}}_\mathcal{A}$ and $\Xi_{\widehat{\mathcal{A}}}$.

section line between $\widehat{\mathcal{P}}_\infty^2$ and $\Xi_{\widehat{\mathcal{A}}}$ is denoted as $\widehat{\gamma}_\infty^2$, which is a timelike geodesic,

$$\widehat{\gamma}_\infty^2 : \rho = \frac{2l_V}{l_U\left(l_V^2 + 4V^2\right)}, \qquad U = -\frac{l_U}{l_V}V, \tag{72}$$

see the red line in Fig.8. Therefore, the intersection surface between the extended entanglement wedge $\widehat{\mathcal{W}}_\mathcal{A}$ and the spacetime slice $\Xi_{\widehat{\mathcal{A}}}$ is a codimension one region bounded by the timelike interval $\widehat{\mathcal{A}}$, the IRT surface $\widehat{\mathcal{E}}$ and the timelike geodesic $\widehat{\gamma}_\infty^2$, see the gray region in the right figure of Fig.8. Hence the timelike geodesic $\widehat{\gamma}_\infty^2$ gives a natural geometric interpretation for the imaginary part of the TEE $S_\mathcal{A}^{(T)}$ (35), and connects the two parts of the IRT surface $\widehat{\mathcal{E}}$ at the past and the future null infinities. As expected, the absolute value of the length of $\widehat{\gamma}_\infty^2$ is $\pi$, i.e.,

$$\ell_{\widehat{\gamma}_\infty^2} = \int_{-\infty}^{\infty} \frac{2l_V}{l_V^2 + 4V^2}dV = \pi. \tag{73}$$

In particular, if we consider the case when $\mathcal{A}$ is a static spacelike interval of length $L$, i.e. $l_U = l_V = L/2$. Under the $(t, x, z)$ coordinates (5), the corresponding timelike geodesic $\widehat{\gamma}_\infty^2$ (72) is given by

$$\widehat{\gamma}_\infty^2 : t^2 - z^2 = -\frac{L^2}{4}, \qquad x = 0. \tag{74}$$

One can easily verify that the asymptotic behavior (when $t \to \pm\infty$) at the null infinity of $\widehat{\gamma}_\infty^2$ is the same as that of the IRT surface $\widehat{\mathcal{E}}$ (34) at the first order derivative level (i.e. the leading order),

$$\widehat{\gamma}_\infty^2 : z \sim t, \qquad \widehat{\mathcal{E}} : z \sim t, \tag{75}$$

which explains why the requirement that the first order derivatives of the timelike geodesic and the spacelike geodesic should be equal at the merging point, which is proposed in [40,41], is reasonable. More interestingly, when we embed the Poincaré AdS$_3$ into the global AdS$_3$, the IRT surface can be extended to the asymptotic boundary, with the anchor points being the two tips of the causal development of the complement $\mathcal{B} = \bar{\mathcal{A}}$. Thus, the extended part can be referred to as the IRT surface of $\mathcal{B}$. In this situation, the IRT surface of $\mathcal{A}$ and that of its complement $\mathcal{B}$ are not the same one. See Fig.14 and appendix B for the details. Also note that, the timelike geodesic $\widehat{\gamma}_\infty^2$ is not the set of fixed points of the modular momentum flow $k_t^{\beta,\text{bulk}}$, hence does not plays the similar role as the IRT surface $\widehat{\mathcal{E}}$ in the bulk replica story.

Similarly, we can also define the length parameter $\widehat{\lambda}$ on $\widehat{\mathcal{E}}$ and establish the function $\widehat{\lambda}(P)$, between any boundary point $P = (U_1, V_1)$ and the length parameter of its partner point $\widehat{H}$ (64) on $\widehat{\mathcal{E}}$, i.e.,

$$\widehat{\lambda}(P) \equiv \widehat{\tau}_{\widehat{H}} = \frac{1}{2} \log\left(\frac{(l_U + 2U_1)(l_V - 2V_1)}{(l_U - 2U_1)(l_V + 2V_1)}\right), \tag{76}$$

where we have used (32). As any boundary point in $\mathcal{D}_{\mathcal{A}}$ has a partner point on $\widehat{\mathcal{E}}$ according to (64), in the same sense any subinterval $\widehat{\mathcal{A}}_i$ of the timelike interval $\widehat{\mathcal{A}}$ (33) corresponds to a geodesic chord $\widehat{\mathcal{E}}_i$ on the IRT surface $\widehat{\mathcal{E}}$. Similar to (56), the timelike partial entanglement entropy (TPEE) $s_{\widehat{\mathcal{A}}}^{(T)}(\widehat{\mathcal{A}}_i)$ that captures the contribution from the subinterval $\widehat{\mathcal{A}}_i$ (58) to the TEE $S_{\widehat{\mathcal{A}}}^{(T)}$ is also given by the length of its partner geodesic chord $\widehat{\mathcal{E}}_i$. More explicitly, consider the subinterval $\widehat{\mathcal{A}}_i$ with the following two endpoints,

$$\widehat{\mathcal{A}}_i : \ P = (U_1, V_1) \to Q = (U_2, V_2), \tag{77}$$

we should have,

$$\begin{aligned}
s_{\widehat{\mathcal{A}}}^{(T)}(\widehat{\mathcal{A}}_i) &= \frac{\text{Length}(\widehat{\mathcal{E}}_i)}{4G} = \frac{\widehat{\lambda}(Q) - \widehat{\lambda}(P)}{4G} \\
&= \frac{1}{8G} \log\left(\frac{(l_U - 2U_1)(l_V + 2V_1)(l_U + 2U_2)(l_V - 2V_2)}{(l_U + 2U_1)(l_V - 2V_1)(l_U - 2U_2)(l_V + 2V_2)}\right).
\end{aligned} \tag{78}$$

We can also use the TPEE $s_{\widehat{\mathcal{A}}}^{(T)}(\widehat{\mathcal{A}}^{\text{reg}})$, where $\widehat{\mathcal{A}}^{\text{reg}}$ is the following regularized timelike interval of $\widehat{\mathcal{A}}$, to regulate the timelike entanglement entropy $S_{\widehat{\mathcal{A}}}^{(T)}$,

$$\widehat{\mathcal{A}}^{\text{reg}} : \ (-l_U/2 + \epsilon_U, l_V/2 - \epsilon_V) \to (l_U/2 - \epsilon_U, -l_V/2 + \epsilon_V). \tag{79}$$

The partner points of the left and the right endpoints of $\widehat{\mathcal{A}}^{\text{reg}}$ on $\widehat{\mathcal{E}}$ are denoted by $\widehat{H}_1$ and $\widehat{H}_2$, respectively. These points are given by

$$V_{\widehat{H}_1} = \frac{l_V}{2} + \frac{\epsilon_U \epsilon_V}{l_U}, \quad z_{\widehat{H}_1} = 2\sqrt{\epsilon_U \epsilon_V}, \qquad V_{\widehat{H}_2} = -\frac{l_V}{2} - \frac{\epsilon_U \epsilon_V}{l_U}, \quad z_{\widehat{H}_2} = 2\sqrt{\epsilon_U \epsilon_V}, \tag{80}$$

where we ignore the higher order terms $\mathcal{O}(\epsilon^3)$. One can easily find that $\widehat{H}_1$ approaches to the lower tip of $\mathcal{D}_{\mathcal{A}}$, while $\widehat{H}_2$ approaches to the upper tip of $\mathcal{D}_{\mathcal{A}}$. Therefore, the timelike entanglement entropy $S_{\widehat{\mathcal{A}}}^{(T)}$ is given by

$$S_{\widehat{\mathcal{A}}}^{(T)} = s_{\widehat{\mathcal{A}}}^{(T)}(\widehat{\mathcal{A}}^{\text{reg}}) = \frac{\text{Length}(\widehat{\mathcal{E}}^{\text{reg}})}{4G} = \frac{1}{4G} \log\left(\frac{l_U l_V}{\epsilon_U \epsilon_V}\right), \tag{81}$$

where $\widehat{\mathcal{E}}^{\text{reg}}$ is the partner geodesic chord of $\widehat{\mathcal{A}}^{\text{reg}}$ on $\widehat{\mathcal{E}}$, and we ignore the higher order terms $\mathcal{O}(\epsilon)$.

Before ending up this section, we also point out that, one can also use the null geodesics on $\mathcal{M}_{\pm}$ to identify the one-to-one correspondence between the boundary modular momentum flow line $\widehat{\mathcal{L}}_{\widehat{H}}$, the point $\widehat{H}$ on $\widehat{\mathcal{E}}$ and the modular slice $\widehat{\mathcal{P}}_{\widehat{H}}$ (67). This prescription looks more similar to the previous work [52, 59]. Let us denote the bulk modular flow lines on the null hypersurfaces $\mathcal{M}_{\pm}$ (30) as $\widehat{\mathcal{L}}_{\widehat{H}}^{\pm}$. Hence, $\widehat{H}, \widehat{\mathcal{L}}_{\widehat{H}}$ and $\widehat{\mathcal{L}}_{\widehat{H}}^{\pm}$ are in one-to-one correspondence,

$$\widehat{H} \leftrightarrow \widehat{\mathcal{L}}_{\widehat{H}} \leftrightarrow \widehat{\mathcal{L}}_{\widehat{H}}^{\pm}, \tag{82}$$

where $\widehat{\mathcal{L}}^{\pm}_{\widehat{H}}$ are also the null geodesics since $\mathcal{M}_{\pm}$ are the Killing horizons of $k_t^{\beta,\text{bulk}}$. More explicitly, $\widehat{\mathcal{L}}^{\pm}_{\widehat{H}}$ are the intersection lines between $\widehat{\mathcal{P}}_{\widehat{H}}$ and $\mathcal{M}_{\pm}$, i.e. the null geodesics connecting the endpoints of $\mathcal{A}$ with $\widehat{H}$,

$$\widehat{\mathcal{L}}^{\pm}_{\widehat{H}}: \ \rho(V) = -\frac{2l_V\left(l_V \mp 2V_{\widehat{H}}\right)}{l_U\left(l_V \mp 2V\right)^2\left(l_V \pm 2V_{\widehat{H}}\right)}, \quad U(V) = \frac{l_U\left(l_V\left(V - 2V_{\widehat{H}}\right) \pm 2V V_{\widehat{H}}\right)}{l_V\left(l_V \mp 2V_{\widehat{H}}\right)}, \quad (83)$$

see the red lines in Fig.7. In other words, $\widehat{H}, \widehat{\mathcal{L}}_{\widehat{H}}$ and $\widehat{\mathcal{L}}^{\pm}_{\widehat{H}}$ are respectively the intersections of $\widehat{\mathcal{P}}_{\widehat{H}}$ with the IRT surface $\widehat{\mathcal{E}}$, the asymptotic boundary and the null hypersurfaces $\mathcal{M}_{\pm}$.

# 4 Topological massive gravity and holographic entanglement entropy with gravitational anomaly

## 4.1 CFT$_2$ with gravitational anomaly and topological massive gravity

For a quantum field theory with gravitational anomalies (not necessarily two-dimensional) which means there is a breakdown of energy-momentum conservation at the quantum level, it cannot be coupled to a dynamical gravitational background [84]. Holographically, the action in the bulk not only includes the Einstein-Hilbert action (with a cosmological constant), but also includes a gravitational Chern-Simons term [28], which is called the topological massive gravity (TMG). From now on, the quantum field theory we study is the two-dimensional CFT with gravitational anomalies in the vacuum state on the plane, which duals to the Poincaré AdS$_3$ in TMG. We will denote the CFT$_2$ with gravitational anomaly as CFT$_2^a$. Such theories are described by two copies of the Virasoro algebra with two unequal central charges $c_L$ and $c_R$. Consider the same boundary spatial interval $\mathcal{A}$ as described in (3). The entanglement entropy $S_{\mathcal{A}}$ for the interval $A$ can be obtained using the replica method [50], as we reviewed in appendix E,

$$\begin{aligned} S_{\mathcal{A}} &= \frac{c_L}{6}\log\left(\frac{l_U}{\epsilon_U}\right) + \frac{c_R}{6}\log\left(\frac{l_V}{\epsilon_V}\right) \\ &= \frac{c_L + c_R}{12}\log\left(\frac{l_U l_V}{\epsilon_U \epsilon_V}\right) + \frac{c_L - c_R}{12}\log\left(\frac{l_U \epsilon_V}{l_V \epsilon_U}\right), \end{aligned} \quad (84)$$

where $\epsilon_U$ and $\epsilon_V$ are the cutoffs along the $U$ direction and $V$ direction, respectively.

The asymptotic symmetry algebra of Einstein gravity has two equal central charges. In order to incorporate the effect of gravitational anomaly, we should consider the TMG [27,85] whose action includes an additional Chern-Simons (CS) term, namely

$$I_{\text{TMG}} = \frac{1}{16\pi G}\int d^3x\sqrt{-g}\left(R - 2\Lambda + \frac{1}{2\mu}\varepsilon^{\alpha\beta\gamma}\left(\Gamma^{\rho}_{\ \alpha\sigma}\partial_{\beta}\Gamma^{\sigma}_{\ \gamma\rho} + \frac{2}{3}\Gamma^{\rho}_{\ \alpha\sigma}\Gamma^{\sigma}_{\ \beta\eta}\Gamma^{\eta}_{\ \gamma\rho}\right)\right), \quad (85)$$

where $\Lambda = -1$ is the cosmological constant with a AdS$_3$ radius $\ell = 1$ and $\mu$ is the coupling constant that characterizes the interaction strength between the CS term and the Einstein-Hilbert term. We denote this correspondence as TMG$_3$/CFT$_2^a$ correspondence. The Einstein gravity can be recovered in the limit $\mu \to \infty$. Compared with the Einstein gravity, the left and the right central charges of the boundary CFT$_2$ of the TMG are not equal any more [86],

$$c_L = \frac{3}{2G}\left(1 + \frac{1}{\mu}\right), \qquad c_R = \frac{3}{2G}\left(1 - \frac{1}{\mu}\right), \quad (86)$$

and the equation of motion is influenced by the CS term [50],

$$R_{\mu\nu} - \frac{1}{2}(R + 2)g_{\mu\nu} = -\frac{1}{\mu}C_{\mu\nu}, \quad (87)$$

where $C_{\mu\nu}$ is the Cotton tensor. Therefore, the TMG also admits the locally AdS$_3$ solutions in the case that the Cotton tensor is vanishing. For example, the BTZ black hole solutions,

$$ds^2 = T_U^2 dU^2 + 2\rho \, dU dV + T_V^2 dV^2 + \frac{d\rho^2}{4\left(\rho^2 - T_U^2 T_V^2\right)}, \tag{88}$$

which can be written in the ADM form,

$$
\begin{aligned}
ds^2 &= -\frac{\left(r^2 - r_-^2\right)\left(r^2 - r_+^2\right)}{r^2} dt^2 + \frac{r^2}{\left(r^2 - r_-^2\right)\left(r^2 - r_+^2\right)} dr^2 + r^2\left(d\varphi + \frac{r_+ r_-}{r^2} dt\right)^2 \\
&= -\left(r^2 + \frac{J^2}{4r^2} - M\right) dt^2 + \frac{1}{\left(r^2 + \frac{J^2}{4r^2} - M\right)} dr^2 + r^2\left(d\varphi + \frac{J}{2r^2} dt\right)^2,
\end{aligned}
\tag{89}
$$

by the following coordinate transformations,

$$
\begin{aligned}
U &= \frac{\varphi + t}{2}, \qquad V = \frac{\varphi - t}{2}, \qquad \rho = 2r^2 - r_-^2 - r_+^2, \\
T_U &= r_+ + r_-, \quad T_V = r_+ - r_-,
\end{aligned}
\tag{90}
$$

and $M, J$ are the physical conserved charges in the Einstein gravity,

$$M = r_+^2 + r_-^2, \qquad J = 2r_+ r_-. \tag{91}$$

However, due to the presence of the CS term, the physical conserved charges in the TMG get shifted [86, 87],

$$\mathcal{M} = M + \frac{J}{\mu}, \qquad \mathcal{J} = J + \frac{M}{\mu}. \tag{92}$$

The black hole entropy in such theories is derived in [31] based on the Wald formula [88–90], and the result shows that the correction of the thermal entropy for the outer horizon from the CS term is proportional to the area (or the length) of the inner horizon [28–31],

$$S_{\text{outer}} = \frac{\ell_{\text{outer horizon}}}{4G} + \frac{\ell_{\text{inner horizon}}}{4G\mu}, \tag{93}$$

where the length of the horizons comes from the integration for their induced metrics. As we reviewed in Sec.2.1, the entanglement entropy of the interval $\mathcal{A}$ (3) is mapped to the thermal entropy of the outer horizon of Rindler $\widetilde{\text{AdS}}_3$ (9). The length of the inner horizon of the Rindler $\widetilde{\text{AdS}}_3$ (9) is given by

$$\ell_{\text{inner horizon}} = T_{\tilde{U}} \Delta \tilde{U} - T_{\tilde{V}} \Delta \tilde{V} = \log\left(\frac{l_U \epsilon_V}{l_V \epsilon_U}\right), \tag{94}$$

where $\Delta \tilde{U}$ and $\Delta \tilde{V}$ are the lengths of $\tilde{\mathcal{A}}^{\text{reg}}$ along the $U$ direction and the $V$ direction, as described in (16). Substituting (18), (94) into (93), we obtain the (regulated) holographic entanglement entropy with gravitational anomaly,

$$S_{\mathcal{A}} = S_{\mathcal{A}}^{\text{n}} + S_{\mathcal{A}}^{\text{a}} = \frac{1}{4G} \log\left(\frac{l_U l_V}{\epsilon_U \epsilon_V}\right) + \frac{1}{4\mu G} \log\left(\frac{l_U \epsilon_V}{l_V \epsilon_U}\right), \tag{95}$$

which is exactly the same as the result (84) evaluated by the replica method with the central charges (86). Here, the first term $S_{\mathcal{A}}^{\text{n}}$ represents the contribution from the Einstein-Hilbert action, while the second term $S_{\mathcal{A}}^{\text{a}}$ represents the correction from the CS term, which we refer to as the normal part and the anomalous part of the entanglement entropy respectively. In [50], the authors set the cutoffs to be $\epsilon_U = \epsilon_V = \epsilon$, then the entanglement entropy $S_{\mathcal{A}}$ is given by,

$$S_{\mathcal{A}} = S_{\mathcal{A}}^{\text{n}} + S_{\mathcal{A}}^{\text{a}} = \frac{1}{4G} \log\left(\frac{l_U l_V}{\epsilon^2}\right) + \frac{1}{4\mu G} \log\left(\frac{l_U}{l_V}\right). \tag{96}$$

## 4.2 Bulk description with gravitational anomaly

In [50], the authors proposed a bulk calculation for the holographic entanglement entropy with gravitational anomaly by generalizing the LM prescription [17], which we gave a brief introduction in section 2.2. Following the calculations in [50], we apply the LM prescription for the spacelike interval $\mathcal{A}$, with the difference that the bulk gravitational theory is the TMG (85) instead of the Einstein gravity. We calculate the partition function $Z_n$ on the replica manifold $\mathcal{M}_n$ and compute the holographic entanglement entropy following (36), and we will find that the Einstein-Hilbert term in the action (85) gives the contribution

$$S_{\text{cone,Einstein}} = -\frac{\epsilon}{4G} \int_{\mathcal{C}} d\lambda \sqrt{g_{\mu\nu}(X) \dot{X}^\mu \dot{X}^\nu} + \mathcal{O}(\epsilon^2), \tag{97}$$

which looks like the familiar RT formula. Here $\epsilon = n - 1$ is a infinitesimal parameter when $n \to 1$, and we use $\lambda$ to parameterize the geodesic. On the other hand, after some reformulation the contribution from the Chern-Simons part of the action (4) can be written as (see [50] for details),

$$S_{\text{cone,CS}} = -\frac{\epsilon}{16\mu G} \int_{\mathcal{C}} d\lambda \frac{1}{\mu} \tilde{\mathbf{n}} \cdot \nabla \mathbf{n}, \tag{98}$$

where $\nabla$ is the covariant derivative alone the worldline $\mathcal{C}$. Here we have introduced a normal frame, where $\tilde{\mathbf{n}}$ and $\mathbf{n}$ are unit spacelike and timelike vectors respectively which are orthogonal to each other and also orthogonal to the unit tangent vector $\mathbf{v}$ along $\mathcal{C}$, i.e.

$$\tilde{\mathbf{n}}^2 = 1, \qquad \mathbf{n}^2 = -1, \qquad \mathbf{n} \cdot \tilde{\mathbf{n}} = 0, \qquad \tilde{\mathbf{n}} \cdot \mathbf{v} = 0, \qquad \mathbf{n} \cdot \mathbf{v} = 0. \tag{99}$$

One can take the worldline with a normal frame as the worldline of a spinning particle, where the information of the normal frame is represented by the direction of the spin vector. To conclude, the holographic entanglement entropy (36) with gravitational anomaly can be obtained by extremizing a worldline action for a spinning particle in the AdS$_3$ bulk, which we refer to as the *twist description* in this paper,

$$S_{\text{HEE}} = \frac{1}{4G} \int_{\mathcal{C}} d\lambda \left( \sqrt{g_{\mu\nu} \dot{X}^\mu \dot{X}^\nu} + \frac{1}{\mu} \tilde{\mathbf{n}} \cdot \nabla \mathbf{n} \right). \tag{100}$$

Finally, the worldline $\mathcal{C}$ will be determined by the extremization of the worldline action.

In this paper, we only focus on the locally AdS$_3$ spacetime, where the worldline $\mathcal{C}$ extremizing the worldline action is precisely a extremal surface,[17] hence the first term in (100) represents the length of the RT surface. The second term presents the correction from the CS term which we denote as $S_{CS}$. Interestingly, the integrand in $S_{CS}$ can be rewritten as a total derivative [50],

$$\tilde{\mathbf{n}} \cdot \nabla \mathbf{n} = \partial_\tau \log((\mathbf{q} - \tilde{\mathbf{q}}) \cdot \mathbf{n}), \tag{101}$$

hence the value of $S_{CS}$ only depends on the normal vectors at the boundary points of $\mathcal{C}$,

$$S_{CS} = \frac{1}{4\mu G} \log \left( \frac{\mathbf{q}(\tau_f) \cdot \mathbf{n}_f - \tilde{\mathbf{q}}(\tau_f) \cdot \mathbf{n}_f}{\tilde{\mathbf{q}}(\tau_i) \cdot \mathbf{n}_i - \tilde{\mathbf{q}}(\tau_i) \cdot \mathbf{n}_i} \right), \tag{102}$$

where the subscripts $f$ and $i$ present the final and the initial points of the worldline respectively, and the two vectors $\mathbf{q}$ and $\tilde{\mathbf{q}}$ determine a reference parallel transported normal frame satisfying,

$$\begin{aligned} \mathbf{q}^2 = -1, \qquad \tilde{\mathbf{q}}^2 = 1, \qquad \mathbf{q} \cdot \dot{X} = 0, \qquad \tilde{\mathbf{q}} \cdot \dot{X} = 0, \\ \mathbf{q} \cdot \tilde{\mathbf{q}} = 0, \qquad \nabla \mathbf{q} = 0, \qquad \nabla \tilde{\mathbf{q}} = 0. \end{aligned} \tag{103}$$

---

[17]When the bulk spacetime is not locally AdS$_3$ spacetime, there could be some non-geodesic solutions for the worldline action (100) [50]. One can consult [50, 91–93] for more discussions about the worldline action (100).

And the explicit expressions of the reference normal frame $\{\mathbf{q}, \tilde{\mathbf{q}}\}$ are given by

$$
\begin{aligned}
\mathbf{q} &= \frac{1}{\sqrt{2 l_U l_V \rho}} \left( l_U \partial_U - l_V \partial_V \right), \\
\tilde{\mathbf{q}} &= \sqrt{\frac{2}{l_U l_V \rho}} \left( -\frac{l_U (U+V)}{l_U + l_V} \partial_U - \frac{l_V (U+V)}{l_U + l_V} \partial_V + 2 \rho \, \partial_\rho \right),
\end{aligned}
\tag{104}
$$

up to an overall sign depends on a choice of handedness.

Note that, the CS term in the action (85) lacks of diffeomorphism invariance, which results in the fact that the $S_{CS}$ correction depends on the choice of the normal vector at the endpoints of $\mathcal{C}$ (see [50] for more discussions). Also, the configuration of the normal frame, i.e. the normal vectors $\mathbf{n}$ and $\tilde{\mathbf{n}}$, cannot be determined by extremizing the worldline action. Fortunately, the worldline $\mathcal{C}$ representing a holographic entanglement entropy anchors on the boundary, hence one can take the following boundary condition for $\mathbf{n}$,

$$
\mathbf{n}_i = \mathbf{n}_f \propto \partial_t,
\tag{105}
$$

where $t$ is the temporal coordinate of the boundary $\text{CFT}_2$, it has been verified in [50] that $S_{CS}$ for this boundary condition exactly matches the anomalous part $S_{\mathcal{A}}^a$ (96) with the cutoffs being $\epsilon_U = \epsilon_V = \epsilon$. Furthermore, in Sec.6, we point out that this boundary condition (105) of $\mathbf{n}$ indeed comes from a specific regulation $\epsilon_U = \epsilon_V$, see (161). Nevertheless, although we have fixed the boundary condition for $\mathbf{n}$, the specific value of $\mathbf{n}$ remains highly non-unique over the entire RT surface. More importantly, unlike the RT surface, the twist description is not purely geometric, which poses difficulties for calculating or even defining the anomalous part of the PEE, as well as the EWCS. One main purpose of this paper is to search for a purely geometric picture of the anomalous part such that $S_{\mathcal{A}}^a$ corresponds to the area (or length) of some newly-defined geometric quantity under a proper regulation, similarly to the case of the RT prescription.

### 4.3 Gravitational anomalous entanglement from the inner RT surface

According to (93), the correction of the thermal entropy of a BTZ black hole from the CS term is proportional to the length of the inner horizon.[18] Also, it is well known that, the entanglement entropy of a boundary interval of $\text{AdS}_3$ is also given by the thermal entropy of the Rindler black string [13, 59]. Then it is natural to think that, the CS correction to the holographic entanglement entropy for intervals should be represented via the inner horizon in the Rindler $\text{AdS}_3$, as well as its pre-image, the IRT surface $\widehat{\mathcal{E}}$ (31). On the other hand, we have reviewed that the CS correction (102) can be represented via the normal frame along the RT surface with no reference to the IRT surface. The main target of this section is to establish the representation of $S_{CS}$ via the IRT surface, and uncover the hidden relationship between the IRT surface and normal frame representations.

Another reason why we relate the $S_{CS}$ to the inner horizon comes from the correction of the modular Hamiltonian. Due to the CS term, the stress tensor of the thermal state (i.e. the Rindler $\widetilde{\text{AdS}}_3$) is corrected in the following way [86],

$$
T_{\tilde{t}\tilde{t}}^{(\mu^{-1})} = T_{\tilde{t}\tilde{t}}^{(0)} + \frac{1}{\mu} T_{\tilde{t}\tilde{x}}^{(0)}, \qquad T_{\tilde{t}\tilde{x}}^{(\mu^{-1})} = T_{\tilde{t}\tilde{x}}^{(0)} + \frac{1}{\mu} T_{\tilde{t}\tilde{t}}^{(0)},
\tag{106}
$$

---

[18]This statement and (107) are our starting points, connecting the correction of the holographic entanglement entropy from the CS term to the IRT surface. However, they are only valid for 3-dimensional TMG, hence our results are only applicable to the holographic two-dimensional CFT with gravitational anomalies. In fact, the derivation of the worldline action (100) is also only valid for 3-dimensional TMG.

where the superscript $(\mu^{-1})$ denotes quantities with the presence of the CS term, while the superscript $(0)$ denotes quantities in the absence of the CS term when $\mu \to \infty$. By repeating the derivation in appendix A, one can find,

$$H_{\text{mod}}^{(\mu^{-1})} = H_{\text{mod}}^{(0)} + \frac{1}{\mu} P_{\text{mod}}^{(0)} \tag{107}$$

hence the correction from the CS term is captured by the modular momentum $P_{\text{mod}}^{(0)}$, which generates the modular momentum flow with the fixed points located on the IRT surface.

In this section, we realize this idea and find the same results as those obtained by using the normal frames (102). Again let us begin with the regularized interval, and its image in the Rindler $\widetilde{\text{AdS}}_3$:

$$\mathcal{A}^{\text{reg}} : (-l_U/2 + \epsilon_U, -l_V/2 + \epsilon_V) \to (l_U/2 - \epsilon_U, l_V/2 - \epsilon_V) , \tag{108}$$

$$\tilde{\mathcal{A}}^{\text{reg}} : (-\Delta \tilde{U}/2, -\Delta \tilde{V}/2) \to (\Delta \tilde{U}/2, \Delta \tilde{V}/2), \tag{109}$$

where $\Delta \tilde{U} = \frac{1}{T_{\tilde{U}}} \log\left(\frac{l_U}{\epsilon_U}\right)$, $\Delta \tilde{V} = \frac{1}{T_{\tilde{V}}} \log\left(\frac{l_V}{\epsilon_V}\right)$. As we have reviewed, after the Rindler transformation, the reduced density matrix $\rho_{\mathcal{A}}$ is mapped to a thermal state in the Rindler spacetime with translation symmetries along $\tilde{x}$ and $\tilde{t}$ directions, hence the entanglement contour for both the entanglement entropy and the timelike entanglement entropy is flat, and the two entanglement entropies satisfy the volume law. After the regulation, the length of $\tilde{\mathcal{A}}^{\text{reg}}$ is measurable, and we can compute the entanglement entropy by calculating the regulated lengths of its partner geodesic chords on the horizons, for example the normal part of the entanglement entropy is calculated by the length of its partner geodesic chord on the outer horizon (18), while the anomalous part of the entanglement entropy comes from the length of its partner on the inner horizon (94).

First, let us consider the normal part of the entanglement entropy and ignore the anomalous contribution. Actually, what we are computing is the contribution to the entropy $S_{\tilde{\mathcal{A}}}^n$ of the Rindler $\widetilde{\text{AdS}}_3$ from the regulated interval $\tilde{\mathcal{A}}^{\text{reg}}$, i.e. the PEE $s_{\tilde{\mathcal{A}}}^n(\tilde{\mathcal{A}}^{\text{reg}})|_{\tilde{\mathcal{A}}^{\text{reg}} \to \tilde{\mathcal{A}}}$. Here $\tilde{\mathcal{A}}$ is the image of $\mathcal{A}$ which has infinite length. Since the PEE structure is invariant under the Rindler transformations, which are symmetries of the CFT, we should have

$$s_{\tilde{\mathcal{A}}}^n(\tilde{\mathcal{A}}^{\text{reg}})|_{\tilde{\mathcal{A}}^{\text{reg}} \to \tilde{\mathcal{A}}} = s_{\mathcal{A}}^n(\mathcal{A}^{\text{reg}})|_{\mathcal{A}^{\text{reg}} \to \mathcal{A}} = S_{\mathcal{A}}^n . \tag{110}$$

The next step is to identify the partner geodesic chord $\mathcal{E}^{\text{reg}}$ of $\mathcal{A}^{\text{reg}}$ on the RT surface $\mathcal{E}$, hence we can compute the PEE using the correspondence (56) between the PEE and geodesic chords on the RT surface $\mathcal{E}$. The computation is now straightforward by replacing the $\mathcal{A}_i$ in (58) and (59) by $\mathcal{A}^{\text{reg}}$ (108). After we ignore the higher order terms $\mathcal{O}(\epsilon^2)$, we get

$$S_{\mathcal{A}}^n = \frac{1}{4G} \log\left(\frac{l_U l_V}{\epsilon_U \epsilon_V}\right). \tag{111}$$

Then we compute the anomalous part of the entanglement entropy using the same prescription,

$$s_{\tilde{\mathcal{A}}}^a(\tilde{\mathcal{A}}^{\text{reg}})|_{\tilde{\mathcal{A}}^{\text{reg}} \to \tilde{\mathcal{A}}} = s_{\mathcal{A}}^a(\mathcal{A}^{\text{reg}})|_{\mathcal{A}^{\text{reg}} \to \mathcal{A}} = S_{\mathcal{A}}^a . \tag{112}$$

Since the anomalous part of the thermal entropy for the Rindler $\widetilde{\text{AdS}}_3$ is proportional to the area of the inner horizon with a coefficient $1/(4\mu G)$, the correspondence between the anomalous PEE (*APEE*) $s_{\mathcal{A}}^a(\mathcal{A}_i)$ and the area of the partner geodesic chord should be modified accordingly,

$$s_{\mathcal{A}}^a(\mathcal{A}^{\text{reg}}) = \frac{\text{Length}(\widehat{\mathcal{E}}^{\text{reg}})}{4G\mu} = \frac{\widehat{\lambda}(Q) - \widehat{\lambda}(P)}{4G}$$

$$= \frac{1}{4\mu G} \log\left(\frac{l_U \epsilon_V}{l_V \epsilon_U}\right), \tag{113}$$

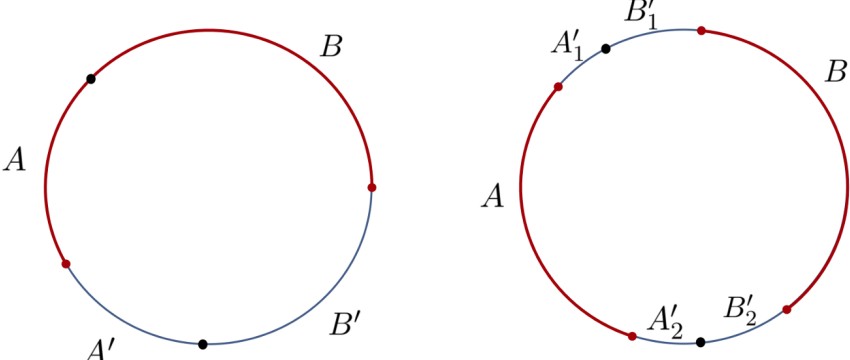

Figure 9: The figures are extracted from [75]. The region $A'B'$ serves as an auxiliary system to purify the bipartite system $\rho_{AB}$. The left figure represents the adjacent case. While the right figure represents the non-adjacent case, where we need to make a further separation, $A' = A'_1 \cup A'_2$ and $B' = B'_1 \cup B'_2$.

where $P$ and $Q$ are the left and the right endpoints of $\mathcal{A}^{\text{reg}}$, and we have replaced $\widehat{\mathcal{A}_i}$ in (78) with $\mathcal{A}^{\text{reg}}$ (108), and ignore the higher order terms $\mathcal{O}(\epsilon^2)$. Finally we get the entanglement entropy (95) for intervals in gravitational anomalous CFT$_2$, which coincides with the results obtained in [51].

One can also identify the partner geodesic chord of $\mathcal{A}^{\text{reg}}$ on the IRT surface using a prescription similar to the so-called swing surface prescription using null geodesics emanating from the endpoints of $\mathcal{A}$. This prescription indeed uses the intersection lines (which are null lines) between the modular momentum slices and the inner horizon in the Rindler $\widetilde{\text{AdS}_3}$. Nevertheless, in the original AdS$_3$, the image of these null lines all anchor on the boundary at the endpoints of $\mathcal{A}$, we need further input to clarify which null line gives the correspondence between the endpoints of $\mathcal{A}^{\text{reg}}$ and their partner points on $\widehat{\mathcal{E}}$. When taking the flat limit, we recover the swing surface picture proposed in [16], see appendix F for more details.

## 5 BPE and EWCS with gravitational anomaly

### 5.1 A brief introduction to the BPE with gravitational anomaly

In general, the balanced partial entanglement entropy (BPE) is a special PEE that satisfies the certain balanced conditions and is claimed to be dual to the EWCS [75,76]. In [75,76], it was shown that the BPE exactly captures the same type of mixed state correlations as the reflected entropy, and is conjectured to be purification independent. These conjectures have passed various tests [51,71,75,76,79,81].

Consider a bipartite system $\mathcal{H}_A \otimes \mathcal{H}_B$ with the density matrix $\rho_{AB}$, we can introduce an auxiliary system $A'B'$ to purify it such that the whole system $ABA'B'$ is in a pure state $|\psi\rangle$, and satisfies $\text{Tr}_{A'B'} |\psi\rangle \langle\psi| = \rho_{AB}$. Then the BPE between $A$ and $B$ is defined as,

$$\text{BPE}(A, B) = s_{AA'}(A)|_{\text{balanced}}, \tag{114}$$

where the subscript "balanced" means that the partition of the auxiliary system $A'B'$ should satisfy the balance conditions, which are listed in the following:

1. When $A$ and $B$ are adjacent, the balance condition is,[19]

$$s_{AA'}(A) = s_{BB'}(B) . \tag{115}$$

2. When $A$ and $B$ are non-adjacent, we need to further decompose the disconnected regions, $A' = A'_1 \cup A'_2$ and $B' = B'_1 \cup B'_2$. In this case, we have two balance conditions,

$$s_{AA'}(A) = s_{BB'}(B) , \qquad s_{AA'}\left(A'_1\right) = s_{BB'}\left(B'_1\right) . \tag{116}$$

See Fig.9 for an illustration. The partition points in the auxiliary system $A'B'$ are called the balanced points which are determined by the balance conditions. When the gravitational anomaly is taken into account, the normal part and the anomalous part of the holographic entanglement entropy should satisfy the balance conditions independently [51,94], for example in the adjacent cases we should have,

$$s^n_{AA'}(A) = s^n_{BB'}(B) , \qquad s^a_{AA'}(A) = s^a_{BB'}(B) , \tag{117}$$

and the BPE can also be decomposed into two parts

$$\text{BPE}^n(A,B) = s^n_{AA'}(A)|_{\text{balanced}} , \qquad \text{BPE}^a(A,B) = s^a_{AA'}(A)|_{\text{balanced}} . \tag{118}$$

For various covariant configurations in the gravitational anomalous CFT$_2$, both the normal and the anomalous parts of the BPE$(A,B)$ have been carried out carefully using the ALC proposal (44) in [51]. For example, let us consider the configurations in the right figure of Fig.9, where $A'_1$ and $A'_2$ satisfy the balance condition. Using the ALC proposal, the BPE$^a(A,B)$ is,

$$\text{BPE}^a(A,B) = s^a_{A'_1 AA'_2}(A)|_{\text{balanced}} = \frac{1}{2}\left(S^a_{A'_1 A} + S^a_{AA'_2} - S^a_{A'_1} - S^a_{A'_2}\right) . \tag{119}$$

Following our analysis on the fine structure of the extended entanglement wedge, the APEE $s^a_{A'_1 AA'_2}(A)$ also has a geometric picture, which is the partner geodesic chord of $A$ on the IRT surface $\widehat{\mathcal{E}}_{A'_1 AA'_2}$ of $A'_1 AA'_2$, we call this geodesic chord $\widehat{\mathcal{E}}_{A'_1 AA'_2}(A)$. More explicitly, we have,

$$s^a_{A'_1 AA'_2}(A) = \frac{\text{Length}(\widehat{\mathcal{E}}_{A'_1 AA'_2}(A))}{4\mu G} . \tag{120}$$

The computation can be explicitly done by adapting this configuration to the formula (78). As we have shown, the ALC proposal and the geometric picture coincide, hence (120) and (119) give the same result. Since the BPE is a special PEE that satisfies the balanced condition (116), its anomalous part BPE$^a(A,B)$ has a natural geometric interpretation in terms of subregions of the inner horizon or IRT surface, which is the main topic in this section.

On the gravity side, the length of the EWCS is no longer enough to reproduce both of the normal and the anomalous parts of the BPE$(A,B)$. We can also decompose the gravity dual of the BPE$(A,B)$ into the normal part and the anomalous part, i.e.

$$E_W(A,B) = E^n_W(A,B) + E^a_W(A,B) , \tag{121}$$

where the normal part is given by

$$E^n_W(A,B) = \frac{1}{4G}\text{Length}(\Sigma_{AB}) , \tag{122}$$

---

[19]In fact, the solution to the balance conditions may not be unique. In such cases, we choose the minimal one as our definition, which called the minimal condition [75].

with $\Sigma_{AB}$ representing the EWCS of the entanglement wedge $\mathcal{W}_{AB}$. As in the non-anomalous configurations [75, 76], the length of the EWCS matches the normal part of the BPE (see also [51] for the covariant configurations)

$$E_W^n(A,B) = \text{BPE}^n(A,B) = \frac{1}{4G}\text{Length}\left(\Sigma_{AB}\right). \tag{123}$$

One the other hand, the anomalous part $E_W^a(A,B)$ is supposed to be the geometric picture for the anomalous part of the BPE. Since the normal part $E_W^n(A,B)$ matches the $\text{BPE}^n(A,B)$, we expect the anomalous part $E_W^a(A,B)$ to match $\text{BPE}^a(A,B)$. According to our previous discussion, the geometric picture for the anomalous part is just the geodesic chord on the IRT surface $\widehat{\mathcal{E}}_{A_1'AA_2'}(A)$, hence we should have

$$E_W^a(A,B) = \text{BPE}^a(A,B) = \frac{1}{4\mu G}\text{Length}(\widehat{\mathcal{E}}_{A_1'AA_2'}(A)). \tag{124}$$

Another construction for the geometric picture of $\text{BPE}^a(A,B)$ was given in [51] using the twist description [50]. Unlike the case for the RT surface, the endpoints of the EWCS are settled in the bulk, naively choosing the time direction as the normal vector in the normal frame is not well motivated, and furthermore the time direction is in general not normal to the EWCS. Interestingly, since the EWCS anchors on the RT surfaces of $AB$ vertically, the authors of [51] proposed that, spacelike normal vector $\tilde{\mathbf{n}}$ can be chosen to be the tangent vector of the RT surfaces of $AB$. With the normal vectors $\tilde{\mathbf{n}}_i$ and $\tilde{\mathbf{n}}_f$ at the endpoints of the EWCS determined, we can compute the CS correction to the EWCS by (102), and the results exactly match the anomalous part of the BPE

$$\text{BPE}^a(A,B) = \frac{1}{4\mu G}\log\left(\frac{\mathbf{q}\left(\tau_f\right)\cdot\mathbf{n}_f - \tilde{\mathbf{q}}\left(\tau_f\right)\cdot\mathbf{n}_f}{\tilde{\mathbf{q}}\left(\tau_i\right)\cdot\mathbf{n}_i - \tilde{\mathbf{q}}\left(\tau_i\right)\cdot\mathbf{n}_i}\right). \tag{125}$$

Both $\mathbf{n}_i$ and $\mathbf{n}_f$ can be determined up to an overall sign according to a choice of handedness. Here we only need to choose the signs to ensure that the term inside the logarithm is positive. This result also coincide with the reflected entropy [95] for gravitational anomalous $\text{CFT}_2$ [96].

In this section, we demonstrate that the geodesic chord $\widehat{\mathcal{E}}_{A_1'AA_2'}(A)$ is also the saddle geodesic connecting the two pieces of the IRT surface $\widehat{\mathcal{E}}_{AB}$ of $A\cup B$, which is exactly the same as the EWCS $\Sigma_{AB}$. Furthermore, in the next section, we discuss the geometric construction of the APEE in the twist description, based on the fine structure of the extended entanglement wedge. In that context, we point out that the right hand side of (125) indeed evaluates the APEE $s^a_{A_1'AA_2'}(A)$, which explains the equality between the two sides of (125), see appendix G for details.

## 5.2 A case study of non-adjacent intervals

### 5.2.1 $\text{BPE}^n(A,B)$ and the EWCS

Let us give an explicit example of two non-adjacent intervals $A$ and $B$ with connected entanglement wedge $\mathcal{W}_{AB}$, where

$$A: (U_1,V_1)\to(U_2,V_2), \qquad B: (U_3,V_3)\to\left(U_4,V_4\right). \tag{126}$$

Due to the conformal symmetry, the EWCS only depends on two conformal invariants–$\eta$ and $\bar{\eta}$,

$$\eta = \frac{(V_2-V_1)\left(V_4-V_3\right)}{(V_3-V_1)\left(V_4-V_2\right)}, \qquad \bar{\eta} = \frac{(U_2-U_1)\left(U_4-U_3\right)}{(U_3-U_1)\left(U_4-U_2\right)}. \tag{127}$$

Therefore, without losing generality, we can fix three endpoints of the interval $A, B$, and set the parameters $(U_4, V_4)$ free, for example,

$$U_1 = V_1 = 0, \qquad U_2 = V_2 = \frac{1}{2}, \qquad U_3 = V_3 = \frac{3}{2}. \tag{128}$$

Furthermore, it is convenient to rewrite the parameters $(U_4, V_4)$ in terms of the conformal invariants $(\eta, \bar{\eta})$ which can simplify the calculation,

$$U_4 = \frac{3(\bar{\eta} - 1)}{2(3\bar{\eta} - 1)}, \qquad V_4 = \frac{3(\eta - 1)}{2(3\eta - 1)}. \tag{129}$$

Since the the entanglement wedge is connected, the RT surface has two pieces $\mathcal{E}_{AB} = \mathcal{E}_1 \cup \mathcal{E}_2$, which are given by

$$
\begin{aligned}
\mathcal{E}_1 : \; & \rho(V) = \frac{(\eta - 1)(3\bar{\eta} - 1)}{V(3\eta - 3 - 2V(3\eta - 1))(\bar{\eta} - 1)}, \quad U(V) = \frac{(3\eta - 1)(\bar{\eta} - 1)}{(\eta - 1)(3\bar{\eta} - 1)} V, \\
\mathcal{E}_2 : \; & \rho(V) = -\frac{2}{3 + 4(-2 + V)V}, \qquad U(V) = V,
\end{aligned}
\tag{130}
$$

where $\mathcal{E}_1$ and $\mathcal{E}_2$ are also the RT surfaces of $AA_2'B_2'B$ and $A_2'B_2'$, respectively, see Fig.10. As we discussed above, the $\Sigma_{AB}$ is a type I spacelike geodesic[20] connecting the RT surfaces $\mathcal{E}_1$ and $\mathcal{E}_2$ while satisfying the extremal condition

$$\frac{\partial L\left(U_{H_1}, V_{H_1}, \rho_{H_1}, U_{H_2}, V_{H_2}, \rho_{H_2}\right)}{\partial V_{H_1}} = \frac{\partial L\left(U_{H_1}, V_{H_1}, \rho_{H_1}, U_{H_2}, V_{H_2}, \rho_{H_2}\right)}{\partial V_{H_2}} = 0, \tag{131}$$

where $L(U_{H_1}, V_{H_1}, \rho_{H_1}, U_{H_2}, V_{H_2}, \rho_{H_2})$ represents the proper distance (C.5) between $H_1$ and $H_2$, and $H_1$ and $H_2$ are the points where $\Sigma_{AB}$ anchors on $\mathcal{E}_1$ and $\mathcal{E}_2$, see Fig.10. The solution of this extremal condition is given by

$$V_{H_1} = \frac{3(\eta - 1)(3\bar{\eta} - 1)}{2 + 8\sqrt{\eta\bar{\eta}} - 6\bar{\eta} + 6\eta(3\bar{\eta} - 1)}, \qquad V_{H_2} = \frac{1}{2} + \frac{1}{1 + 9\sqrt{\eta\bar{\eta}}}, \tag{132}$$

hence, we arrive at

$$E_W^n(A, B) = \frac{1}{4G}\text{Length}(\Sigma_{AB}) = \frac{1}{8G}\log\frac{(\sqrt{\eta} + 1)(\sqrt{\bar{\eta}} + 1)}{(\sqrt{\eta} - 1)(\sqrt{\bar{\eta}} - 1)}. \tag{133}$$

Following [51], we can extend $\Sigma_{AB}$ to an RT surface that anchors on the boundary at $Q_1$ and $Q_2$. The RT surface corresponds to the boundary interval $A_1' A A_2'$ shown in Fig.10. The equation of $\Sigma_{AB}$ with its extension is[21]

$$\rho(V) = \frac{2\tilde{l}_V}{\tilde{l}_U\left(\tilde{l}_V^2 - 4(V - V_0)^2\right)}, \quad U(V) = \frac{\tilde{l}_U}{\tilde{l}_V}(V - V_0) + U_0, \tag{134}$$

according to (C.3) and (132), the parameters $(\tilde{l}_U, \tilde{l}_V, V_0, U_0)$ are given by

$$\tilde{l}_U = \frac{6\sqrt{\bar{\eta}}}{9\bar{\eta} - 1}, \qquad \tilde{l}_V = \frac{6\sqrt{\eta}}{9\eta - 1}, \qquad V_0 = \frac{3(1 - 3\eta)}{2(1 - 9\eta)}, \qquad U_0 = \frac{3(1 - 3\bar{\eta})}{2(1 - 9\bar{\eta})}, \tag{135}$$

---

[20]Here "type I" means that $\Sigma_{AB}$ and its extension can be referred to as the RT surface of a certain boundary interval. See appendix C for details.

[21]We hope that the same symbols "$(\tilde{l}_U, \tilde{l}_V, V_0, U_0)$" with different meanings in (48) and (134) do not cause any confusion.

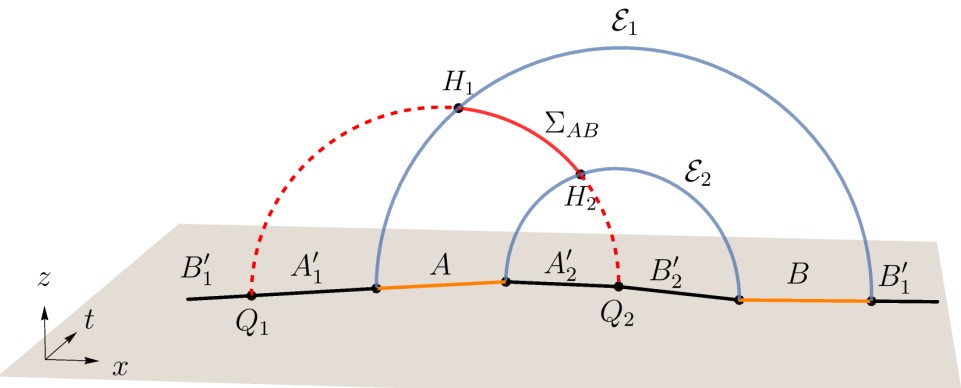

Figure 10: Extracted from [51]. Illustration for the EWCS in the non-adjacent case. The extension of the EWCS $\Sigma_{AB}$ intersects with the asymptotic boundary at the two points $Q_1$ and $Q_2$.

hence the extension of the $\Sigma_{AB}$ intersects with the asymptotic boundary at the two points $Q_1 = \left(U_{Q_1}, V_{Q_1}\right)$ and $Q_2 = \left(U_{Q_2}, V_{Q_2}\right)$, see Fig.10,

$$
\begin{aligned}
U_{Q_1} &= -\frac{\widetilde{l}_U}{2} + \tilde{y} = \frac{1}{2} - \frac{1}{3\sqrt{\bar{\eta}} - 1}, & V_{Q_1} &= -\frac{\widetilde{l}_V}{2} + \tilde{x} = \frac{1}{2} - \frac{1}{3\sqrt{\bar{\eta}} - 1}, \\
U_{Q_2} &= \frac{\widetilde{l}_U}{2} + \tilde{y} = \frac{1}{2} + \frac{1}{3\sqrt{\bar{\eta}} + 1}, & V_{Q_2} &= \frac{\widetilde{l}_V}{2} + \tilde{x} = \frac{1}{2} + \frac{1}{3\sqrt{\bar{\eta}} + 1}.
\end{aligned}
\tag{136}
$$

Without loss of generality, we suppose that $A$ have a smaller size than that of $B$, i.e.

$$
U_4 - U_3 > U_2 - U_1, \quad V_4 - V_3 > V_2 - V_1 \Leftrightarrow \frac{1}{9} < \eta, \quad \bar{\eta} < \frac{1}{3},
\tag{137}
$$

such that the point $Q_1$ is on the left of the interval $A$. Otherwise, it would be on the right-hand side of $B$. The points $Q_1, Q_2$ and the endpoints of $AB$ make a decomposition for the time slice where the bipartite system $AB$ are settled on, see Fig.10. Given the partition points $Q_1$ and $Q_2$, then $A_1'$, $A_2'$, $B_1'$ and $B_2'$ regions are also determined. In [51], the authors show that balance conditions (117) are satisfied by this partition. Therefore,

$$
\begin{aligned}
\mathrm{BPE}^n(A, B) &= s_{A_1'AA_2'}^n(A) = s_{B_1'BB_2'}^n(B) = \frac{1}{8G} \log \frac{\left(\sqrt{\eta} + 1\right)\left(\sqrt{\bar{\eta}} + 1\right)}{\left(\sqrt{\eta} - 1\right)\left(\sqrt{\bar{\eta}} - 1\right)}, \\
\mathrm{BPE}^a(A, B) &= s_{A_1'AA_2'}^a(A) = s_{B_1'BB_2'}^a(B) = \frac{1}{8\mu G} \log \frac{\left(\sqrt{\eta} - 1\right)\left(\sqrt{\bar{\eta}} + 1\right)}{\left(\sqrt{\eta} + 1\right)\left(\sqrt{\bar{\eta}} - 1\right)},
\end{aligned}
\tag{138}
$$

where the PEEs and the APEEs are calculated by the ALC proposal. Hence, we conclude that,

$$
E_W^n(A, B) = \frac{1}{4G} \mathrm{Length}(\Sigma_{AB}) = s_{A_1'AA_2'}^n(A)|_{balanced} = \mathrm{BPE}^n(A, B).
\tag{139}
$$

### 5.2.2 BPE$^a(A, B)$ and the inner EWCS

Given the partition points $Q_1$ and $Q_2$ (136), we can also compute the length of the partner geodesic chord $\widehat{\mathcal{E}}_{A_1'AA_2'}(A)$, and verify the coincidence between it and BPE$^a(A, B)$ (138). More

explicitly, for an arbitrary boundary point $P = (U_P, V_P)$ in the casual development of the interval $A'_1 A A'_2$, the function $\widehat{\lambda}(P)$ is given by (76)

$$\widehat{\lambda}(P) = \frac{1}{2} \log \left( \frac{\left[ \widetilde{l_U} + 2(U_P - U_0) \right] \left[ \widetilde{l_V} - 2(V_P - V_0) \right]}{\left[ \widetilde{l_U} - 2(U_P - U_0) \right] \left[ \widetilde{l_V} + 2(V_P - V_0) \right]} \right), \tag{140}$$

where the parameters $(\widetilde{l_U}, \widetilde{l_V}, V_0, U_0)$ are specified by (135). Therefore, the anomalous part $E_W^a(A, B)$ is

$$\begin{aligned}
E_W^a(A, B) &= \frac{\text{Length}(\widehat{\mathcal{E}}_{A'_1 A A'_2}(A))}{4\mu G} = \frac{\widehat{\lambda}(P_2) - \widehat{\lambda}(P_1)}{4\mu G} \\
&= \frac{1}{8\mu G} \log \frac{\left( \sqrt{\eta} - 1 \right) \left( \sqrt{\bar\eta} + 1 \right)}{\left( \sqrt{\eta} + 1 \right) \left( \sqrt{\bar\eta} - 1 \right)},
\end{aligned} \tag{141}$$

where $P_1$ and $P_2$ are the left and the right endpoints of $A$ respectively, as specified by (126). More importantly, the result aligns exactly with the $\text{BPE}^a(A, B)$ (138) evaluated by the ALC proposal.

More interestingly, we find that the partner geodesic chord $\widehat{\mathcal{E}}_{A'_1 A A'_2}(A)$ is also the saddle geodesic that connecting the two pieces of the IRT surface $\widehat{\mathcal{E}}_{AB}$ of $AB$, which we call the inner EWCS (or IEWCS for short). In such a way, the calculation of $E_W^a(A, B)$ is also an optimization problem, which is the same as the calculation of $E_W^n(A, B)$. Since the entanglement wedge $\mathcal{W}_{AB}$ is in the connected phase, and the RT surface $\mathcal{E}_{AB} = \mathcal{E}_1 \cup \mathcal{E}_2$, where $\mathcal{E}_1$ and $\mathcal{E}_2$ are also the RT surfaces of $AA'_2 B'_2 B$ and $A'_2 B'_2$, respectively. It is natural to take the IRT surface for $AB$ to be

$$\widehat{\mathcal{E}}_{AB} = \widehat{\mathcal{E}}_1 \cup \widehat{\mathcal{E}}_2, \tag{142}$$

which is the union of the two IRT surfaces of $AA'_2 B'_2 B$ and $A'_2 B'_2$, see Fig.11. More explicitly, we can write down the functions for these two IRT surfaces

$$\begin{aligned}
\widehat{\mathcal{E}}_1 &: \rho(V) = \frac{(\eta - 1)(3\bar\eta - 1)}{V(3 - 2V + 3(2V - 1)\eta)(\bar\eta - 1)}, \quad U(V) = \frac{(3 - 2V + 3(2V - 1)\eta)(1 - \bar\eta)}{2(\eta - 1)(3\bar\eta - 1)}, \\
\widehat{\mathcal{E}}_2 &: \rho(V) = \frac{2}{3 + 4(-2 + V)V}, \quad U(V) = 2 - V.
\end{aligned} \tag{143}$$

Similarly, the IEWCS $\widehat{\Sigma}_{AB}$ is the saddle type II spacelike geodesic[22] connecting the IRT surfaces $\widehat{\mathcal{E}}_1$ and $\widehat{\mathcal{E}}_2$. To calculate the IEWCS $\widehat{\Sigma}_{AB}$, we also impose the extremal condition

$$\frac{\partial L(U_{\widehat{H}_1}, V_{\widehat{H}_1}, \rho_{\widehat{H}_1}, U_{\widehat{H}_2}, V_{\widehat{H}_2}, \rho_{\widehat{H}_2})}{\partial V_{\widehat{H}_1}} = \frac{\partial L(U_{\widehat{H}_1}, V_{\widehat{H}_1}, \rho_{\widehat{H}_1}, U_{\widehat{H}_2}, V_{\widehat{H}_2}, \rho_{\widehat{H}_2})}{\partial V_{\widehat{H}_2}} = 0, \tag{144}$$

where we also take $L(U_{\widehat{H}_1}, V_{\widehat{H}_1}, \rho_{\widehat{H}_1}, U_{\widehat{H}_2}, V_{\widehat{H}_2}, \rho_{\widehat{H}_2})$ as (C.5), which represents the proper distance between $\widehat{H}_1$ and $\widehat{H}_2$ but may differs by a negative sign, depending on whether $\widehat{H}_1$ and $\widehat{H}_2$ are on the same sector of the type II spacelike geodesic, see appendix C for details. Nevertheless, the overall sign does not affect our discussion, since it does not affect the solution of this extremal condition. And $\widehat{H}_1$ and $\widehat{H}_2$ are the intersection points of $\widehat{\Sigma}_{AB}$ with the IRT surfaces $\widehat{\mathcal{E}}_{AA'_2 B'_2 B}$ and $\widehat{\mathcal{E}}_{A'_2 B'_2}$ respectively, see Fig.11. The solution of this extremal condition is given by

$$V_{\widehat{H}_1} = \frac{3(\eta - 1)\sqrt{\bar\eta}}{2\left( \sqrt{\eta} - \sqrt{\bar\eta} \right)\left( 1 + 3\sqrt{\eta\bar\eta} \right)}, \qquad V_{\widehat{H}_2} = \frac{3}{2} + \frac{\sqrt{\eta}}{\sqrt{\bar\eta} - \sqrt{\eta}}. \tag{145}$$

---

[22]Here "type II" means that $\widehat{\Sigma}_{AB}$ and its extension can be referred to as the IRT surface of a certain boundary interval. See appendix C for details.

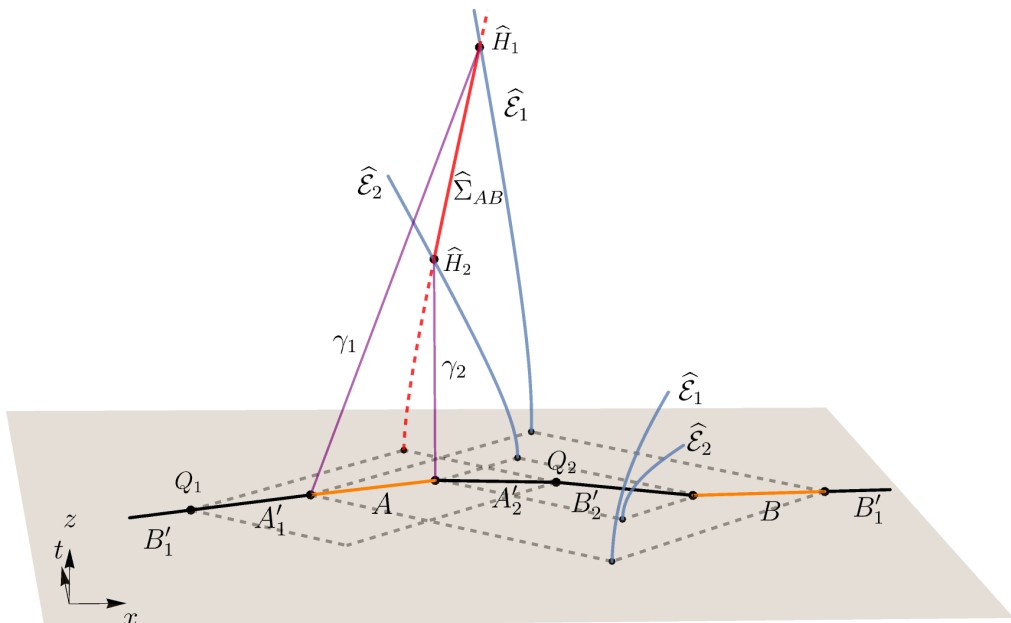

Figure 11: Illustration for the anomalous part $E_W^a(A,B)$ in the non-adjacent case. The dashed line represents the boundary of the causal development. The purple lines $\gamma_1$ and $\gamma_2$ represent the null geodesics. The IEWCS $\widehat{\Sigma}_{AB}$ (the red solid line) and its extension (the red dashed line) can be referred to as the IRT surface of the interval $A_1'AA_2'$ whose endpoints are also $Q_1$ and $Q_2$.

Interestingly, $\widehat{\Sigma}_{AB}$ and its extension can also be referred to as the IRT surface $\widehat{\mathcal{E}}_{A_1'AA_2'}$ of the interval $A_1'AA_2'$, whose equation is given by

$$\widehat{\mathcal{E}}_{A_1'AA_2'}: \ \rho(V) = -\frac{2\widetilde{l}_V}{\widetilde{l}_U\left(\widetilde{l}_V^{\,2} - 4(V-V_0)^2\right)}, \qquad U(V) = -\frac{\widetilde{l}_U}{\widetilde{l}_V}(V-V_0) + U_0, \qquad (146)$$

see Fig.11 for an illustration. Here, the parameters $(\widetilde{l}_U, \widetilde{l}_V, V_0, U_0)$ are also given by (135). According to (32), the length parameter of the IRT surface $\widehat{\mathcal{E}}_{A_1'AA_2'}$ is,

$$\widehat{\tau} = \frac{1}{2}\log\left(\frac{2(V-V_0) - \widetilde{l}_V}{2(V-V_0) + \widetilde{l}_V}\right). \qquad (147)$$

Therefore, by substituting (145) into the above equation, the anomalous part $E_W^a(A,B)$ is given by the length of $\widehat{\Sigma}_{AB}$ divided by $4\mu G$,

$$E_W^a(A,B) = \frac{\text{Length}(\widehat{\Sigma}_{AB})}{4\mu G} = \frac{\widehat{\tau}_{\widehat{H}_2} - \widehat{\tau}_{\widehat{H}_1}}{4\mu G} = \frac{1}{8\mu G}\log\frac{\left(\sqrt{\overline{\eta}}-1\right)\left(\sqrt{\overline{\eta}}+1\right)}{\left(\sqrt{\overline{\eta}}+1\right)\left(\sqrt{\overline{\eta}}-1\right)}, \qquad (148)$$

which coincides with (141) as we expected. Moreover, we emphasize that the IEWCS $\widehat{\Sigma}_{AB}$ is precisely the partner geodesic chord of $A$ on the IRT surface $\widehat{\mathcal{E}}_{A_1'AA_2'}$, which is $\widehat{\mathcal{E}}_{A_1'AA_2'}(A)$. This can be shown by demonstrating that the points $\widehat{H}_1$ and $\widehat{H}_2$ are the partner points corresponding to the left and right endpoints of $A$, respectively. In appendix D, we present a trick for locating the partner point $\widehat{H}$ on the IRT surface $\widehat{\mathcal{E}}$ of the boundary point $P$. More explicitly, the partner point $\widehat{H}$ is the intersection point between the null hypersurface introduced by $P$ and the IRT

surface $\widehat{\mathcal{E}}$, see Fig.15. Now, let us use this trick to indicate $\widehat{H}_1$ and $\widehat{H}_2$ are the partner points corresponding to the left and the right endpoints of $A$, respectively. For example, according to (C.12), the null hypersurface $\mathcal{N}_1$ introduced by the left endpoint $(U_1, V_1)$ of the interval $A$ is given by

$$\mathcal{N}_1 : \ \rho = -\frac{1}{2UV} \, . \tag{149}$$

One can easily verify that the null hypersurface $\mathcal{N}_1$ intersects with the IRT surface $\widehat{\mathcal{E}}_{A_1' A A_2'}$ (146) at $\widehat{H}_1$ (145). In other words, $\widehat{H}_1$ is the partner point on $\widehat{\mathcal{E}}_{A_1' A A_2'}$ of the left endpoint of $A$ according to the discussion in appendix D, and the geodesic connecting the left endpoint of $A$ with $\widehat{H}_1$ is a null geodesic, see Fig.11. A similar discussion applies to the right endpoints of $A$, which finishes our proof.

On the other hand, as the saddle geodesic connecting the two pieces of the IRT surface $\widehat{\mathcal{E}}_{AB}$, the IEWCS $\widehat{\Sigma}_{AB}$ also represents the timelike mixed state correlation between the timelike intervals $\widehat{A}$ and $\widehat{B}$. In fact, as we mention above, both the EWCS and the IEWCS should be understood as the mixed state correlations between two causal developments, $\mathcal{D}_A$ and $\mathcal{D}_B$, rather than just between the intervals $A$ and $B$, or $\widehat{A}$ and $\widehat{B}$, due to the Lorentz symmetry. Moreover, one can easily verify that the result (148) corresponding to the IEWCS cannot be derived from the EWCS (133) through analytical continuation.

# 6 Reproducing the twist description from the partner geodesic chord on the IRT surfaces

In this section, we aim to construct the geometric picture of the anomalous partial entanglement entropy in the twist description, and the interval $\mathcal{A}$ we consider is specified by (3). In Sec.3.2.2, we have analyzed the fine structure of the extended entanglement wedge using the modular momentum slices, and identified the partnership between points in the causal development $\mathcal{D}_{\mathcal{A}}$ and points on the RT and the IRT surfaces using two saddle geodesics $\widehat{\gamma}_P^1$ (48) and $\widehat{\gamma}_P^2$ (62), which are mapped to the curves along the $\tilde{\rho}$ direction in the Rindler $\widetilde{\mathrm{AdS}_3}$. For the boundary point $P = (U_1, V_1)$, the corresponding normalized tangent vectors at $H$ (the partner point of $P$ on $\mathcal{E}$) for $\mathcal{E}$, $\widehat{\gamma}_P^1$, and $\widehat{\gamma}_P^2$ are denoted as $v_{\mathcal{E}}|_H$, $v_{\widehat{\gamma}_P^1}|_H$, and $v_{\widehat{\gamma}_P^2}|_H$, respectively, which are given by

$$v_{\mathcal{E}}|_H = \frac{\xi_1 \xi_3}{2\xi_5^2} \left( l_U \partial_U + l_V \partial_V \right) + \frac{8 \left( l_V U_1 + l_U V_1 \right) \xi_5}{l_U l_V \xi_1 \xi_3} \partial_\rho \, ,$$

$$v_{\widehat{\gamma}_P^1}|_H = \frac{2 l_U l_V}{\xi_5^2} \left( \xi_1 V_1 \partial_U + \xi_3 U_1 \partial_V \right) - \frac{4 \left( l_U^2 l_V^2 - 16 U_1^2 V_1^2 \right)}{l_U l_V \xi_1 \xi_3} \partial_\rho \, , \tag{150}$$

$$v_{\widehat{\gamma}_P^2}|_H = \frac{1}{2\xi_5^2} \left( -l_U \xi_1 \xi_4 \partial_U + l_V \xi_2 \xi_3 \partial_V \right) + \frac{8 \left( l_U V_1 - l_V U_1 \right) \xi_5}{l_U l_V \xi_1 \xi_3} \partial_\rho \, ,$$

where $\left( \xi_1, \xi_2, \xi_3, \xi_4, \xi_5 \right)$ are,

$$\xi_1 = l_U^2 - 4U_1^2 \, , \quad \xi_2 = l_U^2 + 4U_1^2 \, , \quad \xi_3 = l_V^2 - 4V_1^2 \, , \quad \xi_4 = l_V^2 + 4V_1^2 \, , \quad \xi_5 = l_U l_V + 4U_1 V_1 \, . \tag{151}$$

More interestingly, these tangent vectors satisfy the following property,

$$(v_{\widehat{\gamma}_P^1}|_H)^2 = (v_{\mathcal{E}}|_H)^2 = 1 \, , \quad (v_{\widehat{\gamma}_P^2}|_H)^2 = -1 \, ,$$

$$v_{\mathcal{E}}|_H \cdot v_{\widehat{\gamma}_P^1}|_H = v_{\mathcal{E}}|_H \cdot v_{\widehat{\gamma}_P^2}|_H = v_{\widehat{\gamma}_P^2}|_H \cdot v_{\widehat{\gamma}_P^1}|_H = 0 \, , \tag{152}$$

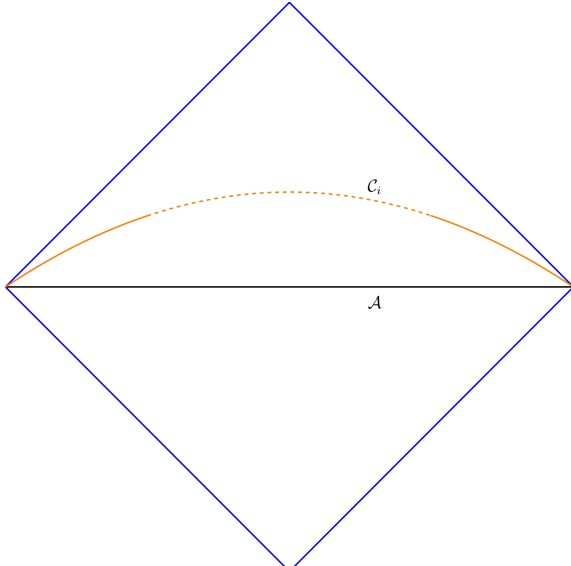

Figure 12: The blue lines represent the boundary of the causal development $\mathcal{D}_{\mathcal{A}}$. The orange curve $\mathcal{C}$ has the same endpoints with the interval $\mathcal{A}$, and the dashed part represents a subcurve $\mathcal{C}_i$.

which is exactly the same as (99). This gives a natural configuration for the normal vectors $\mathbf{n}, \tilde{\mathbf{n}}$ across the entire RT surface $\mathcal{E}$. More explicitly, consider an arbitrary boundary curve $\mathcal{C}$ (not necessarily a straight line), which has the same endpoints with the interval $\mathcal{A}$, see Fig.12. Given any point $P$ on $\mathcal{C}$, one can determine two corresponding saddle geodesics $\hat{\gamma}_P^{1,2}$, as well as their tangent vectors $v_{\hat{\gamma}_P^1}|_H$ and $v_{\hat{\gamma}_P^2}|_H$ at the partner point $H$ on $\mathcal{E}$. Then it is a natural choice to define these vectors as the values of the normal vectors $\mathbf{n}, \tilde{\mathbf{n}}$ at the point $H$. Since there is a one-to-one correspondence between the points on $\mathcal{C}$ and points on $\mathcal{E}$, we can determine the normal vectors $\mathbf{n}, \tilde{\mathbf{n}}$ across the entire RT surface using the same prescription

$$\tilde{\mathbf{n}}|_H := v_{\hat{\gamma}_P^1}|_H \,, \qquad \mathbf{n}|_H := v_{\hat{\gamma}_P^2}|_H \,. \tag{153}$$

Although the RT surface for $\mathcal{C}$ and $\mathcal{A}$ are the same, the configuration of the normal vectors $\mathbf{n}, \tilde{\mathbf{n}}$ on the RT surface $\mathcal{E}$ depends on the position of every point on $\mathcal{C}$. On the other hand, given a configuration of the normal vectors $\mathbf{n}, \tilde{\mathbf{n}}$ on $\mathcal{E}$, we can determine the boundary partner points of $\mathcal{E}$ by shooting the spacelike geodesics to the boundary with the initial direction $\tilde{\mathbf{n}}$.[23] Then the union of all the points where these geodesics anchor on the boundary form a unique spacelike curve $\mathcal{C}$ homologous to $\mathcal{E}$. Therefore, the boundary curve $\mathcal{C}$ and the configuration of the normal vectors $\mathbf{n}, \tilde{\mathbf{n}}$ are in one-to-one correspondence.

Given a configuration of the normal vectors $\mathbf{n}, \tilde{\mathbf{n}}$ on $\mathcal{E}$, the corresponding boundary curve is denoted as $\mathcal{C}$. In [50], the authors claimed that the anomalous part $S_{\mathcal{A}}^a$ is given by

$$S_{\mathcal{A}}^a = \frac{1}{4\mu G} \int_{\mathcal{E}} d\tau \, \tilde{\mathbf{n}} \cdot \nabla \mathbf{n} = \frac{1}{4\mu G} \int_{\mathcal{E}} d \log((\mathbf{q} - \tilde{\mathbf{q}}) \cdot \mathbf{n}) \,, \tag{154}$$

where the reference frame $\{\mathbf{q}, \tilde{\mathbf{q}}\}$ is given by (101). For arbitrary subregion $\mathcal{E}_i$ on $\mathcal{E}$, it determines a corresponding subcurve $\mathcal{C}_i$ based on the one-to-one correspondence between the points on $\mathcal{C}$ and those on $\mathcal{E}$, see Fig.12 for the subcurve $\mathcal{C}_i$. It is natural to interpret the part

---

[23]For an arbitrary point $H$ on $\mathcal{E}$, the corresponding modular Hamiltonian slice $\mathcal{P}_H$ is the set of the saddle spacelike geodesics that are normal to $\mathcal{E}$, hence the geodesic determined by the normal vector $\tilde{\mathbf{n}}|_H$ corresponds to a saddle geodesic which has an intersection point with the boundary.

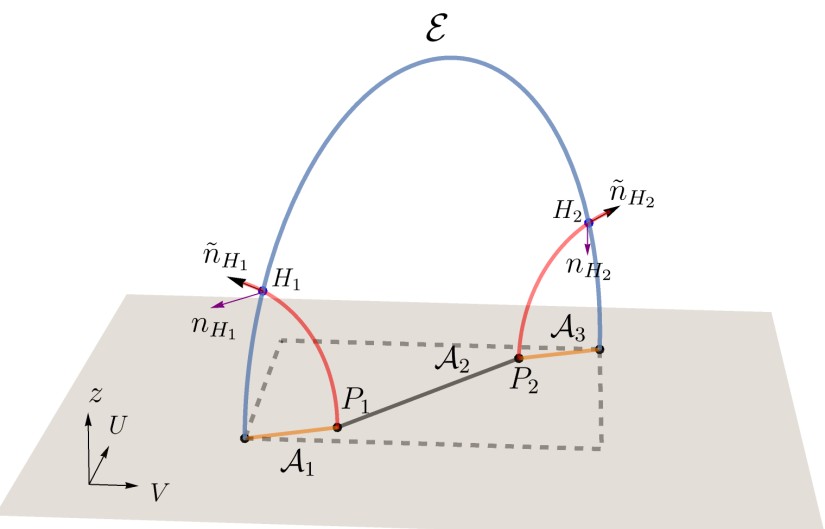

Figure 13: Illustration for the APEE of the subinterval $\mathcal{A}_2$ in the twist description. The two red curves are the geodesics normal to the RT surface $\mathcal{E}$. The two black arrows represent $\tilde{\mathbf{n}}|_{H_1}$ and $\tilde{\mathbf{n}}|_{H_2}$, respectively. The two purple arrows represent $\mathbf{n}|_{H_1}$ and $\mathbf{n}|_{H_2}$, respectively.

associated with $\mathcal{E}_i$ as the APEE $s_{\mathcal{A}}^a (C_i)$, since the configuration of the normal vectors $\mathbf{n}, \tilde{\mathbf{n}}$ on $\mathcal{E}_i$ is determined by the subcurve $\mathcal{C}_i$, as specified by (153), i.e.

$$s_{\mathcal{A}}^a (C_i) = \frac{1}{4\mu G} \int_{\mathcal{E}_i} d\tau \, \tilde{\mathbf{n}} \cdot \nabla \mathbf{n} = \frac{1}{4\mu G} \log \left( \frac{(\mathbf{q} - \tilde{\mathbf{q}})|_{H_2} \cdot \mathbf{n}|_{H_2}}{(\mathbf{q} - \tilde{\mathbf{q}})|_{H_1} \cdot \mathbf{n}|_{H_1}} \right), \tag{155}$$

where $H_1$ and $H_2$ are the partner points of the left and the right endpoints of $\mathcal{C}_i$ on $\mathcal{E}$, respectively. We find the above expression only depends on the endpoints of $\mathcal{C}_i$, which suggests us to define a function for the boundary point $P = (U_1, V_1)$ according to the following definition,

$$\bar{\lambda}(P) = \log((\mathbf{q} - \tilde{\mathbf{q}})|_H \cdot \mathbf{n}|_H) = \frac{1}{2} \log \left( \frac{(l_U + 2U_1)(l_V - 2V_1)}{(l_U - 2U_1)(l_V + 2V_1)} \right) = \hat{\tau}_{\widehat{H}}, \tag{156}$$

where $H$ and $\widehat{H}$ are the partner points of $P$ on $\mathcal{E}$ and on $\widehat{\mathcal{E}}$, as specified by (50) and (64), respectively. The reference normal frame $\{\mathbf{q}, \tilde{\mathbf{q}}\}$ is given by (104), and the normal vectors $\mathbf{n}, \tilde{\mathbf{n}}$ are given by (150) and (153). $\hat{\tau}_{\widehat{H}}$ is the length parameter of $\widehat{\mathcal{E}}$ of the point $\widehat{H}$, which is given by (32). For any subinterval $\mathcal{A}_i$ whose two endpoints are $P = (U_1, V_1)$ and $Q = (U_2, V_2)$, the APEE $s_{\mathcal{A}}^a (A_i)$ should equal to,

$$
\begin{aligned}
s_{\mathcal{A}}^a (A_i) &= \frac{1}{4\mu G} \int_{\mathcal{E}_i} d\tau \, \tilde{\mathbf{n}} \cdot \nabla \mathbf{n} = \frac{\bar{\lambda}(Q) - \bar{\lambda}(P)}{4\mu G} \\
&= \frac{1}{8\mu G} \log \left( \frac{(l_U - 2U_1)(l_V + 2V_1)(l_U + 2U_2)(l_V - 2V_2)}{(l_U + 2U_1)(l_V - 2V_1)(l_U - 2U_2)(l_V + 2V_2)} \right),
\end{aligned}
\tag{157}
$$

where $\mathcal{E}_i$ is the partner geodesic chord of $\mathcal{A}_i$ on $\mathcal{E}_i$, and the configuration of the normal vectors $\mathbf{n}, \tilde{\mathbf{n}}$ is determined by $\mathcal{A}_i$, see Fig.13 for an illustration. More importantly, the result aligns exactly with the APEE in the IRT description, based on the fine structure of the extended entanglement wedge, i.e. (78) divided by $\mu$.

Now we turn to the anomalous part $S_{\mathcal{A}}^a$ (154) with the configuration of the normal vectors $\mathbf{n}, \tilde{\mathbf{n}}$ are determined by a boundary curve $\mathcal{C}$. However, to explicitly calculate $S_{\mathcal{A}}^a$ (154), we

need to adopt the holographic regulation $z = \epsilon$, because the metric diverges at the asymptotic boundary. That is, we consider the initial and the final points of the integration in $S_{\mathcal{A}}^a$ to be located on the $z = \epsilon$ surface, which also implies that we simultaneously regulate the boundary curve $\mathcal{C}$, because the points on $\mathcal{C}$ and those on $\mathcal{E}$ are in one-to-one correspondence. Therefore, we can only focus on the regulation for the boundary curve $\mathcal{C}$. We denote the regularized curve as $\mathcal{C}^{\mathrm{reg}}$ by imposing the regulation for $\mathcal{C}$ at the following boundary points,

$$P_1 = (-l_U/2 + \epsilon_U, -l_V/2 + \epsilon_V), \qquad P_2 = (l_U/2 - \epsilon_U, l_V/2 - \epsilon_V), \tag{158}$$

which is equivalent to making a holographic regulation at the partner points $H_1, H_2$ (50) of $P_1, P_2$ on $\mathcal{E}$,

$$\rho_{H_1} = \rho_{H_2} = \frac{1}{2\epsilon_U \epsilon_V} \Leftrightarrow z_{H_1} = z_{H_2} = 2\sqrt{\epsilon_U \epsilon_V}. \tag{159}$$

And the anomalous part $S_{\mathcal{A}}^a$ (154) is given by the part associated with $\mathcal{C}^{\mathrm{reg}}$,

$$S_{\mathcal{A}}^a = \frac{1}{4\mu G} \int_{\mathcal{E}^{\mathrm{reg}}} d\tau \, \tilde{n} \cdot \nabla n = s_{\mathcal{A}}^a(\mathcal{C}_i) = \frac{1}{4\mu G} \log\left(\frac{l_U \epsilon_V}{l_V \epsilon_U}\right), \tag{160}$$

where $\mathcal{E}^{\mathrm{reg}}$ is the partner geodesic chord of $\mathcal{C}^{\mathrm{reg}}$ on $\mathcal{E}$, and we have used (157). And the result aligns exactly with that (113) in the IRT description as we expected. Furthermore, we can also discuss the asymptotic behavior (i.e. the boundary condition) of the normal vector $\mathbf{n}$. For example, the normal vector $\mathbf{n}|_{H_1}$ for the cutoff point $P_1$ (158) is given by (150),

$$\mathbf{n}|_{H_1} = -\frac{z_{H_1}}{2}\left(\sqrt{\frac{\epsilon_U}{\epsilon_V}}\partial_U - \sqrt{\frac{\epsilon_V}{\epsilon_U}}\partial_V\right). \tag{161}$$

One can easily verify that $\mathbf{n}|_{H_1}$ is proportional to the vector $\partial_t$ where $t$ is the temporal coordinate of the boundary CFT, only when a special regulation $\epsilon_U = \epsilon_V$ is taken.

# 7 Summary and discussions

In this work we have studied two relatively independent topics associated to the inner horizon of the Rindler $\widetilde{\mathrm{AdS}}_3$. First, we identify the pre-image of the inner horizon in the original $\mathrm{AdS}_3$, which coincides with the spacelike geodesic representing the real part of the holographic timelike entanglement entropy. We call it the inner RT surface, which is an extremal surface and is invariant under the bulk modular momentum flow. One can apply the bulk replica algorithms associated to the IRT surface which gives the IRT surface a physical interpretation as a von Neumann entropy that equals to the two-point function of the twist operators settled at the endpoints of a timelike interval. We also introduce the concept of the extended entanglement wedge by including the region between the RT and IRT surfaces. We identify a timelike geodesic at the boundary of the extended entanglement wedge which reproduces the imaginary part of the holographic timelike entanglement entropy. Nevertheless, the points on this timelike geodesic are not the fixed points of the modular momentum flow, which makes its role in the analog replica story [17] of the extended entanglement wedge unclear. We hope to revisit this point in the future. We introduce the fine structure of the entanglement wedge and extended entanglement wedge, via slicing them using the modular slices and the modular momentum slices respectively, which gives a partnership between points in the causal development and the points on the RT and IRT surfaces. The (timelike) partial entanglement entropy for subintervals in the causal development is then given by the length of the geodesics chord on the RT (IRT) surface, which consists of the partner points of the subinterval.

Second, in the duality between the TMG and the gravitational anomalous CFT$_2$, we reproduce the gravitational anomalous correction to the holographic entanglement entropy and the BPE (or reflected entropy) using the length of the geodesic chords on the IRT surface. More interestingly, the geodesic chord that captures the anomalous part of the BPE is also a saddle geodesic connecting different pieces of the IRT surface of the mixed state, which means it can also be understood as an EWCS anchored on the IRT surface, hence captures an optimized quantum information quantity. This is a natural generalization of the observation that, the gravitational anomalous correction to the black hole entropy in TMG is proportional to the area of the inner horizon. Previously the gravitational anomalous correction to the entanglement entropy and the BPE are described by measuring how much the RT surface [50] or the EWCS [51] are twisted after introducing a normal frame configuration on the RT surface. Nevertheless, certain prescription for the normal frame configuration on the RT surface is needed before we can measure the degree of twist, which lacks physical interpretation. We show that, our geometric picture using the geodesic chords on the IRT surfaces is equivalent to the twist description based on the normal frame configurations given in [50, 51].

In the sections 4, 5 and 6, when we say the CS correction to an entropy quantity, we mean an entropy quantity associated to the outer horizon, or the RT surface. One can also consider the CS correction to the entropy associated to the inner horizon, either by applying the world-line action formalism [50] (see [97] for details, and see [16] for a similar calculation), or the fine structure analysis introduced in this paper. On the contrary, this correction is proportional to the length of the outer horizon. Since we have associated the IRT surface as the geometric picture for the timelike entanglement entropy, this CS correction may also be taken as the CS correction to the timelike entanglement entropy. If we take the flat limit (where the AdS radius approaches infinity), this CS correction reproduces CS correction to the holographic entanglement entropy in the 3-dimensional flat holography [33, 35, 36, 98].[24] In particular, the authors in [97] applied the worldline action to evaluate the holographic entanglement entropy in 3d flat holography and obtained the same result as the field theory calculation [16, 36].

In [16, 52, 53], the geometric picture of the HEE in 3d flat holography is proposed to be the so-called *swing surface*, which consists of two null geodesics[25] (the so-called ropes), and a spacelike extremal surface connecting the two ropes (the so-called bench surface). Furthermore, in [52] it was pointed out that, the HEE calculated by the bench is indeed a PEE, which correspond to length of the bench, a geodesic chord on the infinite RT surface. At the same time, the ropes give another tool[26] to determine the point-to-point partnership between the points of the boundary interval and the points on the RT surface. Readers who are interested in the relationship between the swing surface and the fine structure analysis using the modular slices, should consult [52] for more details, as well as the related discussions in Appendix F.

Unlike the RT surface, the IRT surface extends deep inside the AdS bulk. One can further explore geometry reconstruction using the IRT surface. Also we can define the analog of the partial entanglement entropy threads [78, 80, 82] using the IRT surfaces, which may be helpful for understanding the geometric reconstruction for covariant or even timelike geometric quantities.

---

[24]The 3d flat holographic is the correspondence between the asymptotically flat gravity and a 2-dimensional field theory with BMS$_3$ symmetries which lives on the future null infinity, see [16, 97, 99, 100] for details. The gravity can also be the TMG, and in this case both of the two central charges in the dual BMS field theory are non-zero.

[25]These two null geodesics emanate from the two endpoints of the boundary interval and enters into the bulk along the $r$ direction, which is null, in the Bondi gauge.

[26]In this paper we used the geodesics $\gamma_P$ instead of the null lines. Actually, the null ropes are just the boundary of the modular slices, which can also be used to identify the point-to-point partnership between the points on the boundary interval and the points on the RT surface.

In the context of the AdS/CFT, it was recently pointed out that the twist along the RT surface is also an observable even in pure AdS$_3$ without gravitational anomaly [101]. From the perspective of operators, the twist commutes with the operator of the area of the RT surface. However, the physical meaning of the twist is still not well understood. In this paper, we find a close relationship between the twist along the RT surface and the IRT surface, hence the twist may have a physical interpretation associated with the timelike entanglement.

Recently, there are some new achievements on the study of the timelike entanglement, see for example [37, 38, 61, 102–104]. In [61] the authors defined a so-called timelike modular Hamiltonian, which is obtained by analytic continuation for the modular Hamiltonian, to reconstruct the bulk operators, and the timelike geodesic that represents the geometric picture of the imaginary part does not play a similar role in the reconstruction of the bulk operators. It is interesting to re-derive their results in our new context of the timelike entanglement, which does not depend on the analytic continuation. Also generalizing our discussion to higher dimensions or other holographic theories are interesting future directions.

## Acknowledgments

We thank Ashish Chandra, Debarshi Basu, Rob Myers and Yiwei Zhong for helpful discussions. QW thanks the "Holographic Duality and Models of Quantum Computation" conference hold at Bali island for helpful discussions with the participants during the conference.

**Funding information** This work is supported by NSFC Grant No.12447108. HZ is supported by SEU Innovation Capability Enhancement Plan for Doctoral Students (Grant No. CXJH_SEU 24137).

## A The modular Hamiltonian and the modular momentum

As we have mentioned in Sec.2.1 and Sec.2.2, the modular Hamiltonian and the modular momentum for the interval $\mathcal{A}$ (3) are the generators of translations for the Rindler temporal and spatial coordinates respectively, thus they can be expressed as the integrals of the stress tensor in the Rindler space,

$$H_{\text{mod}} = \int_{-\infty}^{\infty} d\tilde{x}\, T_{\tilde{t}\tilde{t}}\,, \qquad P_{\text{mod}} = \int_{-\infty}^{\infty} d\tilde{x}\, T_{\tilde{t}\tilde{x}}\,. \tag{A.1}$$

Using (23) and (12), we can map the modular Hamiltonian $H_{\text{mod}}$ and the modular momentum $P_{\text{mod}}$ to the original space,

$$
\begin{aligned}
H_{\text{mod}} &= \pi \int_{-\infty}^{\infty} \left( \frac{d\tilde{U}}{\beta_{\tilde{U}}} + \frac{d\tilde{V}}{\beta_{\tilde{V}}} \right) \left( \left( \frac{\partial U}{\partial \tilde{t}} \right)^2 T_{UU} + \left( \frac{\partial V}{\partial \tilde{t}} \right)^2 T_{VV} \right) \\
&= \int_{\mathcal{A}} dU \frac{l_U^2 - 4U^2}{8l_U} T_{UU} + \int_{\mathcal{A}} dV \frac{l_V^2 - 4V^2}{8l_V} T_{VV}\,, \\
P_{\text{mod}} &= \pi \int_{-\infty}^{\infty} \left( \frac{d\tilde{U}}{\beta_{\tilde{U}}} + \frac{d\tilde{V}}{\beta_{\tilde{V}}} \right) \left( \frac{\partial U}{\partial \tilde{t}} \frac{\partial U}{\partial \tilde{x}} T_{UU} + \frac{\partial V}{\partial \tilde{t}} \frac{\partial V}{\partial \tilde{x}} T_{VV} \right) \\
&= \int_{\mathcal{A}} dU \frac{l_U^2 - 4U^2}{8l_U} T_{UU} - \int_{\mathcal{A}} dV \frac{l_V^2 - 4V^2}{8l_V} T_{VV}\,.
\end{aligned}
\tag{A.2}
$$

# B  The IRT surface in the global AdS$_3$

The CFT$_2$ on the cylinder is dual to the global AdS$_3$,

$$ds^2 = -\left(1+r^2\right)dt^2 + \frac{1}{1+r^2}dr^2 + r^2 d\theta^2, \tag{B.1}$$

with the following identifications,

$$\theta \sim \theta + 2\pi, \qquad t \sim t + 2\pi. \tag{B.2}$$

Without loss of generality, we consider the static interval $A$ on the $t=0$ slice with the length $\ell_\theta$ ($\ell_\theta < \pi$). The corresponding Rindler transformation is,

$$\tilde{U} = \frac{1}{T_{\tilde{U}}}\text{arctanh}\left(\frac{\tan\left(\frac{\pi U}{L_U}\right)}{\tan\left(\frac{\pi l_U}{2L_U}\right)}\right), \qquad \tilde{V} = \frac{1}{T_{\tilde{V}}}\text{arctanh}\left(\frac{\tan\left(\frac{\pi V}{L_V}\right)}{\tan\left(\frac{\pi l_V}{2L_V}\right)}\right), \tag{B.3}$$

where $(U,V)$ are the light-cone coordinates and are defined as,

$$U = \frac{L_U}{2\pi}\left(\theta + t\right), \qquad V = \frac{L_V}{2\pi}\left(\theta - t\right), \tag{B.4}$$

and the parameters $l_U, l_V$ are given by

$$l_U = \frac{L_U}{2\pi}\ell_\theta, \qquad l_V = \frac{L_V}{2\pi}\ell_\theta. \tag{B.5}$$

Similarly, there are also two modular flows, one is the modular Hamiltonian flow $k_t^\alpha$ and the other one is the modular momentum flow $k_t^\beta$,

$$
\begin{aligned}
k_t^\alpha &= \frac{\pi}{T_{\tilde{U}}}\partial_{\tilde{U}} - \frac{\pi}{T_{\tilde{V}}}\partial_{\tilde{V}} \\
&= \pi\left(\cos t \cos\theta - \cos\left(\frac{\ell_\theta}{2}\right)\right)\csc\left(\frac{\ell_\theta}{4}\right)\sec\left(\frac{\ell_\theta}{4}\right)\partial_t - 2\pi\csc\left(\frac{\ell_\theta}{2}\right)\sin t \sin\theta\,\partial_\theta, \\
k_t^\beta &= \frac{\pi}{T_{\tilde{U}}}\partial_{\tilde{U}} + \frac{\pi}{T_{\tilde{V}}}\partial_{\tilde{V}} \\
&= -2\pi\csc\left(\frac{\ell_\theta}{2}\right)\sin t \sin\theta\,\partial_t\,\pi\left(\cos t \cos\theta - \cos\left(\frac{\ell_\theta}{2}\right)\right)\csc\left(\frac{\ell_\theta}{4}\right)\sec\left(\frac{\ell_\theta}{4}\right)\partial_\theta.
\end{aligned}
\tag{B.6}
$$

One can easily verify that the endpoints of $A$ are the fixed points of $k_t^\alpha$, and those of $\widehat{A}$ are the fixed points of $k_t^\beta$, respectively. Here, $\widehat{A}$ is the partner interval that connects the two tips of the causal development $\mathcal{D}_A$, which is a pure timelike interval with the length $\ell_\theta$ along the $\theta$ direction, see Fig.14. The bulk Rindler transformation also maps the original global AdS$_3$ (B.1) to the Rindler $\widetilde{\text{AdS}_3}$ (9), and the radial coordinate transformation is given by

$$
\begin{aligned}
\tilde{\rho} = \frac{T_{\tilde{U}}T_{\tilde{V}}}{4}\Big(&2 + \left(2+4r^2\right)\cos l_\theta + 4\left(1+r^2\right)\cos(2t) \\
&+ 8r\cos\theta\left(-2\sqrt{1+r^2}\cos(l_\theta/2)\cos t + r\cos\theta\right)\Big)\csc^2(l_\theta/2).
\end{aligned}
\tag{B.7}
$$

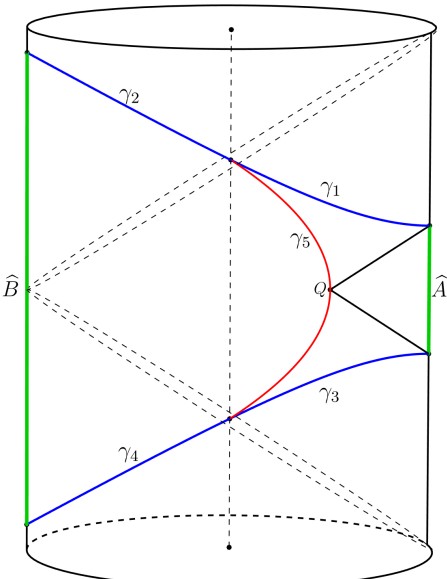

Figure 14: The surfaces enclosed by the dash curve represent the null infinities of the Poincaré patch. The blue curves $\gamma_1$ and $\gamma_3$ represent the IRT surface of $\widehat{A}$, while the blue curves $\gamma_2$ and $\gamma_4$ represent the IRT surface of $\widehat{B}$, where $B$ is the complement of $A$. The black line is the boundary of the entanglement wedge of $A$ on the $\theta = 0$ slice. The red curve $\gamma_5$ represents the boundary of the extended entanglement wedge on the $\theta = 0$ slice, which intersects with the RT surface $\mathcal{E}$ at the point $Q$.

According to the holographic dictionary, the bulk modular flows can be obtained by replacing the boundary global generators in the boundary modular flows with their bulk dual Killing vectors,

$$
\begin{aligned}
k_t^{\alpha,\text{bulk}} &= \frac{\pi \csc\left(\frac{l_\theta}{4}\right) \sec\left(\frac{l_\theta}{4}\right)}{\sqrt{1+r^2}} \left( r \cos t \cos \theta - \sqrt{1+r^2} \cos\left(\frac{l_\theta}{2}\right) \right) \partial_t \\
&\quad - \frac{2\pi}{r} \sqrt{1+r^2} \csc\left(\frac{l_\theta}{2}\right) \sin t \sin \theta \, \partial_\theta + 2\pi \sqrt{1+r^2} \cos \theta \csc\left(\frac{l_\theta}{2}\right) \sin t \, \partial_r \,, \\
k_t^{\beta,\text{bulk}} &= -\frac{2\pi r}{\sqrt{1+r^2}} \csc\left(\frac{l_\theta}{2}\right) \sin t \sin \theta \, \partial_t \\
&\quad + \frac{\pi \tan\left(\frac{l_\theta}{4}\right)}{r} \left( r - r \cot^2\left(\frac{l_\theta}{4}\right) + \sqrt{1+r^2} \cos t \cos \theta \csc^2\left(\frac{l_\theta}{4}\right) \right) \partial_\theta \\
&\quad + 2\pi \sqrt{1+r^2} \cos t \csc\left(\frac{l_\theta}{2}\right) \sin \theta \,.
\end{aligned}
\tag{B.8}
$$

As discussed in Sec.2, solving $k_t^{\alpha,\text{bulk}} = 0$ and $k_t^{\beta,\text{bulk}} = 0$ give us the RT surface $\mathcal{E}$ and the IRT surface $\widehat{\mathcal{E}}$, respectively,

$$
\mathcal{E} : r = \frac{\cot\left(\frac{\ell_\theta}{2}\right)}{\sqrt{\cos^2 \theta \csc^2\left(\frac{l_\theta}{2}\right) - \cot^2\left(\frac{l_\theta}{2}\right)}} \,, \qquad t = 0 \,,
\tag{B.9}
$$

$$\widehat{\mathcal{E}}: \begin{cases} \gamma_1 : t = -\arccos\left(\dfrac{r\cos(\ell_\theta)}{\sqrt{1+r^2}}\right), & \theta = 0, \\[3mm] \gamma_2 : t = -\arccos\left(-\dfrac{r\cos(\ell_\theta/2)}{\sqrt{1+r^2}}\right), & \theta = \pi, \\[3mm] \gamma_3 : t = \arccos\left(\dfrac{r\cos(\ell_\theta/2)}{\sqrt{1+r^2}}\right), & \theta = 0, \\[3mm] \gamma_4 : t = \arccos\left(-\dfrac{r\cos(\ell_\theta/2)}{\sqrt{1+r^2}}\right), & \theta = \pi, \end{cases}$$

where $\gamma_1$ and $\gamma_2$ smoothly join together at the null infinity of the Poincaré patch, as do $\gamma_3$ and $\gamma_4$. More interestingly, $\gamma_1$ intersects the asymptotic boundary at the upper endpoint of $\widehat{A}$, while $\gamma_2$ intersects the asymptotic boundary at the upper endpoint of $\widehat{B}$, where $B$ is the complement of $A$, see Fig.14. One can easily verify that the RT surface $\mathcal{E}$ and the IRT surface $\widehat{\mathcal{E}}$ are indeed mapped to the outer and the inner horizons of the Rindler $\widetilde{\mathrm{AdS}_3}$, respectively,

$$\tilde{\rho}|_{\mathcal{E}} = T_{\tilde{U}} T_{\tilde{V}}, \qquad \tilde{\rho}|_{\widehat{\mathcal{E}}} = -T_{\tilde{U}} T_{\tilde{V}}. \tag{B.10}$$

On the other hand, we can obtain a timelike geodesic representing the boundary of the intersection surface between the extended entanglement wedge $\widehat{\mathcal{W}}_A$ and the hypersurface $\theta = 0$ by following the derivation of Sec.3.2.2,

$$\gamma_5 : r = \cot\left(\frac{\ell_\theta}{2}\right)\cos\tau, \qquad t(\tau) = \arctan\left(\sin\left(\frac{\ell_\theta}{2}\right)\tan\tau\right), \tag{B.11}$$

which connects $\gamma_1$ and $\gamma_3$ at the null infinities of the Poincaré patch. This means $\gamma_2$ and $\gamma_4$ are not in $\widehat{\mathcal{W}}_A$. Therefore $\gamma_1$ and $\gamma_3$ should be referred to as the IRT surface of $\widehat{A}$, while $\gamma_2$ and $\gamma_4$ should be referred to as the IRT surface of $\widehat{B}$. And the length of the timelike geodesic $\gamma_5$ is also $\pi$.

## C  Spacelike and null geodesics in the Poincaré AdS$_3$

In this appendix, we list some useful properties of spacelike and null geodesics in the Poincaé AdS$_3$. We mainly focus on two types of spacelike geodesics in the Poincaré AdS$_3$. One is connected, which we call type I spacelike geodesic, which anchors on the boundary at two space-like separated boundary points, while type II spacelike geodesic is disconnected and anchors at two timelike separated boundary points. The type I spacelike geodesic is,

$$\rho(V) = \frac{2l_V}{l_U\left(l_V^2 - 4(V-V_0)^2\right)}, \qquad U(V) = \frac{l_U}{l_V}(V-V_0) + U_0 \qquad (l_U, l_V > 0), \tag{C.1}$$

which can be referred to as the RT surface of the boundary spatial interval $\mathcal{I}$

$$\mathcal{I}: (-l_U/2 + U_0, -l_V/2 + V_0) \to (l_U/2 + U_0, l_V/2 + V_0). \tag{C.2}$$

For the type I spacelike geodesic, which connects the two spacelike points $(U_1, V_1, \rho_1)$ and $(U_2, V_2, \rho_2)$, the parameters $(l_U, l_V, V_0, U_0)$ are

$$\begin{aligned} l_U &= \frac{X}{2(V_2 - V_1)\rho_1\rho_2}, \quad l_V = \frac{X}{2(U_2 - U_1)\rho_1\rho_2}, \\ V_0 &= \frac{\rho_2 + \rho_1(-1 + 2(U_1 - U_2)(V_1 + V_2)\rho_2)}{4(U_1 - U_2)\rho_1\rho_2}, \\ U_0 &= \frac{\rho_2 + \rho_1(-1 + 2(U_1 + U_2)(V_1 - V_2)\rho_2)}{4(V_1 - V_2)\rho_1\rho_2}, \end{aligned} \tag{C.3}$$

where $X$ is,

$$
\begin{aligned}
X &= \sqrt{\rho_1^2 + 2\rho_1\rho_2(\rho_1 Y - 1) + (\rho_2 + \rho_1\rho_2 Y)^2}, \\
Y &= 2(U_1 - U_2)(V_1 - V_2).
\end{aligned}
\tag{C.4}
$$

Hence the proper distance between $(U_1, V_1, \rho_1)$ and $(U_2, V_2, \rho_2)$ is,

$$
L(U_1, V_1, \rho_1, U_2, V_2, \rho_2) = \frac{1}{2}\log\left(\frac{(X + \rho_1 - \rho_2 + Y\rho_1\rho_2)(X + \rho_2 - \rho_1 + Y\rho_1\rho_2)}{(X + \rho_1 - \rho_2 - Y\rho_1\rho_2)(X + \rho_2 - \rho_1 - Y\rho_1\rho_2)}\right).
\tag{C.5}
$$

Similarly, the equation of the type II spacelike geodesic is,

$$
\rho(V) = -\frac{2l_V}{l_U\left(l_V^2 - 4(V - V_0)^2\right)}, \qquad U(V) = -\frac{l_U}{l_V}(V - V_0) + U_0 \qquad (l_U, l_V > 0),
\tag{C.6}
$$

which has two connected sectors

$$
\text{Left sector: } V < V_0 - \frac{l_V}{2}, \qquad \text{Right sector: } V > V_0 + \frac{l_V}{2},
\tag{C.7}
$$

and can be also referred to as the IRT surface of the interval $\mathcal{I}$ (C.2). The partner interval $\widehat{\mathcal{I}}$ is the timelike interval connecting the two tips of the causal development of $\mathcal{I}$, i.e.

$$
\widehat{\mathcal{I}}: (-l_U/2 + U_0, l_V/2 + V_0) \to (l_U/2 + U_0, -l_V/2 + V_0).
\tag{C.8}
$$

For the type II spacelike geodesic, which connects the two spacelike points $(U_1, V_1, \rho_1)$ and $(U_2, V_2, \rho_2)$, the parameters $(l_U, l_V, V_0, U_0)$ are,

$$
\begin{aligned}
l_U &= \frac{X}{2(V_2 - V_1)\rho_1\rho_2}, \quad l_V = \frac{X}{2(U_1 - U_2)\rho_1\rho_2}, \\
V_0 &= \frac{\rho_2 + \rho_1(-1 + 2(U_1 - U_2)(V_1 + V_2)\rho_2)}{4(U_1 - U_2)\rho_1\rho_2}, \\
U_0 &= \frac{\rho_2 + \rho_1(-1 + 2(U_1 + U_2)(V_1 - V_2)\rho_2)}{4(V_1 - V_2)\rho_1\rho_2},
\end{aligned}
\tag{C.9}
$$

where $X$ is also given by (C.4). The proper distance between $(U_1, V_1, \rho_1)$ and $(U_2, V_2, \rho_2)$ is also given by (C.5) when they are on the same sector of the type II spacelike geodesic (C.6). When they are on the different sectors, for example, $(U_1, V_1, \rho_1)$ is on the Left sector, $(U_2, V_2, \rho_2)$ is on the Right sector, we define the proper distance between them to be,

$$
\begin{aligned}
L(U_1, V_1, \rho_1, U_2, V_2, \rho_2) &= \int_{-\infty}^{V_1} \sqrt{ds^2} + \int_{V_2}^{\infty} \sqrt{ds^2} \\
&= -\frac{1}{2}\log\left(\frac{(X + \rho_1 - \rho_2 + Y\rho_1\rho_2)(X + \rho_2 - \rho_1 + Y\rho_1\rho_2)}{(X + \rho_1 - \rho_2 - Y\rho_1\rho_2)(X + \rho_2 - \rho_1 - Y\rho_1\rho_2)}\right),
\end{aligned}
\tag{C.10}
$$

where $ds^2$ is the induced line element on (C.6) and $X, Y$ are (C.4), and the difference between (C.10) with (C.5) is only up to a overall sign.

The null geodesic, which intersects the asymptotic boundary at the boundary point $(U_1, V_1)$, is given by

$$
\rho(V) = \frac{C^2}{(V_1 - V)^2}, \qquad U(V) = U_1 + \frac{V_1 - V}{2C^2},
\tag{C.11}
$$

where $C$ is an arbitrary constant. The null hypersurface introduced by the boundary point $(U_1, V_1)$ is,

$$
\rho = \frac{1}{2(U - U_1)(V_1 - V)}.
\tag{C.12}
$$

In particular, the null geodesic (C.11) with the coefficient $C = 1/\sqrt{2}$ is,

$$
\rho(V) = \frac{1}{2(V_1 - V)^2}, \qquad U(V) = U_1 + V_1 - V.
\tag{C.13}
$$

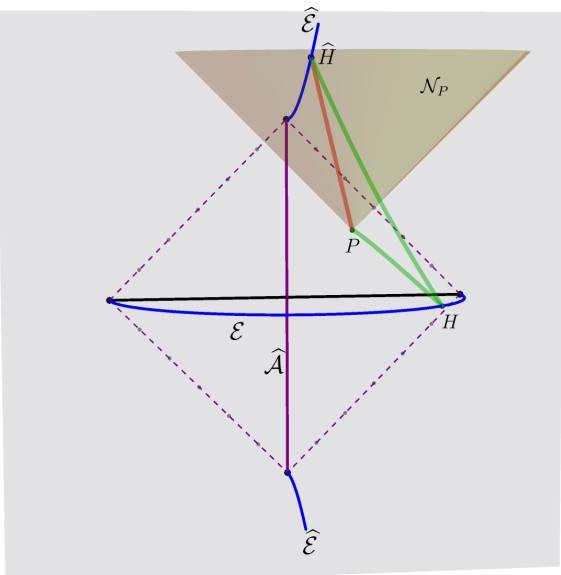

Figure 15: The yellow surface represents the null hypersurface $\mathcal{N}_P$ introduced by $P$. The two green curves represent the saddle geodesics $\widehat{\gamma}_P^1$ and $\widehat{\gamma}_P^2$, respectively. The partner point $\widehat{H}$ is the intersection point between $\mathcal{N}_P$ and $\widehat{\mathcal{E}}$. And the red curve presents the null geodesic connecting $P$ and $\widehat{H}$.

## D  A trick for locating the partner point $\widehat{H}$ on $\widehat{\mathcal{E}}$

Consider a boundary point $P = (U_1, V_1)$ in $\mathcal{D}_{\mathcal{A}}$, and the null hypersurface introduced by $P$ via (C.12), which we denote as $\mathcal{N}_P$. There is only one intersection point $\bar{H}$ between $\mathcal{N}_P$ and the IRT surface $\widehat{\mathcal{E}}$ (31), which is given by

$$\bar{H} = \mathcal{N}_P \cap \widehat{\mathcal{E}}: \ \ V_{\bar{H}} = -\frac{l_V\left(l_U l_V - 4U_1 V_1\right)}{4\left(l_V U_1 - l_U V_1\right)}, \tag{D.1}$$

which is exactly the same as the partner point $\widehat{H}$ (64) of $P$ on the IRT surface $\widehat{\mathcal{E}}$. In other words, the partner point $\widehat{H}$ of $P$ is precisely the intersection point between the null hypersurface $\mathcal{N}_P$ and the IRT surface $\widehat{\mathcal{E}}$, see Fig.15 for an illustration.

## E  Replica trick for the CFT$_2$ with gravitational anomaly

The generators of the CFT$_2$ on the plane are given by

$$L_n = U^{n+1}\partial_U, \qquad \bar{L}_n = V^{n+1}\partial_V, \tag{E.1}$$

where $L_{\pm}, L_0$ and $\bar{L}_{\pm}, \bar{L}_0$ are the only global generators. The charges $\mathcal{L}_n, \bar{\mathcal{L}}_n$ associated with these generators form the Virasoro algebra which is the central extension of the Witt algebra,

$$\begin{aligned}
\left[\mathcal{L}_n, \mathcal{L}_m\right] &= (n-m)\mathcal{L}_{n+m} + \frac{c_L}{12}n\left(n^2-1\right)\delta_{n+m,0}, \\
\left[\bar{\mathcal{L}}_n, \bar{\mathcal{L}}_m\right] &= (n-m)\bar{\mathcal{L}}_{n+m} + \frac{c_R}{12}n\left(n^2-1\right)\delta_{n+m,0},
\end{aligned} \tag{E.2}$$

where $c_L$ and $c_R$ are the left-moving central charge and the right-moving central charge, respectively. For the interval $\mathcal{A}$ (3), the entanglement entropy $S_{\mathcal{A}}$ is defined as the von Neumann

entropy $S_{\mathcal{A}} = -\mathrm{Tr}\rho_{\mathcal{A}}\log\rho_{\mathcal{A}}$ of the reduced density matrix $\rho_{\mathcal{A}}$. To calculate the entanglement entropy, we can first consider the $n$-order Rényi entropy $S_{\mathcal{A}}^{(n)} = \frac{1}{1-n}\log\mathrm{Tr}\rho_{\mathcal{A}}^n$, and then take the limit $n \to 1$. The replica method [105, 106] proposes the partition function $\mathcal{Z}_n = \mathrm{Tr}\rho_{\mathcal{A}}^n$ of the $n$-order replica manifold can be calculated by the two-point function of the twist operators, namely

$$\mathcal{Z}_n = \mathrm{Tr}\rho_{\mathcal{A}}^n = \left\langle \Phi_n(a)\bar{\Phi}_n(b)\right\rangle = \left(\frac{\epsilon_U}{l_U}\right)^{2h_L}\left(\frac{\epsilon_V}{l_V}\right)^{2h_R}, \tag{E.3}$$

where $a$ and $b$ are the left and the right endpoints of the interval $\mathcal{A}$, respectively. The cutoffs $\epsilon_U, \epsilon_V$ are introduced to make the action $\mathcal{Z}_n$ dimensionless and ensure the entanglement entropy to vanish in the limit $l_U \to \epsilon_U, l_V \to \epsilon_V$. The conformal dimensions of twist operator for the left-moving sector and the right-moving sector are $h_L = \frac{c_L}{24}\left(n - \frac{1}{n}\right)$ and $h_R = \frac{c_R}{24}\left(n - \frac{1}{n}\right)$, which can be determined by the conformal Ward identities. Note that for the $\mathrm{CFT}_2$ with the gravitational anomaly ($c_L \neq c_R$), the twist operator possesses non-zero spin $s_n$ [50],

$$\Delta_n = h_L + h_R = \frac{c_L + c_R}{24}\left(n - \frac{1}{n}\right), \qquad s_n = h_L - h_R = \frac{c_L - c_R}{24}\left(n - \frac{1}{n}\right), \tag{E.4}$$

where $\Delta_n$ is the scaling dimension of the twist operator. Furthermore, the $n$-order Rényi entropy is given by

$$S_{\mathcal{A}}^{(n)} = \frac{n+1}{12n}\left(c_L\log\left(\frac{l_U}{\epsilon_U}\right) + c_R\log\left(\frac{l_V}{\epsilon_V}\right)\right). \tag{E.5}$$

Accordingly, the entanglement entropy $S_{\mathcal{A}}$ is,

$$S_{\mathcal{A}} = \frac{c_L}{6}\log\left(\frac{l_U}{\epsilon_U}\right) + \frac{c_R}{6}\log\left(\frac{l_V}{\epsilon_V}\right). \tag{E.6}$$

# F The regulated geodesic chord on the inner RT surface and its flat limit

As discussed in Sec. 3.2.2, the regulated interval $\mathcal{A}^{\mathrm{reg}}$ corresponds to a geodesic chord on the IRT surface $\widehat{\mathcal{E}}$. More explicitly, the partner points on $\widehat{\mathcal{E}}$ of the endpoints of $\mathcal{A}^{\mathrm{reg}}$ are denoted as the cutoff points $Q_1$ and $Q_2$, respectively, which are given by (64),

$$\begin{aligned} V_{Q_1} &= \frac{l_V\left(l_V\epsilon_U + (l_U - 2\epsilon_U)\epsilon_V\right)}{2l_U\epsilon_V - 2l_V\epsilon_U} \sim \frac{l_V\left(l_V\epsilon_U + l_U\epsilon_V\right)}{2l_U\epsilon_V - 2l_V\epsilon_U}, \\ V_{Q_2} &= -\frac{l_V\left(l_V\epsilon_U + (l_U - 2\epsilon_U)\epsilon_V\right)}{2l_U\epsilon_V - 2l_V\epsilon_U} \sim -\frac{l_V\left(l_V\epsilon_U + l_U\epsilon_V\right)}{2l_U\epsilon_V - 2l_V\epsilon_U}, \end{aligned} \tag{F.1}$$

where $V_{Q_1}(V_{Q_2})$ represents the $V$-coordinate of $Q_1(Q_2)$, and "$\sim$" denotes that we perform a Taylor expansion for the infinitesimal constants $\epsilon_U$ and $\epsilon_V$. In particular, if we choose two equal cutoffs, i.e. $\epsilon_U = \epsilon_V = \epsilon$, the corresponding cutoff points $Q_1, Q_2$ are,

$$V_{Q_1} = \frac{l_V\left(l_V + l_U\right)}{2\left(l_U - l_V\right)}, \qquad V_{Q_2} = -\frac{l_V\left(l_V + l_U\right)}{2\left(l_U - l_V\right)}. \tag{F.2}$$

We denote the regulated part of $\widehat{\mathcal{E}}$ as $\gamma$, which extends from $Q_1$ to $V = \infty$, and then from $V = -\infty$ to $Q_2$,

$$\gamma = \widehat{\mathcal{E}}^{\mathrm{reg}}: \quad \rho = -\frac{2l_V}{l_U\left(l_V^2 - 4V^2\right)}, \qquad U = -\frac{l_U}{l_V}V \qquad \left(V < V_{Q_1} \text{ or } V > V_{Q_2}\right), \tag{F.3}$$

see Fig.16 for an illustration. Therefore, $\gamma$ represents the partner geodesic chord on $\widehat{\mathcal{E}}$ corresponds to $S_{\mathcal{A}}^a$ (96) with the cutoffs being $\epsilon_U = \epsilon_V = \epsilon$. However, the IRT surface $\widehat{\mathcal{E}}$ is not homologous to $\mathcal{A}$, such that we can introduce two auxiliary null geodesics $\gamma_-, \gamma_+$ emanating from the endpoints of $\mathcal{A}$, which are the bulk modular momentum flow lines on $\mathcal{M}_\pm$, to make the composite surface $\gamma_{\mathcal{A}} = \gamma \cup \gamma_- \cup \gamma_+$ homologous to $\mathcal{A}$, similarly to [16]. We emphasize that $\gamma_\pm$ are not the fixed points of $k_t^{\beta,\text{bulk}}$, hence do not represent the bulk replica symmetry. More explicitly, $\gamma_\pm$ are the bulk modular momentum flow lines connecting $\partial_\pm \mathcal{A} = \pm(l_U/2, l_V/2)$ with $Q_1(Q_2)$, which are given by (83)

$$\gamma_+ = \widehat{\mathcal{L}}_{Q_2}^+ : \rho(V) = \frac{2}{(l_V - 2V)^2}, \qquad U(V) = \frac{l_U + l_V}{2} - V,$$

$$\gamma_- = \widehat{\mathcal{L}}_{Q_1}^- : \rho(V) = \frac{2}{(l_V + 2V)^2}, \qquad U(V) = -\frac{l_U + l_V}{2} - V. \tag{F.4}$$

The anomalous part $S_{\mathcal{A}}^a$ (96) can be given by the total length of $\gamma_{\mathcal{A}}$

$$\begin{aligned} S_{\mathcal{A}}^a &= \frac{\text{Length}(\gamma_{\mathcal{A}})}{4\mu G} = \frac{\text{Length}(\gamma)}{4\mu G} \\ &= \frac{\widehat{\tau}_{Q_2} - \widehat{\tau}_{Q_1}}{4\mu G} = \frac{1}{4\mu G} \log\left(\frac{l_U}{l_V}\right), \end{aligned} \tag{F.5}$$

where $\widehat{\tau}$ is the length parameter of the IRT surface $\widehat{\mathcal{E}}$. One can see this is very similar to the case in the flat holography. In fact, this is reasonable. The authors in [16] calculated the holographic entanglement entropy by taking the flat limit ($\ell \to \infty$) for the Rindler method. In the flat limit, the outer horizon of Rindler $\widetilde{\text{AdS}_3}$ is pushed to infinity, thus leaving only the inner horizon in the bulk. More interestingly, they find that the flat limit of the area of the inner horizon exactly matches the holographic entanglement entropy in the flat holography. This implies that the holographic entanglement entropy in the flat holography can, to some extent, be referred to as originating from the inner horizon of the Rindler $\widetilde{\text{AdS}_3}$, rather than the outer horizon. Now we show that the geometric picture of the holographic entanglement entropy in the flat holography (including the null geodesics), can also be derived by taking the flat limit of our discussions here.[27]

The infinite-dimensional Bondi-Metzner-Sachs (BMS$_3$) group [108–112] is the asymptotic symmetry of three-dimensional flat spacetime at the null infinity, hence the Flat/BMS$_3$ correspondence which we called the flat holography was proposed in [32, 33]. The holographic entanglement entropy, and its geometric picture are given in [16] via the Rindler method

$$S_{\mathcal{A}}^{\text{BMS}} = \frac{\text{Length}\left(\gamma^{\text{BMS}}\right)}{4G}, \tag{F.6}$$

where the superscript "BMS" is used to distinguish it from the AdS/CFT, and $\gamma^{\text{BMS}}$ is the fixed points of the bulk modular flow $k_t^{\text{BMS,bulk}}$ (or bulk replica symmetry). The spacelike geodesic $\gamma$ is connected to the endpoints of the boundary interval $\mathcal{A}^{\text{BMS}}$ on null infinity by two null geodesics $\gamma_\pm^{\text{BMS}}$. The null geodesics $\gamma_\pm^{\text{BMS}}$ are the orbits of the endpoints $\partial \mathcal{A}^{\text{BMS}}$ under the bulk modular flow $k_t^{\text{BMS,bulk}}$ and they are two $r$ coordinate lines, see (F.11) later. Here we only focus on the null-orbifold whose metric is

$$ds^2 = -2du\,dr + r^2 d\phi^2, \tag{F.7}$$

which duals to the zero temperature BMSFT on the plane. We consider a boundary interval $\mathcal{A}^{\text{BMS}}$

$$\mathcal{A}^{\text{BMS}} : \left(-l_u/2, -l_\phi/2\right) \to \left(l_u/2, l_\phi/2\right), \tag{F.8}$$

---

[27]See also [107] for earlier discussions.

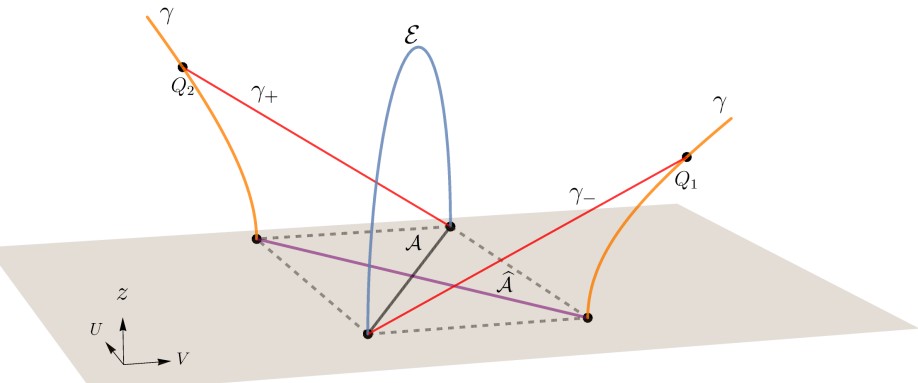

Figure 16: The black and the purple lines represent the interval $\mathcal{A}$ and its partner interval $\widehat{\mathcal{A}}$, respectively. The blue line is the RT surface $\mathcal{E}$. The orange line is the IRT surface $\widehat{\mathcal{E}}$. The red lines $\gamma_+$ and $\gamma_-$ are two null geodesics emanating from the endpoints of $\mathcal{A}$, which are introduced to make the composite surface $\gamma_{\mathcal{A}} = \gamma_+ \cup \gamma_- \cup \gamma$ homologous to $\mathcal{A}$. They intersect $\widehat{\mathcal{E}}$ at the cutoff points $Q_1$ and $Q_2$, and $\gamma$ is the regulated part of $\widehat{\mathcal{E}}$.

and the corresponding modular flows $k_t^{\text{BMS}}$ and $k_t^{\text{BMS,bulk}}$ are [16]

$$k_t^{\text{BMS}} = -\frac{\pi}{2l_\phi}\left(\left(l_\phi^2 - 4\phi^2\right)\partial_\phi + \left(l_u l_\phi + 4\frac{l_u}{l_\phi}\phi^2 - 8u\phi\right)\partial_u\right),$$

$$k_t^{\text{BMS,bulk}} = -\frac{\pi}{2l_\phi}\left(\left(l_\phi^2 - 4\phi^2 + \frac{8\left(l_\phi u - l_u \phi\right)}{l_\phi r}\right)\partial_\phi + \left(l_u l_\phi + 4\frac{l_u}{l_\phi}\phi^2 - 8u\phi\right)\partial_u\right. \tag{F.9}$$
$$\left. + \left(\frac{8l_u}{l_\phi} + 8r\phi\right)\partial_r\right),$$

the spacelike geodesic $\gamma^{\text{BMS}}$ is just the fixed points of $k_t^{\text{BMS,bulk}}$, i.e.

$$k_t^{\text{BMS,bulk}}|_{\gamma^{\text{BMS}}} = 0, \tag{F.10}$$

and $\gamma_\pm^{\text{BMS}}$ are two $r$ coordinate lines passing the endpoints of $\mathcal{A}^{\text{BMS}}$,

$$\gamma_+^{\text{BMS}} : u = \frac{l_u}{2}, \quad \phi = \frac{l_\phi}{2}, \qquad \gamma_-^{\text{BMS}} : u = -\frac{l_u}{2}, \quad \phi = -\frac{l_\phi}{2}. \tag{F.11}$$

Now we return to consider the flat limit of the geometric picture of the anomalous part $S_{\mathcal{A}}^a$. We recover the AdS radius $\ell$ such that the metric of Poincaré AdS$_3$ is,

$$ds^2 = \ell^2\left(2\rho dU dV + \frac{d\rho^2}{4\rho^2}\right), \tag{F.12}$$

which is the solution of the Einstein-Hilbert action plus the cosmological constant term,

$$S = \frac{1}{16\pi G}\int d^3x \sqrt{-g}\left(R - 2\Lambda\right), \tag{F.13}$$

where the cosmological constant $\Lambda = -1/\ell^2$. The cosmological constant $\Lambda$ will vanish in the limit $\ell \to \infty$, which we called the flat limit of the AdS/CFT. When the cosmological constant

vanishes, there is only the Einstein-Hilbert term, whose solution is just the flat spacetime. In order to consider the flat limit properly, we rewritten the metric (F.12) in the BMS gauge [113],

$$ds^2 = r^2 \left( d\phi^2 - \frac{du^2}{\ell^2} \right) - 2 du dr \,, \tag{F.14}$$

by the following coordinate transformation,

$$U = \frac{1}{2} \left( \ell\phi + u - \frac{\ell^2}{r} \right) , \qquad V = \frac{1}{2} \left( \ell\phi - u + \frac{\ell^2}{r} \right) , \qquad \rho = \frac{2r^2}{\ell^4} \,. \tag{F.15}$$

One can easily verify that the Poincaré-BMS metric (F.14) will reduce to the null-orbifold (F.7) in the flat limit. According to the coordinate transformation (F.15), the parameters $l_U, l_V$ can be written in terms of $l_u, l_\phi$,

$$l_U = \frac{\ell l_\phi + l_u}{2} , \qquad l_V = \frac{\ell l_\phi - l_u}{2} \,. \tag{F.16}$$

According to the coordinate transformation (F.15) and the parameters (F.16), we can rewrite the modular momentum flow vectors $k_t^\beta$ (28) and $k_t^{\beta,\text{bulk}}$ (26) in the Poincaré-BMS coordinate $(u, \phi, r)$,

$$k_t^\beta = \frac{\pi}{2\left(l_u^2 - \ell^2 l_\phi^2\right)} \left( \left( l_u^3 + 8 l_\phi u \ell^2 \phi - l_u \left( 4u^2 + \ell^2 \left( l_\phi^2 + 4\phi^2 \right) \right) \right) \partial_u \right.$$
$$\left. + \left( l_u^2 l_\phi - l_\phi^3 \ell^2 - 8 l_u u \phi + 4 l_\phi \left( u^2 + \ell^2 \phi^2 \right) \right) \partial_\phi \right) ,$$

$$k_t^{\beta,\text{bulk}} = \frac{\pi}{2\left(l_u^2 - \ell^2 l_\phi^2\right)} \left( \left( l_u^3 - l_u \left( \ell^2 \left( l_\phi^2 + 4\phi^2 \right) + 4u^2 \right) + 8 \ell^2 l_\phi u \phi \right) \partial_u \right.$$
$$+ \left( l_u^2 l_\phi r + 8 l_u \phi \left( \ell^2 - ru \right) + l_\phi \left( -\ell^2 r \left( l_\phi^2 - 4\phi^2 \right) + 4 ru^2 - 8 \ell^2 u \right) \right) \partial_\phi$$
$$\left. - 8 \left( \left( l_u \left( \ell^2 - ru \right) + \ell^2 l_\phi r \phi \right) \right) \partial_r \right) .$$

After taking the flat limit, the only difference between $k_t^\beta, k_t^{\beta,\text{bulk}}$ and $k_t^{\text{BMS}}, k_t^{\text{BMS,bulk}}$ is an overall coefficient negative sign, which does not change the physical essence. Therefore, in the flat limit, the spacelike geodesic $\gamma$, which is the set of the fixed points of $k_t^{\beta,\text{bulk}}$, will reduce to $\gamma^{\text{BMS}}$, the fixed point of $k_t^{\text{BMS,bulk}}$. Similarly, we can also rewrite the null geodesics $\gamma_\pm$ (F.4) in the Poincaré-BMS coordinate $(u, \phi, r)$,

$$\gamma_+ : u = \frac{l_u}{2}, \quad \phi = \frac{l_\phi}{2}, \qquad \gamma_- : u = -\frac{l_u}{2}, \quad \phi = -\frac{l_\phi}{2}, \tag{F.17}$$

which exactly match the null geodesics $\gamma_\pm^{\text{BMS}}$ (F.11) even without taking the flat limit. Therefore, the geometric picture of the holographic entanglement entropy in the flat holography can be obtained by taking the flat limit for the geometric picture of the anomalous part $S_{\mathcal{A}}^a$ in the AdS/CFT with gravitational anomaly. Furthermore, for the mixed state correlation, the EWCS in the flat holography (including the geometric picture) given in [114] can be also derived by taking the flat limit of the IEWCS $\widehat{\Sigma}_{AB}$, which is specified by (145) and (146).

Interestingly, all the null geodesics (C.13) are the $r$ coordinate lines even without taking the flat limit. More explicitly, the parameters $(U_1, V_1)$ which characterize the null geodesics (C.13) can be rewritten in terms of the parameters in the Poincaré-BMS coordinate according to the coordinate transformation (F.15),

$$U_1 = \frac{1}{2} \left( \ell\phi_1 + u_1 \right) , \qquad V_1 = \frac{1}{2} \left( \ell\phi_1 - u_1 \right) . \tag{F.18}$$

According to (F.15), the null geodesics (C.13) can be rewritten in the Poincaré-BMS coordinate,

$$u = u_1, \qquad \phi = \phi_1, \tag{F.19}$$

which are the $r$ coordinate lines in the BMS gauge.

# G  The equality between the two sides of (125)

In this appendix, we demonstrate the equality between the two sides of equation (125) by showing that the right-hand side of (125) indeed evaluates the APEE $s^a_{A'_1 AA'_2}(A)$, and thus equals to the left-hand side. Consider the configuration of $A'_1 AA'_2$ described by (128) and (136). The RT surfaces $\mathcal{E}_1$ and $\mathcal{E}_2$ can be also understood as the saddle spacelike geodesics $\widehat{\gamma}^1_{P_1}$ and $\widehat{\gamma}^1_{P_2}$ corresponding to the left and the right endpoints of $A$ for the interval $A'_1 AA'_2$, where $P_1$ and $P_2$ are the endpoints of $A$. This is because $\mathcal{E}_1$ and $\mathcal{E}_2$ are both normal to the EWCS $\Sigma_{AB}$, as shown in Fig.10. Therefore, the endpoints $H_1$ and $H_2$ (described by (132)) of $\Sigma_{AB}$ are the partner points of the left and the right endpoints of $A$ on the RT surface $\mathcal{E}_{A'_1 AA'_2}$, respectively. For (125), the authors in [51] chose the following boundary condition of the spacelike normal vectors $\tilde{\mathbf{n}}_i$ and $\tilde{\mathbf{n}}_f$ for the EWCS $\Sigma_{AB}$,

$$\tilde{\mathbf{n}}_i = v_{\widehat{\gamma}^1_{P_1}}|_{H_1}, \qquad \tilde{\mathbf{n}}_f = v_{\widehat{\gamma}^1_{P_2}}|_{H_2}, \tag{G.1}$$

where $v_{\widehat{\gamma}^1_{P_1}}|_{H_1}$ and $v_{\widehat{\gamma}^1_{P_2}}|_{H_2}$ represent the normalized tangent vectors of $\mathcal{E}_1$ and $\mathcal{E}_2$ at the endpoints $H_1$ and $H_2$ of $\Sigma_{AB}$, respectively. According to the discussion in Sec.6, the above equation also implies the boundary condition of the timelike normal vectors $\mathbf{n}_i$ and $\mathbf{n}_f$,

$$\mathbf{n}_i = v_{\widehat{\gamma}^2_{P_1}}|_{H_1}, \qquad \mathbf{n}_f = v_{\widehat{\gamma}^2_{P_2}}|_{H_2}, \tag{G.2}$$

where $\widehat{\gamma}^2_{P_1}$ and $\widehat{\gamma}^2_{P_2}$ are the saddle timelike geodesics corresponding to the two endpoints of $A$ for the interval $A'_1 AA'_2$. Therefore, $\mathbf{n}_i$ and $\mathbf{n}_f$ are given by (150) up to a translation $(U_1, V_1) \to (U_1 - U_0, V_1 - V_0)$ and a variable replacement $l_U, l_V \to \tilde{l}_U, \tilde{l}_V$, where $(\tilde{l}_U, \tilde{l}_V, U_0, V_0)$ are specified by (135). In fact, this choice can also ensure that the term inside the logarithm in (125) is positive, see (156). According to (156), the right-hand side of (125) is given by

$$\text{RHS} = \frac{\widehat{\tau}_{\widehat{H}_2} - \widehat{\tau}_{\widehat{H}_1}}{4\mu G} = s^a_{A_1 AA'_2}(A), \tag{G.3}$$

where $\widehat{H}_1$ and $\widehat{H}_2$ are the partner points of the endpoints $P_1$ and $P_2$ of $A$, respectively. Therefore, the two sides of (125) are equal.

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
