# Peer review of "Timelike and gravitational anomalous entanglement from the inner horizon"

_SciPost Physics, doi:SciPost Phys. 18, 204 (2025)_

## Round 1 · Referee Report · Anonymous (Referee 1) · 2025-3-22

Strengths

  1. Novel Insights: The manuscript offers a fresh perspective on holographic entanglement entropy by introducing the inner RT surface (IRT surface) and investigating its role in gravitational anomalies.

2.Clear Structure: The paper is well-organized, with topics clearly separated and ideas logically progressing. The modular slices, PEE, and gravitational anomalous corrections sections are particularly detailed and accessible.

  1. Connection to Previous Work: The authors effectively build upon previous studies in holographic entanglement, particularly regarding the EWCS and holographic entanglement entropy in TMG. The integration of gravitational anomalies into the entanglement framework is valuable.

Weaknesses

  1. The manuscript occasionally presents complex mathematical derivations without sufficient explanation, which could make it difficult for readers unfamiliar with the specific methods employed (e.g., worldline actions, regulated geodesic chords, etc.).

Recommendation: Provide more detailed explanations for key steps in the mathematical derivations, especially in the sections involving the IRT surface and the replica trick.

  1. Discussing the mixed state correlation and its connection to the IRT surface is interesting but could benefit from a broader generalization. Including a more comprehensive analysis of how these results extend to other non-AdS geometries or more general holographic setups would be helpful.

Recommendation: Consider discussing the potential generalization of the IRT surface and its relevance to more generic holographic models.

  1. The physical interpretation of the inner RT surface (IRT surface) and its connection to timelike entanglement entropy is not immediately intuitive. While the authors explain the concept clearly in technical terms, a more accessible explanation for a broader audience might be helpful.

Recommendation: A more intuitive explanation of the significance of the IRT surface, perhaps with more physical analogies, would make the results more accessible.

Specific Questions/Concerns: What is the physical motivation for choosing the inner RT surface over other possible surfaces, particularly in the context of gravitational anomalies?

Could you provide a more detailed explanation of how the timelike entanglement entropy is related to the imaginary part of the holographic entanglement entropy and the role of the inner RT surface in this context?

The paper mentions correcting the holographic entanglement entropy due to the Chern-Simons term. How general is this result? Does it hold for all gravitational anomalies, or is it specific to certain anomalies?

How does the proposed framework for mixed-state correlations (inner EWCS) compare with other methods for dealing with mixed-state holography, such as the swing surface or the island prescription?

Report

This paper contributes to the study of holographic entanglement entropy in gravitational anomalies. The introduction of the inner RT surface provides new insights into the holographic description of mixed-state entanglement. However, some sections could benefit from more precise explanations and further elaboration, particularly in the mathematical and physical interpretation of the results. I recommend acceptance of the manuscript after these revisions.

Requested changes

No

Recommendation

Ask for minor revision

  • validity: high
  • significance: good
  • originality: good
  • clarity: good
  • formatting: excellent
  • grammar: excellent

Author:  Qiang Wen  on 2025-05-13  [id 5476]

(in reply to Report 1 on 2025-03-22)

Firstly, we thank the referee very much for a careful reading of our paper and so many useful suggestions, which are important to improve the readability and self-consistence of this paper. In the following, we try to (partially) address the comments of the referee in details. All the adjustments in replying the comments from the referee are marked in blue.

{1. The manuscript occasionally presents complex mathematical derivations without sufficient explanation, which could make it difficult for readers unfamiliar with the specific methods employed (e.g., worldline actions, regulated geodesic chords, etc.).
Recommendation: Provide more detailed explanations for key steps in the mathematical derivations, especially in the sections involving the IRT surface and the replica trick.}

A: In page 11 and 12 we added a brief review on the bulk the replica algorithms in the bulk gravity, and argued that this algorithms could be applied to the IRT surface. Then, since the IRT surface is related to a von Neumann entropy calculated by the replica trick, it is natural to take the IRT surface as (part of) the geometric picture of the timelike entanglement.
In page 13 and 14, we added some background introduction for the fine structure analysis based on the modular slices, and the correspondence between the PEE and geodesic chords. In page 24 and 25, we gave a systematical introduction for the worldline action prescription which computes the CS correction to the holographic entanglement entropy.

{2. Discussing the mixed state correlation and its connection to the IRT surface is interesting but could benefit from a broader generalization. Including a more comprehensive analysis of how these results extend to other non-AdS geometries or more general holographic setups would be helpful.
Recommendation: Consider discussing the potential generalization of the IRT surface and its relevance to more generic holographic models.}

A: In the examples of non-AdS holography, like the warped CFT/(warped) AdS and 3d flat holography, the causal development of an interval is a strip, which doesn’t has the other two tips where the IRT surface anchors. In the warped CFT case, one can still explore the IRT surface by studying the pre-image of the inner horizon in the Rindler bulk, which may not be a geodesic anchored on the boundary in the original AdS3. While for the 3d flat holography case, there is only one horizon in the Rindler, which is just the inner horizon of the Rindler AdS before taking the flat limit. We can also study the IRT surface for asymptotic AdS3 spacetime where the causal development is also in a shape of a diamond, hence we can directly identify the extremal surface anchored on the two tips as the IRT surface. The higher dimensional cases are also very interesting, but more complicated.
We added two paragraphs to discuss possible generalization of our story to study the CS correction to the entropy quantities associated to the inner horizon, to the timelike entanglement entropy and to the holographic entanglement entropy in 3d flat holography. Also the relation between our fine structure using the modular slices and the swing surfaces are briefly mentioned. We don’t intend to discuss this in details as the paper is already too long and we should avoid introducing too many new concepts which may distracts the readers' attention. We gave the references for readers who are particularly interested in this relation.

{3. The physical interpretation of the inner RT surface (IRT surface) and its connection to timelike entanglement entropy is not immediately intuitive. While the authors explain the concept clearly in technical terms, a more accessible explanation for a broader audience might be helpful.
Recommendation: A more intuitive explanation of the significance of the IRT surface, perhaps with more physical analogies, would make the results more accessible.}

A: In page 11 and 12, the replica story we added to the IRT surface implies that, the length of the IRT surface computes the two-point function of twist operators settled at the two endpoints of a timelike interval. Nevertheless, the timelike curve that gives the imaginary part of the timelike entanglement entropy arises when we change from the Euclidean signature to the Lorentzian signature, which has nothing to do with the replica story. We wish to investigate the role of this timelike surface further in the future.

Specific Questions/Concerns:

{a. What is the physical motivation for choosing the inner RT surface over other possible surfaces, particularly in the context of gravitational anomalies?}

A: The first motivation is the observation that the CS correction to the BTZ black hole entropy in TMG is proportional to the area of the inner horizon. On the other hand, the modular Hamiltonian has correction proportional to the modular momentum, and the IRT surface is the set of fixed points under the modular momentum flow. See the first two paragraphs in section 4.3 in page 26.

{b. Could you provide a more detailed explanation of how the timelike entanglement entropy is related to the imaginary part of the holographic entanglement entropy and the role of the inner RT surface in this context?}

A: As we have mentioned in our reply to the main comment 3, the timelike curve that gives the imaginary part of the timelike entanglement entropy arises when we change from the Euclidean signature to the Lorentzian signature, which has nothing to do with the replica story. We wish to investigate the role of this timelike surface further in the future. The statistic interpretation for the timelike entanglement entropy based on reduced density matrix is so far not clear in the literature.

{c. The paper mentions correcting the holographic entanglement entropy due to the Chern-Simons term. How general is this result? Does it hold for all gravitational anomalies, or is it specific to certain anomalies?}

A: Our discussion is confined in the holographic CFT that duals to the TMG. Only in this case, the CS correction is observed to be related to the area of the inner horizon. We clarify this confinement further in the first paragraph in section 4.1, see page 23.

{d. How does the proposed framework for mixed-state correlations (inner EWCS) compare with other methods for dealing with mixed-state holography, such as the swing surface or the island prescription?}

A: See our reply to the main comment 2.

---

## Round 1 · Referee Report · Anonymous (Referee 2) · 2025-4-9

Report

In this paper, the authors study the time like entanglement entropy and the inner RT surface. The time like entanglement entropy is a hot topics in these days. And the inner horizon was also an hot topic many years ago.
The authors get the following results. First, they reproduce the time like entanglement entropy with the inner horizon. Second, they revisit the entanglement entropy of TMG with the inner horizon. In particular, they show that the correction from the topological term can be reproduced by inner RT surface. They also suggest an inner EWCS. These results are quite interesting and inspiring.
Based on these facts, I suggest the paper to be published.

Recommendation

Publish (meets expectations and criteria for this Journal)

---

## Round 2 · Referee Report · Anonymous (Referee 1) · 2025-5-21

Report

The authors have adequately addressed the issues outlined in the last report.

Recommendation

Publish (meets expectations and criteria for this Journal)

---

## Round 2 · List of Changes

The changes are marked in blue. See also the details in our response to the report 1.

---

## Editorial Decision

published